# Quantifying Aleatoric Uncertainty of the Treatment Effect: A Novel Orthogonal Learner

**Valentyn Melnychuk[1],\*, Stefan Feuerriegel[1], Mihaela van der Schaar[2]**
[1]LMU Munich & Munich Center for Machine Learning (MCML), Germany
[2]University of Cambridge & Alan Turing Institute, United Kingdom
\*Correspondence: `melnychuk@lmu.de`

## Abstract

Estimating causal quantities from observational data is crucial for understanding the safety and effectiveness of medical treatments. However, to make reliable inferences, medical practitioners require not only estimating averaged causal quantities, such as the conditional average treatment effect, but also understanding the randomness of the treatment effect as a random variable. This randomness is referred to as *aleatoric uncertainty* and is necessary for understanding the probability of benefit from treatment or quantiles of the treatment effect. Yet, the aleatoric uncertainty of the treatment effect has received surprisingly little attention in the causal machine learning community. To fill this gap, we aim to quantify the aleatoric uncertainty of the treatment effect at the covariate-conditional level, namely, the conditional distribution of the treatment effect (CDTE). Unlike average causal quantities, the CDTE is *not* point identifiable without strong additional assumptions. As a remedy, we employ partial identification to obtain sharp bounds on the CDTE and thereby quantify the aleatoric uncertainty of the treatment effect. We then develop a novel, orthogonal learner for the bounds on the CDTE, which we call AU-learner. We further show that our AU-learner has several strengths in that it satisfies Neyman-orthogonality and, thus, quasi-oracle efficiency. Finally, we propose a fully-parametric deep learning instantiation of our AU-learner.

## 1 Introduction

Estimating causal quantities from observational data is crucial for decision-making in medicine [9, 12, 22, 30, 70]. For example, medical practitioners are interested in estimating the effect of chemotherapy vs. immunotherapy on patient survival from electronic health records to understand the best treatment strategies in cancer care. Here, common estimation targets are *averaged* causal quantities such as the average treatment effect (ATE) and the conditional average treatment effect (CATE), yet averaged causal quantities do not allow for understanding the variability of the treatment effect.

What is needed for the reliability of causal quantities in medicine? To obtain *reliable* causal quantities, one often needs to "move beyond the mean" [44, 68] and consider the inherent randomness in the treatment effect as a random variable. This randomness is referred to as *aleatoric uncertainty* [17, 60, 110]. Quantifying the aleatoric uncertainty of the treatment effect is relevant in medical practice to understand the probability of benefit from treatment [26, 60] and the quantiles and variance of the treatment effect [5, 17, 26, 33, 59]. As an example, averaged quantities such as the CATE would simply suggest a positive effect for some patients, while the probability of benefit from treatment can inform patients about the odds of being negatively affected by the treatment. Hence, aleatoric uncertainty of the treatment effect promises additional, fine-grained insights beyond simple averages.

Methods for quantifying the aleatoric uncertainty of the treatment effect have gained surprisingly little attention in the causal machine learning community. So far, machine learning for treatment

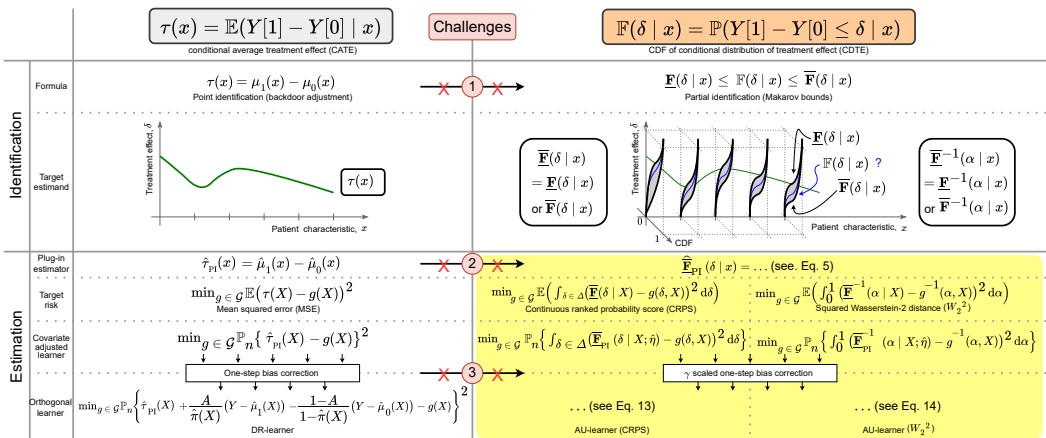

Figure 1: Identification and estimation of the conditional distribution of the treatment effect (CDTE) (=our setting) compared to the (well-studied) identification and estimation of the CATE. In this paper, we focus specifically on the CDF of the CDTE, $\mathbb{P}(Y[1] - Y[0] \leq \delta \mid x)$, shown in orange. Our main contribution relates to the estimation, shown in yellow. However, moving from CATE identification and estimation to our setting comes with important challenges: ① CATE (shown in **green**) is point identifiable but the CDTE is *not* (shown in **blue**); ② there is *no* closed-form expression of the target estimand in terms of nuisance functions and, because of that, CATE learners cannot be directly adapted for estimation; and ③ CATE is an unconstrained target estimand whereas Makarov bounds (shown in gray) are monotonous and contained in the interval $[0, 1]$.

effect estimation was primarily focused on estimating averaged causal quantities [20, 57, 64, 69, 96, 98, 114, 125, 128]. Some research aims to quantify the epistemic uncertainty in treatment effect estimation [51] or the total uncertainty (but without distinguishing the types of uncertainty) [1, 67, 76]. Other works focused on the aleatoric uncertainty of the potential outcomes [13, 31, 65, 66, 94][1] or on contrasts between distributions of potential outcomes (also known as distributional treatment effects) [16, 29, 62, 92, 100].[2] However, to the best of our knowledge, there is no comprehensive meta-learning theory for the estimation of *the aleatoric uncertainty in the treatment effect*.

In this paper, we aim to quantify the aleatoric uncertainty of the treatment effect at the covariate-conditional level in the form of a conditional distribution of the treatment effect (CDTE). Knowing the CDTE would automatically allow one to compute the above-mentioned quantities of aleatoric uncertainty, namely, the probability of benefit from treatment and the quantiles and variance of the treatment effect, at both population and covariate-conditional levels.

## 1.1 Challenges

Yet, the identification and estimation of the CDTE in contrast to CATE come with three **challenges** as follows (see Fig. 1):

Challenge ① is that the CDTE does **not** allow for *point identifiable*, neither in the potential outcomes framework nor in randomized control trials due to the fundamental problem of causal inference as counterfactual outcomes can not be observed [26, 88]. We thus employ *partial identification* [48] to obtain bounds on the CDTE and thereby quantify the aleatoric uncertainty of the treatment effect. Specifically, we focus on Makarov bounds [87, 126, 129] that give sharp bounds for both the cumulative distribution function (CDF) and the quantiles of the CDTE.

Challenge ② is that there is **no** closed-form expression of the target estimand in terms of nuisance functions. Because of this, existing CATE learners cannot be directly adapted to our task of estimating Makarov bounds. For example, there are *no* orthogonal learners in the general setting, and existing

---

[1]In the Neyman-Rubin potential outcomes framework [109], a potential outcome $Y[a]$ refers to the value an outcome variable would take for an individual under a specific treatment or intervention $a$. Each individual has multiple potential outcomes – one for each possible treatment condition – but only one of these outcomes is observed.

[2]Notably, (a) the distributional treatment effects and (b) the distribution of the treatment effect (our setting) are both different interpretationally and inferentially. That is, (a) provide contrasts between distributions of potential outcomes and are point identifiable, and (b) work on the distribution of the difference of potential outcomes and are only partially identifiable. See Appendix A for further details.

approaches only use naïve plug-in estimators/learners. Furthermore, even the derivation of the orthogonal loss is non-trivial as there is no efficient influence function at hand for the Makarov bounds.

Challenge ③ is that CATE is an unconstrained target estimand whereas Makarov bounds are **monotonous** and **contained** in the interval $[0, 1]$. Notably, any constraints of the target estimand could be violated by orthogonal learners [71, 125]. Therefore, an orthogonal learner for Makarov bounds needs to be carefully adapted, especially to perform well in low-sample settings.

## 1.2 Our contributions

In this paper, we develop a novel, orthogonal learner for estimating Makarov bounds which we call *AU-learner*, which allows to quantify the **a**leatoric **u**ncertainty of the treatment effect. Our *AU-learner* addresses all of the above-mentioned challenges ① – ③. Further, our *AU-learner* has several useful theoretical properties, such as satisfying Neyman-orthogonality and, thus, quasi-oracle efficiency [98]. Finally, we propose a flexible, fully-parametric deep learning instantiation of our *AU-learner*. For this, we make use of conditional normalizing flows and call our method AU-CNFs.

To summarize, our contributions are as follows: [3]

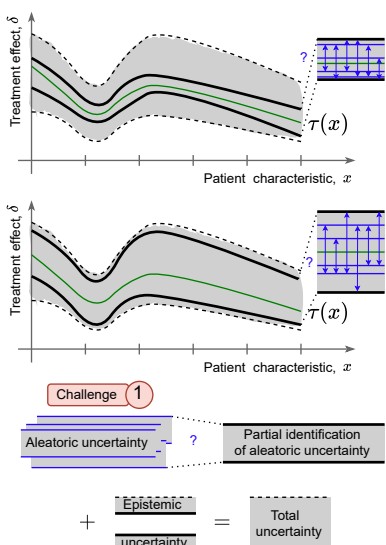

Figure 2: Total uncertainty of the treatment effect can have different sources. Both upper and lower plots have the same total uncertainty but vastly different aleatoric and epistemic components. Yet, aleatoric uncertainty is non-identifiable (see Challenge ①).

1. We derive a novel, orthogonal leaner called *AU-learner* to quantify the aleatoric uncertainty of the treatment effect. For this, we estimate Makarov bounds on the CDF/quantiles of the conditional distribution of the treatment effect (CDTE).
2. We prove several favorable theoretical properties of our *AU-learner*, such as Neyman-orthogonality and, thus, quasi-oracle efficiency.
3. We propose a flexible deep learning instantiation of our *AU-learner* based on conditional normalizing flows, which we call AU-CNFs, and demonstrate its effectiveness over several benchmarks.

## 2 Related Work

In the following, we briefly summarize the existing works on uncertainty quantification in the potential outcomes framework; on the identification of the CDTE; and on the estimation of Makarov bounds. For a more detailed overview of literature, we refer to Appendix A.

**Uncertainty quantification in the potential outcomes framework.** The (total) uncertainty of a predictive model in machine learning is generally split into (a) epistemic and (b) aleatoric uncertainty [41, 50].[4] This split is important, as it informs a decision-maker about the source of uncertainty (see Fig. 2), especially in the context of the potential outcomes framework. (a) Epistemic uncertainty was studied for predictive models targeting at identifiable averaged causal quantities, such as conditional average potential outcomes (CAPOs) and CATE [51, 52]. (b) Aleatoric uncertainty, on the other hand, is *only* identifiable for potential outcomes [88]. Prominent methods focus on interventional (counterfactual) quantities such as: (i) CDF/quantiles estimation [6, 13, 31, 43, 86]; (ii) density estimation [65, 66, 91, 94, 97, 102, 124]; and (iii) distributional distances (also known as distributional treatment effects) estimation [16, 29, 62, 92, 100]. Yet, our work differs substantially from the above, as we aim at inferring the aleatoric uncertainty of the treatment effect, which is only partially identifiable.

---

[3]Code is available at `https://github.com/Valentyn1997/AU-CNFs`.

[4]Epistemic uncertainty [50] relates to the uncertainty of fitting a model on finite data and reduces to zero as data size grows. In contrast, aleatoric uncertainty originates from the inherent randomness of the outcome and is irreducible wrt. data size.

Table 1: Overview of methods for estimating Makarov bounds on the CDF/quantiles of the CDTE.

| Work | Covariate-conditional | Limitations | Estimator/learner | Orthogonal | Instantiation |
|---|---|---|---|---|---|
| Fan et al. [26] | (✓) | — | Plug-in | ✗ | Empirical CDF |
| Ruiz et al. [110] | ✓ | — | Plug-in | ✗ | Conditional mean with hold-out residuals |
| Lee [77], Cui et al. [17] | ✓ | — | Plug-in | ✗ | Kernel density estimation |
| Kallus [60] | ✗ | Binary outcome | A-IPTW/NO | ✓ | Random forest & causal forest |
| Ji et al. [54] | ✗ | Optimization assumptions | A-IPTW/NO | ✓ | Homoskedastic Gaussian linear model |
| Semenova [112] | ✗ | Propensity score is known | IPTW | ✓ | — |
| **Our paper** (*AU-learner*) | ✓ | — | A-IPTW/NO | ✓ | Conditional normalizing flows (AU-CNFs) |

(A-)IPTW: (augmented) inverse propensity of treatment weighted; NO: Neyman-orthogonal

**Identification of the distribution of the treatment effect.** Point identification of the distribution of the treatment effect (or, equivalently, a joint distribution of potential outcomes) is only possible under additional assumptions on the data-generating mechanism. A common example is, e. g., invertibility of latent outcome noise [2, 3, 6]. Other works have rather focused on partial identification [48]. For example, [88] proposed sharp bounds for the distribution of the treatment effect under a monotonicity assumption. Later, assumption-free sharp bounds were proposed for both the joint CDF of potential outcomes [26] and for the variance of treatment effect [5], both known as Fréchet-Hoeffding bounds [39, 49]. Finally, [26, 28, 33] proposed sharp bounds on the CDF/quantiles of the treatment effect without any additional assumptions, so-called *Makarov bounds* [87, 126]. Makarov bounds were further generalized [81, 129] and applied to other settings [17, 25, 27, 36, 59] but different from ours.

**Estimation of Makarov bounds.** Table 1 provides a comparison of key methods for estimating Makarov bounds, at both covariate-conditional and population levels. Existing methods build mainly upon plug-in (single-stage) estimators/learners. Examples are methods tailored for randomized controlled trials [26, 110] and for potential outcomes framework [17, 77, 110]. Crucially, these methods are *not* orthogonal and, thus, are sensitive to the misspecification of the nuisance functions. Nevertheless, we include the latter methods [17, 77] as baselines for our experiments as they use a highly flexible CDF estimator based on kernel density estimators. Some works also developed efficient estimators for Makarov bounds at the population level (analogous to the two-stage Neyman-orthogonal learners at the covariate-conditional level) but only in highly restricted settings. In particular, [60] is restricted to binary outcomes, [112] assumed a known propensity score, and [54] made special optimization assumptions.[5] In addition, all three works [54, 60, 112] suggest fixing a value of $\delta/\alpha$, which the CDF/quantiles of the treatment effect are evaluated at; when our work suggests targeting at several values of $\delta/\alpha$ at once. Therefore, the previous methods are *not* applicable to our general setting of estimating covariate-conditional level Makarov bounds.

**Research gap.** To the best of our knowledge, we are the first to propose an orthogonal learner for estimating Makarov bounds on the CDF/quantiles of conditional distribution of the treatment effect.

## 3 Identification of Distribution of Treatment Effect

**Notation.** Let capital letters $X, A, Y, \Delta$ denote random variables and small letters $x, a, y, \delta$ their realizations from domains $\mathcal{X}, \mathcal{A}, \mathcal{Y}, \Delta$. Let $\mathbb{P}(Z)$ denote a distribution of some random variable $Z$, and let $\mathbb{P}(Z = z)$ be the corresponding density or probability mass function. Furthermore, $\pi(x) = \mathbb{P}(A = 1 \mid X = x)$ is propensity score, $\mu_a(x) = \mathbb{E}(Y \mid X = x, A = a)$ are conditional expectations, and $\mathbb{F}_a(y \mid x) = \mathbb{P}(Y \leq y \mid x, a)$ is a conditional outcome CDF. For other conditional quantities or distributions, we use short forms whenever possible; e. g., $\mathbb{E}(Y \mid x) = \mathbb{E}(Y \mid X = x)$. Further, $\mathbb{P}_n\{f(Z)\} = \frac{1}{n} \sum_{i=1}^n f(z_i)$ is a sample average of a random $f(Z)$, where $n$ is the sample size. We denote linear rectifier functions as $[x]_+ = \max(x, 0)$ and $[x]_- = \min(x, 0)$, and sup/inf convolutions of two functions [116] $f_1(\cdot \mid x), f_2(\cdot \mid x)$ as $(f_1 \overline{\ast} f_2)_{\mathcal{Y}}(\delta \mid x) = \sup_{y \in \mathcal{Y}}\{f_1(y \mid x) - f_2(y - \delta \mid x)\}$ and $(f_1 \underline{\ast} f_2)_{\mathcal{Y}}(\delta \mid x) = \inf_{y \in \mathcal{Y}}\{f_1(y \mid x) - f_2(y - \delta \mid x)\}$.

**Problem setup.** We consider the standard setting of the Neyman–Rubin potential outcomes framework [109]. That is, we have an observational dataset $\mathcal{D}$ with a binary treatment $A \in \mathcal{A} = \{0, 1\}$, potentially high-dimensional covariates $X \in \mathcal{X} \subseteq \mathbb{R}^{d_x}$ and a continuous outcome $Y \in \mathcal{Y} \subseteq \mathbb{R}$. For instance, a typical scenario is in cancer therapy, where the outcome is tumor growth, the treatment is whether chemotherapy is given, and the covariates include patient details like age and sex. We define a joint random variable $Z = (X, A, Y)$. $\mathcal{D} = \{x_i, a_i, y_i\}_{i=1}^n$ is sampled i.i.d. from the observational

---

[5]The work in [54] assumes the possibility of finding feasible Kantorovich dual functions to a target functional (e. g., CDF of the treatment effect). By doing so, the authors are able to infer valid partial identification bounds on very general functionals; yet, the sharpness can not be practically guaranteed.

distribution $\mathbb{P}(Z) = \mathbb{P}(X, Y, A)$, where $n$ is the sample size. The potential outcomes framework then makes three (causal) assumptions, i.e., (1) *consistency*: if $A = a$, then $Y[a] = Y$; (2) *overlap*: $\mathbb{P}(0 \leq \pi(X) \leq 1) = 1$; and (3) *exchangeability*: $A \perp\!\!\!\perp (Y[0], Y[1]) \mid X$.

**Treatment effect distribution.** In this paper, we refer to the *treatment effect* $\Delta = Y[1] - Y[0]$ as a random variable.[6] The CATE is given by $\tau(x) = \mathbb{E}(\Delta \mid x)$, which is identifiable as $\mu_1(x) - \mu_0(x)$ under the causal assumptions (1)–(3). We are interested in identifying a *conditional distribution of the treatment effect (CDTE)*, specifically, its CDF or quantiles:

$$\mathbb{F}(\delta \mid x) = \mathbb{P}(\Delta \leq \delta \mid x) = \mathbb{P}(Y[1] - Y[0] \leq \delta \mid x), \quad \delta \in \Delta \tag{1}$$

$$\mathbb{F}^{-1}(\alpha \mid x) = \inf\{\delta \in \Delta \mid \alpha \leq \mathbb{F}(\delta \mid x)\}, \quad \alpha \in [0, 1]. \tag{2}$$

The CDF and the quantiles of the CDTE are point *non*-identifiable due to the fundamental problem of causal inference, i.e., that the counterfactual outcome, $Y[1 - A]$, is never observed. This is illustrated in Fig. 3, where both conditional potential outcome distributions are identifiable as $\mathbb{P}(Y[a] \mid x) = \mathbb{P}(Y \mid x, a)$; but a conditional joint distribution, $\mathbb{P}(Y[0], Y[1] \mid x)$, and the CDTE, $\mathbb{P}(\Delta \mid x)$, are not.

**Partial identification of the CDTE.** Fan et al. [26] proposed pointwise sharp bounds on the CDF and the quantiles of the CDTE, so-called *Makarov bounds* [87, 126, 129]. Given that the outcome $Y$ is continuous, the Makarov bounds for the CDF of the CDTE are given by linearly rectified sup/inf convolutions [116] of conditional CDFs of potential outcomes:

$$\underline{\mathbf{F}}(\delta \mid x) \leq \mathbb{F}(\delta \mid x) \leq \overline{\mathbf{F}}(\delta \mid x),$$
$$\underline{\mathbf{F}}(\delta \mid x) = [(\mathbb{F}_1 \,\overline{*}\, \mathbb{F}_0)_{\mathcal{Y}}(\delta \mid x)]_+ \quad \text{and} \quad \overline{\mathbf{F}}(\delta \mid x) = 1 + [(\mathbb{F}_1 \,\underline{*}\, \mathbb{F}_0)_{\mathcal{Y}}(\delta \mid x)]_- \,, \tag{3}$$

where $\delta \in \Delta$, and $\mathbb{F}_a(y \mid x)$ is the CDF of $\mathbb{P}(Y[a] \mid x)$. Similarly, Makarov bounds can be formulated for the quantiles of the CDTE:

$$\overline{\mathbf{F}}^{-1}(\alpha \mid x) \leq \mathbb{F}^{-1}(\alpha \mid x) \leq \underline{\mathbf{F}}^{-1}(\alpha \mid x),$$

$$\underline{\mathbf{F}}^{-1}(\alpha \mid x) = \begin{cases} (\mathbb{F}_1^{-1} \,\underline{*}\, \mathbb{F}_0^{-1})_{[\alpha, 1]}(\alpha \mid x), & \text{if } \alpha \neq 0, \\ \mathbb{F}_1^{-1}(0 \mid x) - \mathbb{F}_0^{-1}(1 \mid x), & \text{if } \alpha = 0, \end{cases} \quad \text{and} \quad \overline{\mathbf{F}}^{-1}(\alpha \mid x) = \begin{cases} (\mathbb{F}_1^{-1} \,\overline{*}\, \mathbb{F}_0^{-1})_{[0, \alpha]}(\alpha - 1 \mid x), & \text{if } \alpha \neq 1, \\ \mathbb{F}_1^{-1}(1 \mid x) - \mathbb{F}_0^{-1}(0 \mid x), & \text{if } \alpha = 1, \end{cases} \tag{4}$$

where $\alpha \in [0, 1]$, and $\mathbb{F}_a^{-1}(u \mid x)$ are the quantiles of $\mathbb{P}(Y[a] \mid x)$. Then, under the causal assumptions (1)–(3), conditional distributions of potential outcomes coincide with observed ones, i.e., $\mathbb{P}(Y[a]) = \mathbb{P}(Y \mid x, a)$. Notably, Makarov bounds on the CDFs are CDFs themselves, but these CDFs do not correspond to the solution of the partial identification task (which implies pointwise sharpness). We refer to Appendix B for more illustrations about the inference of the Makarov bounds, the explanation of pointwise sharpness, and Makarov bounds for categorical/mixed-type outcomes.

## 4 An AU-learner for estimating Makarov bounds

In the following, we develop a theory of orthogonal learning for Makarov bounds, which then gives rise to our *AU-learner*. For this, we first review the plug-in learner and its shortcomings. Motivated by this, we then derive two-stage learners. Here, we first present a novel CA-learner in an intermediate step and finally our *AU-learner*. Note that both are novel but we frame our contributions around the *AU-learner* because of favorable theoretical properties. For notation, we use over- and underlines as in, e.g., $\overline{\underline{\mathbf{F}}}(\delta \mid x)$ to refer to the upper and/or lower bound.

### 4.1 Single-stage learners for Makarov bounds

**Plug-in learner.** A naïve way to construct estimators for Makarov bounds [17, 26, 77, 110] is to estimate conditional outcome CDFs, $\widehat{\mathbb{F}}_a(y \mid x)$, and plug them into Eq. (3):

$$\underline{\widehat{\mathbf{F}}}_{\text{PI}}(\delta \mid x) = \left[(\widehat{\mathbb{F}}_1 \,\overline{*}\, \widehat{\mathbb{F}}_0)_{\mathcal{Y}}(\delta \mid x)\right]_+ \quad \text{and} \quad \overline{\widehat{\mathbf{F}}}_{\text{PI}}(\delta \mid x) = 1 + \left[(\widehat{\mathbb{F}}_1 \,\underline{*}\, \widehat{\mathbb{F}}_0)_{\mathcal{Y}}(\delta \mid x)\right]_-. \tag{5}$$

The plug-in learner for the bounds of the quantiles, $\overline{\underline{\widehat{\mathbf{F}}}}_{\text{PI}}^{-1}(\alpha \mid x)$, can be obtained in a similar way but where one uses Eq. (4) and relies on the estimators of either the conditional outcome quantiles or inverse of the estimated conditional outcome CDFs, $\widehat{\mathbb{F}}_a^{-1}(u \mid x)$.

---

[6]Also known as an individual treatment effect. The term 'individual' should not be confused with the term 'covariate-conditional', which refers to the causal quantity and not the random variable itself.

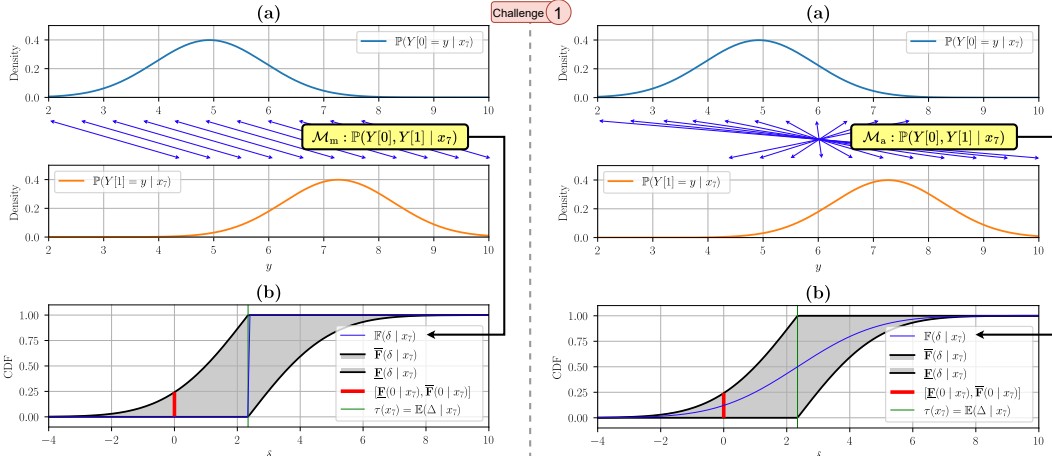

Figure 3: An example showing point non-identifiability of the distribution of the treatment effect based on the $i = 7$-th instance of the semi-synthetic IHDP100 dataset [46]. Shown are two data-generation models, indistinguishable in potential outcomes framework or RCTs, i.e., a monotone, $\mathcal{M}_m$, and an antitone, $\mathcal{M}_a$. For both models we also plot **(a)** conditional densities of potential outcomes, $\mathbb{P}(Y[a] = y \mid x_7)$ and conditional joint laws of potential outcomes, $\mathbb{P}(Y[0], Y[1] \mid x_7)$; and **(b)** corresponding CDFs of the CDTE (shown in **blue**), $\mathbb{F}(\delta \mid x_7) = \mathbb{P}(Y[1] - Y[0] \leq \delta \mid x_7)$, together with Makarov bounds (shown in gray ) and point identifiable CATE (shown in **green**), $\tau(x_7) = \mathbb{E}(\Delta \mid x_7) \approx 2.342$. Non-identifiability of the CDTE is easy to see: Both data-generation models have the same conditional distributions of potential outcomes but different conditional joint laws and, thus, different CDTEs. The latter figures, **(b)**, also demonstrate the bounds on the probability of benefit from treatment (a special case of Makarov bounds), $\mathbb{P}(Y[1] - Y[0] \leq 0 \mid x_7) \in [0, 0.242]$. Hence, Makarov bounds are informative almost everywhere (except $\delta = \tau(x_7)$).

**Shortcomings.** The plug-in learner suffers from two important shortcomings [96]. **(a)** The plug-in learner does *not* account for the selection bias, meaning that $\mathbb{F}_1$ is estimated better for the treated population and $\mathbb{F}_0$ for the untreated. Hence, it could be necessary to *re-weight the loss* wrt. to the propensity score. A remedy is to employ an inverse propensity of treatment weighted (IPTW) learner for both $\mathbb{F}_0$ and $\mathbb{F}_1$ [6, 43] (see Appendix C). **(b)** The plug-in learner does *not* target the Makarov bounds directly but rather the conditional outcome distributions. Therefore, it is *unclear* how to incorporate an inductive bias that *the Makarov bounds are less heterogeneous than either of the conditional outcome CDFs* (i.e., the Makarov bounds can depend on a subset of covariates $X$). The second shortcoming thus motivates our derivation of two-stage learners.

### 4.2 Two-stage learners for Makarov bounds

In order to address the above shortcomings of the plug-in learners, a two-stage learning theory was proposed [15, 35, 72, 96]. Yet, the two-staged learning theory is primarily built for simple target estimands (e.g., CATE) and therefore requires *non-trivial adaptations* by us to extend to Makarov bounds, which we do in the following.

**Working model & target risk.** Our two-stage learners seek to find the best approximation of the ground-truth Makarov bounds, functional target estimands, by a (parametric) *working model* $\mathcal{G}$. Formally, the working model is given $\mathcal{G} = \{g(\delta, x) \mid g : \Delta \times \mathcal{X} \to [0, 1]; g(\cdot, x) \text{is non-decreasing}\}$. The best approximation $g_*$ is then obtained by minimizing a (population) *target risk* via $g_* = \arg\min_{g \in \mathcal{G}} \mathcal{L}(g).$[7] In our setting, $\mathcal{L}$ is chosen as some distributional distance between the target estimands and the working model. Specifically, we use continuous ranked probability score (CRPS) [40, 117] as a target risk for learning Makarov bounds on the CDF via

$$\overline{\mathcal{L}}_{\text{CRPS}}(g) = \mathbb{E}\left(\int_\Delta \left(\overline{\mathbf{F}}(\delta \mid X) - g(\delta, X)\right)^2 \mathrm{d}\delta\right), \tag{6}$$

or squared Wasserstein-2 distance as a target risk for learning Makarov bounds on the quantiles via

$$\overline{\mathcal{L}}_{W_2^2}(g^{-1}) = \mathbb{E}\left(\int_0^1 \left(\overline{\mathbf{F}}^{-1}(\alpha \mid X) - g^{-1}(\alpha, X)\right)^2 \mathrm{d}\alpha\right). \tag{7}$$

---

[7]By postulating a restricted working model class, $\mathcal{G}$, we might compromise on the sharpness and end up having looser bounds. Yet, this looseness bias is only relevant in the infinite data regime. In the finite-sample regime, the feasibility of the low-error estimation is a much more important problem.

The target risks in Eq. (6) and (7) cannot be directly minimized as we do not observe the ground-truth Makarov bounds. Yet, due to the identifiability results in Sec. 3, the Makarov bounds depend on the nuisance functions, $\mathbb{F}_a(y \mid x)$, which can be estimated from the observational data. We perform that in the following.

From now on, we denote the ground-truth nuisance functions as $\eta$ and their estimates as $\hat{\eta}$. Also, we make the dependence on the target risks of the nuisance functions explicit; that is, we write the target risk as $\mathcal{L}(g, \eta)$ and the target estimands as $\overline{\mathbf{F}}(\delta \mid X; \eta)$ and $\overline{\mathbf{F}}^{-1}(\alpha \mid X; \eta)$.

**Covariate-adjusted learner.** A straightforward way to estimate and then minimize the target risk is to plug-in the estimates of conditional outcome CDFs, $\widehat{\mathbb{F}}_a(y \mid x)$, into Eq. (6) and (7), respectively. This yields the so-called *covariate-adjusted (CA) learner*, which aims at minimizing the following losses (empirical risks):

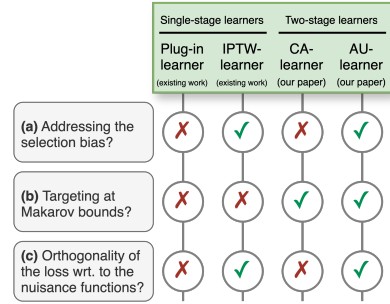

Figure 4: Comparison of learners for estimating Makarov bounds.

$$
\widehat{\overline{\mathcal{L}}}_{\text{PI, CRPS}}(g, \hat{\eta} = (\widehat{\mathbb{F}}_0, \widehat{\mathbb{F}}_1)) = \mathbb{P}_n\Big\{ \int_\Delta \big(\overline{\mathbf{F}}_{\text{PI}}(\delta \mid X; \hat{\eta}) - g(\delta, X)\big)^2 \, d\delta \Big\}, \tag{8}
$$

$$
\widehat{\overline{\mathcal{L}}}_{\text{PI}, W_2^2}(g^{-1}, \hat{\eta} = (\widehat{\mathbb{F}}_0^{-1}, \widehat{\mathbb{F}}_1^{-1})) = \mathbb{P}_n\Big\{ \int_0^1 \big(\overline{\mathbf{F}}_{\text{PI}}^{-1}(\alpha \mid X; \hat{\eta}) - g^{-1}(\alpha, X)\big)^2 \, d\alpha \Big\}, \tag{9}
$$

where we call both $\overline{\mathbf{F}}_{\text{PI}}(\delta \mid x; \hat{\eta}) = \widehat{\overline{\mathbf{F}}}_{\text{PI}}(\delta \mid x)$ and $\overline{\mathbf{F}}_{\text{PI}}^{-1}(\alpha \mid x; \hat{\eta}) = \widehat{\overline{\mathbf{F}}}_{\text{PI}}^{-1}(\alpha \mid x)$ *pseudo-CDFs* and *pseudo-quantiles*, respectively, and where both can be obtained from Eq. (5).

The CA-learner addresses the shortcoming **(b)** of the plug-in learner from above in that loss minimization in the equations above targets directly at Makarov bounds. However, the shortcoming **(a)** of the selection bias still persists. Furthermore, a new shortcoming **(c)** now emerges: The losses can be highly sensitive to badly estimated nuisance functions so that $\widehat{\mathcal{L}}_{\text{PI}}(g, \eta)$ and $\widehat{\mathcal{L}}_{\text{PI}}(g, \hat{\eta})$ differ significantly. Next, we develop an orthogonal learner that addresses all of the shortcomings.

**One-step bias correction.** In order to address the before-mentioned shortcomings of the CA-learner, we employ the concept of (Neyman-)orthogonal losses [15, 35]. Informally, orthogonal losses are first-order insensitive to the misspecification of the nuisance functions, which introduces many favorable properties such as quasi-oracle efficiency [98] and double robustness [127]. The CA-learner losses in Eq. (8) and (9) can be made orthogonal by performing a *one-step bias correction* [10, 63]. The one-step bias correction requires the knowledge of an *efficient influence function*, which has not yet been derived for Makarov bounds ($\rightarrow$ Challenge ②). Hence, the following theorem presents one of our main theoretical results.

**Theorem 1** (Efficient influence function for Makarov bounds). *Let $\mathbb{P}$ denotes $\mathbb{P}(Z) = \mathbb{P}(X, A, Y)$, and let $y_{\mathcal{Y}}^{\overline{*}}(\cdot \mid x)$ and $u_{[\alpha,1]}^*(\cdot \mid x)$ be argmax/argmin sets of the convolutions $(\mathbb{F}_1 \overline{*} \mathbb{F}_0)_{\mathcal{Y}}(\cdot \mid x)$ and $(\mathbb{F}_1^{-1} \underline{*} \mathbb{F}_0^{-1})_{[\alpha,1]}(\cdot - 0 \mid x)$, respectively. Then, under mild conditions on the conditional outcome distributions and for almost all values of $\delta \in \Delta$ and for all values of $\alpha \in (0,1)$ (see Appendix D), average Makarov bounds are pathwise differentiable. Further, the corresponding efficient influence functions, $\phi$, are as follows:*

$$
\phi(\mathbb{E}(\overline{\mathbf{F}}(\delta \mid X)); \mathbb{P}) = \underline{C}(\delta, Z; \eta) + \overline{\mathbf{F}}(\delta \mid X; \eta) - \mathbb{E}(\overline{\mathbf{F}}(\delta \mid X; \eta)) \tag{10}
$$

$$
\phi(\mathbb{E}(\overline{\mathbf{F}}^{-1}(\alpha \mid X)); \mathbb{P}) = \underline{C}^{-1}(\alpha, Z; \eta) + \overline{\mathbf{F}}^{-1}(\alpha \mid X; \eta) - \mathbb{E}(\overline{\mathbf{F}}^{-1}(\alpha \mid X; \eta)),
$$

$$
\underline{C}(\delta, Z; \eta) = I(X; \eta)\Big[\frac{A}{\pi(X)}\big(\mathbb{1}\{Y \leq y^*\} - \mathbb{F}_1(y^* \mid X)\big) - \frac{1-A}{1-\pi(X)}\big(\mathbb{1}\{Y \leq y^* - \delta\} - \mathbb{F}_0(y^* - \delta \mid X)\big)\Big], \tag{11}
$$

$$
\underline{C}^{-1}(\alpha, Z; \eta) = \frac{A}{\pi(X)}\left(\frac{\mathbb{1}\{Y \leq \mathbb{F}_1^{-1}(u^* \mid X)\} - u^*}{\mathbb{P}(Y = \mathbb{F}_1^{-1}(u^* \mid X) \mid X, A = 1)}\right) - \frac{1-A}{1-\pi(X)}\left(\frac{\mathbb{1}\{Y \leq \mathbb{F}_0^{-1}(u^* - \alpha + 0 \mid X)\} - (u^* - \alpha + 0)}{\mathbb{P}(Y = \mathbb{F}_0^{-1}(u^* - \alpha + 0 \mid X) \mid X, A = 0)}\right), \tag{12}
$$

*where $I(X; \eta) = \mathbb{1}\{(\mathbb{F}_1 \overline{*} \mathbb{F}_0)_{\mathcal{Y}}(\delta \mid X) > 0\}$; $y^*$ is some value from the finite set $y_{\mathcal{Y}}^{\overline{*}}(\delta \mid X)$; $u^*$ is some value from the finite set $u_{[\alpha,1]}^*(\alpha \mid X)$; and $\overline{C}(\delta, Z; \eta)$ and $\overline{C}^{-1}(\delta, Z; \eta)$ can be then obtained by swapping the symbols $\{\overline{*}, >, y_{\mathcal{Y}}^{\overline{*}}, u_{[\alpha,1]}^*, -0, +0\}$ to $\{\underline{*}, <, y_{\mathcal{Y}}^{\underline{*}}, u_{[0,\alpha]}^{\overline{*}}, -1, +1\}$.*

*Proof.* See Appendix D.

In the above theorem, we use red color to show the nuisance functions of $\mathbb{P}$ that are influencing the target estimand, i.e., averaged Makarov bounds. Therein, we also provide a Corollary 1, where we derive efficient influence functions for the target risks from Eq. (6) and (7), namely $\phi(\mathcal{L}(g); \mathbb{P})$.

**Note on infinite argmax/argmin sets.** When argmax/argmin sets of the sup/inf-convolutions are infinite, average Makarov bounds are pathwise non-differentiable, and, thus, one-step bias correction is not possible as statistical inference becomes non-regular [47]. This result also holds for other causal quantities that contain sup/inf operators (e.g., for the policy value of the optimal treatment strategy [84, 106]). Although, there exist approaches to perform inference in the non-regular setting [84], we focus solely on the regular setting where pathwise differentiability holds (see Appendix D for a discussion on the generality of such a setting).

**Orthogonal leaner (*AU-learner*).** Given the derived efficient influence function for the target risks, we perform a $\gamma$-scaled one-step bias correction [63] of the CA-learner losses, namely, $\widehat{\mathcal{L}}_{\mathrm{PI}}(g, \hat{\eta}) + \gamma \mathbb{P}_n\{\phi(\mathcal{L}(g); \hat{\mathbb{P}})\}$. The latter then yields our novel orthogonal *AU-learner* (see Corollary 2 in Appendix D). Our *AU-learner* effectively resolves all the above-mentioned shortcomings (see a comparison in Fig. 4). Formally, it aims at minimizing one of the following losses:

$$\widehat{\overline{\mathcal{L}}}_{\mathrm{AU,\,CRPS}}(g, \hat{\eta} = (\hat{\pi}, \widehat{\mathbb{F}}_0, \widehat{\mathbb{F}}_1)) = \mathbb{P}_n\Big\{ \int_{\varDelta} \big(\overline{\mathbf{F}}_{\mathrm{AU}}(\delta, Z; \hat{\eta}, \gamma) - g(\delta, X)\big)^2 \, \mathrm{d}\delta \Big\}, \tag{13}$$

$$\widehat{\overline{\mathcal{L}}}_{\mathrm{AU},W_2^2}(g^{-1}, \hat{\eta} = (\hat{\pi}, \widehat{\mathbb{F}}_0^{-1}, \widehat{\mathbb{F}}_1^{-1})) = \mathbb{P}_n\Big\{ \int_0^1 \big(\overline{\mathbf{F}}_{\mathrm{AU}}^{-1}(\alpha, Z; \hat{\eta}, \gamma) - g^{-1}(\alpha, X)\big)^2 \, \mathrm{d}\alpha \Big\}, \tag{14}$$

$$\overline{\mathbf{F}}_{\mathrm{AU}}(\delta, Z; \hat{\eta}, \gamma) = \overline{\mathbf{F}}_{\mathrm{PI}}(\delta \mid X; \hat{\eta}) + \gamma \overline{C}(\delta, Z; \hat{\eta}) \quad \text{and} \quad \overline{\mathbf{F}}_{\mathrm{AU}}^{-1}(\alpha, Z; \hat{\eta}, \gamma) = \overline{\mathbf{F}}_{\mathrm{PI}}^{-1}(\alpha \mid X; \hat{\eta}) + \gamma \overline{C}^{-1}(\alpha, Z; \hat{\eta}),$$

where $\overline{C}(\delta, Z; \hat{\eta})$ and $\overline{C}^{-1}(\alpha, Z; \hat{\eta})$ are given by Eq. (11) and (12), respectively; and $\gamma \in (0, 1]$ is a scaling hyperparameter. We present a meta-algorithm of our *AU-learner* (with the CRPS target risk) based on cross-fitting in Algorithm 1 (*AU-learner* with the $W_2^2$ target risk follows analogously).

**Scaling hyperparameter.** The scaling hyperparameter $\gamma$ is introduced to tackle Challenge ③ from above, namely, that the pseudo-CDF term, $\overline{\mathbf{F}}_{\mathrm{AU}}(\delta, Z; \hat{\eta}, \gamma)$, is not guaranteed to be a valid CDF for $\gamma > 0$ (both monotonicity wrt. $\delta$ and $[0, 1]$-constraint can be violated).[8] The same happens with the pseudo-quantiles of the *AU-learner*, $\overline{\mathbf{F}}_{\mathrm{AU}}^{-1}(\alpha, Z; \hat{\eta}, \gamma)$, which could be non-monotonous wrt. $\alpha$. The main intuition behind scaling is that it interpolates between the full *AU-learner* ($\gamma = 1$), that has favorable theoretical properties; and the CA-learner ($\gamma = 0$), for which the pseudo-CDFs and pseudo-quantiles are valid CDFs and quantiles, respectively (we refer to Appendix E with visual examples). Hence, scaling mimics a learning rate of a Newton-Raphson method (usually considered as an analogy to the one-step bias correction [34]). We found fixed values for the scaling hyperparameter $\gamma$ to work well in all of our experiments and to improve the low-sample performance of our *AU-learner*.

### 4.3 Theoretical properties of AU-learner

In the following, we formulate our second main theoretical result. For the results to hold, the nuisance functions $\eta = (\pi, \mathbb{F}_0, \mathbb{F}_1)/\eta = (\pi, \mathbb{F}_0^{-1}, \mathbb{F}_1^{-1})$ need to be estimated independently from the second stage model $g/g^{-1}$. This could be done by either assuming a not-too-flexible class of models (such as a Donsker class of estimators and fitting all the models on the same dataset $\mathcal{D}$) or by using a generic approach of cross-fitting [15, 35].

---

**Algorithm 1** *AU-learner* (CRPS) via cross-fitting

---

1: **Input.** Training dataset $\mathcal{D} = \{x_i, a_i, y_i\}_{i=1}^n$, scaling $\gamma \in (0, 1]$, folds $K \geq 2$, $\delta$-grid $\{\delta_j \in \varDelta\}_{j=1}^{n_\delta}$
2: **Output.** Estimator of Makarov bounds $\widehat{\overline{g}}(\delta, x)$
3: **for** $k \in \{1, \dots, K\}$ **do** ▷ *First stage*
4:     Use $\{x_i, a_i, y_i\}_{k-1 \neq (i \bmod K)}$ to fit $\hat{\eta} = (\hat{\pi}, \widehat{\mathbb{F}}_0, \widehat{\mathbb{F}}_1)$
5:     **for** $i : k - 1 = (i \bmod K)$ **do**
6:         Use $\hat{\eta}$ to infer the pseudo-CDF for the $\delta$-grid: $\big\{\overline{\mathbf{F}}_{\mathrm{AU}}(\delta_j, z_i; \hat{\eta})\big\}_{j=1}^{n_\delta}$ (Eq. (13))
7:     **end for**
8: **end for**
9: Fit the working model based on $\delta$-grid: ▷ *Second stage* $\widehat{\overline{g}} = \arg\min_{g \in \mathcal{G}} \widehat{\overline{\mathcal{L}}}_{\mathrm{AU,\,CRPS}}(g, \hat{\eta})$

---

**Theorem 2** (Neyman-orthogonality of AU-learner (informal))**.** *Under the assumptions of the Theorem 1, the following holds for* AU-learner *from Algorithm 1 with the scaling hyperparameter $\gamma = 1$:*

1. **Neyman-orthogonality.** *Losses in Eq.* (13) *and Eq.* (14) *are first-order insensitive wrt. to the misspecification of the nuisance functions.*

---

[8]The issue of pseudo-outcomes violating constraints was also raised wrt. DR-learner for CATE [71, 125] when the outcome space is bounded (e.g., $\mathcal{Y} = [0, 1]$) and for the CDFs of potential outcomes [43]. Yet, the optimal solution remains an open research question.

2. ***Quasi-oracle efficiency.*** *The bias from the misspecification of the nuisance functions is of second order.*

We refer to the Appendix D for the detailed formulation of the theorem and the proof. Notably, the quasi-oracle efficiency [98] means that our *AU-learner* with (sufficiently fast) estimated nuisance functions performs nearly identical to the *AU-learner* with the ground-truth nuisance functions.

## 5 Neural instantiation with AU-CNFs

We now introduce a flexible fully-parametric instantiation of our *AU-learner*, which we call AU-CNFs. Therein, we employ conditional normalizing flows (CNFs) [105, 121] as the main backbone for our *AU-learner*. CNFs are a flexible neural probabilistic method with tractable conditional densities, CDFs, and quantiles. Importantly, all three attributes of the CNFs (densities, CDFs, and quantiles) can be used for both training via back-propagation and inference [115], which makes them a perfect model for both stages of our *AU-learner*.

**Architecture.** The architecture of our AU-CNFs is inspired by interventional normalizing flows (INFs) [94] (a two-stage model for efficient estimation of potential outcomes densities). Our AU-CNFs consist of several CNFs corresponding to the two stages of learning, namely a *nuisance CNF*, which fits the nuisance functions, $(\hat{\pi}, \widehat{\mathbb{F}}_0, \widehat{\mathbb{F}}_1)$ or, equivalently, $(\hat{\pi}, \widehat{\mathbb{F}}_0^{-1}, \widehat{\mathbb{F}}_1^{-1})$; and *two target CNFs*, which implement second stage working models for upper and lower bounds, $\overline{\mathcal{G}}$ and $\underline{\mathcal{G}}$, respectively.

**Training & implementation.** At the first stage of AU-CNFs learning, the nuisance CNF aims at maximizing the conditional log-likelihood and minimizing a binary cross-entropy via a joint loss. Then, we generate the pseudo-CDFs and pseudo-pseudo quantiles, as described in Algorithm 1. Therein, we set the $\delta/\alpha$-grid size to $n_\delta = n_\alpha = 50$ and discretize the $\mathcal{Y}$-space/$[0, 1]$-interval to infer the argmax/argmin values, $\hat{y}^{\overline{\mp}}/\hat{u}^{\overline{\mp}}$. Then, we proceed with the second stage of AU-CNFs learning, where we set $\gamma = 0.25$ for the CRPS loss and $\gamma = 0.01$ for the $W_2^2$ loss. We found the fixed values of $\gamma$ to work well in *all of the synthetic and semi-synthetic experiments* (except for the IHDP100 dataset, where the overlap assumption is violated). We use the same training data for two stages of learning, as (regularized) CNFs as neural networks belong to the Donsker class of estimators [123]. We refer to Appendix F for more details on our AU-CNFs.

## 6 Experiments

We now evaluate our *AU-learner*. For this, we use (semi-)synthetic benchmarks with the ground-truth conditional CDFs/quantiles of potential outcomes, i.e., $\mathbb{F}_a(y \mid x)/\mathbb{F}_a^{-1}(u \mid x)$. In this way, we can infer the ground-truth Makarov bounds and use them for evaluation.

**Evaluation metric.** We use evaluation metrics based on the target risks (as introduced in Sec 4.2). Specifically, we report root continuous ranked probability score (rCRPS) and Wasserstein-2 distance ($W_2$) based on training data (in-sample) and test data (out-sample).

**Baselines.** We compare the proposed hierarchy of learners from Sec. 4 with CNFs as backbones. These are the plug-in learner (**Plug-in CNF**), IPTW-learner (**IPTW-CNF**), CA-learners (**CA-CNFs** (**CRPS** / $W_2^2$)), and AU-learners (**AU-CNFs** (**CRPS** / $W_2^2$)). The only relevant baseline found in the literature is a plug-in learner based on kernel density estimation [17, 77, 110].[9] For this, we used distributional kernel mean embeddings [97] (**Plug-in DKME**), a standard conditional kernel density estimation method. Details on the baselines are in Appendix G.

**Synthetic data.** We adapt the synthetic data generator ($d_x = 2$) from [61, 93] by creating three settings with different conditional outcome distributions: normal, multi-modal, and exponential (see data generation details in Appendix H). In the synthetic data, *the ground-truth Makarov bounds are less heterogeneous than the potential outcomes*, and, hence, two-stage learners are expected to perform the best. We sample $n_{\text{train}} \in \{100; 250; 500; 750; 1000\}$ training and $n_{\text{test}} = 1000$ test datapoints. The out-sample results are in Fig. 5. Here, our AU-CNFs perform the best wrt. rCRPS in the majority of settings and different sizes of training data. We also report the results wrt. $W_2^2$ in Appendix I.

---

[9]The work of [54] focuses on averaged (population-level) Makarov bounds for fixed values of $\delta/\alpha$ and it is unclear how to extend it to our setting.

Table 2: Results for HC-MNIST. Reported: median out-sample rCRPS $\pm$ sd / $W_2 \pm$ sd over 10 runs.

| | B: upper | |
|---|---|---|
| | rCRPS$_{\text{out}}$ | $W_{2\,\text{out}}$ |
| Plug-in CNF | $0.399 \pm 0.162$ | $1.051 \pm 0.514$ |
| IPTW-CNF | $0.385 \pm 0.501$ | $0.986 \pm 1.936$ |
| CA-CNFs (CRPS) | $0.382 \pm 0.045$ | $1.029 \pm 0.119$ |
| CA-CNFs ($W_2^2$) | $0.494 \pm 0.229$ | $1.239 \pm 0.541$ |
| AU-CNFs (CRPS) | $\mathbf{0.347 \pm 0.114}$ | $\mathbf{0.940 \pm 0.389}$ |
| AU-CNFs ($W_2^2$) | $0.422 \pm 0.189$ | $1.065 \pm 0.523$ |

| | B: lower | |
|---|---|---|
| | rCRPS$_{\text{out}}$ | $W_{2\,\text{out}}$ |
| Plug-in CNF | $0.391 \pm 0.104$ | $1.003 \pm 0.333$ |
| IPTW-CNF | $0.447 \pm 0.218$ | $1.141 \pm 0.917$ |
| CA-CNFs (CRPS) | $0.388 \pm 0.051$ | $1.011 \pm 0.146$ |
| CA-CNFs ($W_2^2$) | $0.446 \pm 0.148$ | $1.089 \pm 0.341$ |
| AU-CNFs (CRPS) | $\mathbf{0.357 \pm 0.097}$ | $\mathbf{0.948 \pm 0.344}$ |
| AU-CNFs ($W_2^2$) | $0.415 \pm 0.190$ | $1.025 \pm 0.520$ |

Lower = better (best in bold)

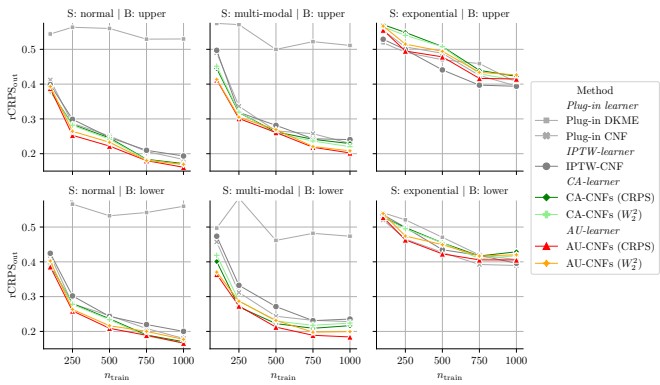

Figure 5: Results for synthetic experiments with varying size of training data, $n_{\text{train}}$, in 3 settings: normal, multi-modal, and exponential. Reported: mean out-sample rCRPS over 20 runs.

**HC-MNIST dataset.** HC-MNIST is a high-dimensional semi-synthetic dataset ($d_x = 785$), built on top of the MNIST dataset [52] (see details in Appendix H). Here, the heterogeneity of Makarov bounds is also smaller than that of potential outcomes, reflecting inductive biases in the real world. We report the out-sample performance of different methods in Table 2 (the Plug-in DKME is omitted due to a too-long runtime). Therein, our AU-CNFs (CRPS) achieve the best performance and, thus, scale well with the dataset size and the dimensionality of covariates. Further, our AU-CNFs ($W_2^2$) improve the performance of CA-CNFs ($W_2^2$). In general, we observe that the loss based on the CDF distance (i. e., CRPS) has a lower variance and is easier to fit. In Appendix I, we additionally report the results for another popular semi-synthetic benchmark, IHDP100 [46, 114].

**Case study.** In Appendix J, we provide a real-world case study based on the observational dataset from [7]. Therein, we demonstrate how our *AU-learner* (AU-CNFs) can be used to estimate the effectiveness of lockdowns during the COVID-19 pandemic. We estimate the probability of benefit from intervention (a special case of Makarov bounds with $\delta = 0$). As expected, we observe a drop in the incidence rate is highly probable after the implementation of a strict lockdown.

## 7 Discussion

**Low-sample & asymptotic performance.** In several experiments, especially in low-sample settings, the CA-learner or even the plug-in approach are performing nearly as well or even sometimes better than the *AU-learner*. This can be expected, as the best low-sample learner and the asymptotically best learner can, in general, be different [20], and there is no single "one-fits-all" data-driven solution to choose the former one [19]. This can be explained by too small dataset sizes or the severe overlap violations (as is the case with the IHDP100 dataset; see Appendix I). Yet, only our quasi-oracle efficient AU-learner offers asymptotic properties in the sense that it is asymptotically closest to the oracle (see Fig. 4). We thus argue for a pragmatic choice in practice (i. e., in the absence of ground-truth counterfactuals or additional RCT data) where our *AU-learner* should be the preferred method for the covariate-conditional Makarov bounds even in low-sample data.

**Future work.** Our work sets a foundation for several extensions to estimate covariate-conditional Makarov bounds. For example, the estimation of the interval probabilities (see Appendix B) of the treatment effect can provide a connection with the existing works on total uncertainty with conformal prediction [1]. Additionally, one might want to study possible extensions of Makarov bounds tailored to high-dimensional outcomes.

**Limitations & broader impact.** Our work is subject to the standard assumptions of the potential outcomes framework. We further make assumptions on the outcome distribution, though these are very mild. Nevertheless, we expect our work to have a positive impact, as it will help to improve the reliability of decision-making in medicine and other safety-critical fields.

**Conclusion.** We are the first to offer a theory of orthogonal learning to quantify the aleatoric uncertainty of the treatment effect at the covariate-conditional level and present flexible neural instantiation.

## Acknowledgments

This paper is supported by the DAAD program "Konrad Zuse Schools of Excellence in Artificial Intelligence", sponsored by the Federal Ministry of Education and Research. Additionally, the authors would like to thank Lars van der Laan, Dennis Frauen, and Alicia Curth for their helpful remarks and comments on the content of this paper. SF also acknowledges funding from the Swiss National Science Foundation (SNSF) via Grant 186932.

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

# A    Extended Related Work

**Partial identification and sensitivity models in the potential outcomes framework.** *Interventional quantities* (such as CAPOs and distributions of potential outcomes) and some counterfactual quantities (e. g., CATE) are point identifiable in potential outcomes framework (see Fig. 6). However, relaxation of the unconfoundedness assumption renders those quantities partially identifiable or even non-identifiable identifiable [89]. In this case, additional sensitivity models can be employed. Examples are the marginal sensitivity model [11, 23, 37, 38, 52, 53, 56, 119] and outcome sensitivity model [11, 104]. Other approaches suggest using instrumental variables [4, 45] or noisy proxy variables [42, 120]. *Counterfactual quantities*, on the other hand, are inherently non-identifiable (e. g., joint distribution of potential outcomes [26], expected counterfactual outcomes of (un)treated [95], and distribution of treatment effect [26]). There is no general approach for deriving sharp bounds for partially identifiable causal quantities, therefore the above-mentioned works are not relevant in our setting.

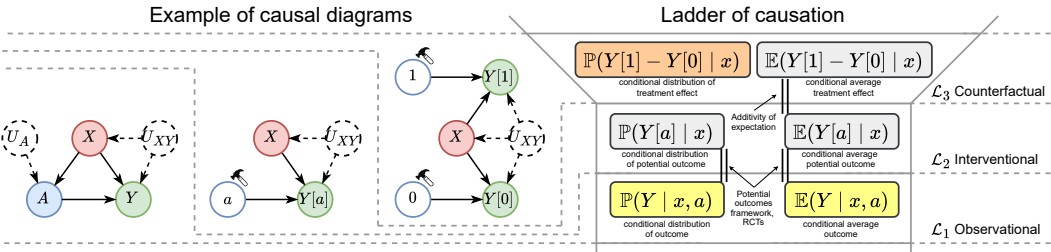

Figure 6: Pearl's ladder of causation [8, 101] containing different observational, interventional, and counterfactual quantities related to the potential outcomes framework. Here, $X$ are covariates, $A$ is a binary treatment, and $Y$ is a continuous outcome. We also plot three exemplar causal diagrams, satisfying the assumptions of the potential outcomes framework, for each layer of causation, correspondingly. Quantities with the light gray background can be expressed with the quantities from lower layers (e. g., conditional average treatment effect) and quantities with yellow background require the information from the same layer. In this paper, we are interested in the CDTE, $\mathbb{P}(Y[1] - Y[0] \mid x)$, shown in orange. Notably, point non-identifiability of the CDTE can be seen with a parallel worlds network (the causal diagram of the counterfactual layer).

**Distributional treatment effects and the distribution of the treatment effect.** There are two seemingly similar notions in the existing literature: (a) distributional treatment effects [16, 29, 62, 92, 100] and (b) the distribution of the treatment effect [6, 26, 28, 33, 88]. However, (a) and (b) have two important differences:

- (i) Interpretation. Distributional treatment effects represent the differences between different distributional aspects of the potential outcomes (e.g., Wasserstein distances, KL-divergence, or the quantile differences) [62]. Hence, they can answer questions like "How are 10% of the worst-possible outcomes with treatment different from the worst 10% of the outcomes without treatment?". Here, the two groups (treated and untreated) of the worst 10% contain, in general, different individuals. This is problematic in many applications like clinical decision support and drug approval. Here, the aim is not to compare individuals from treated vs. untreated groups (where the groups may differ due to various, unobserved reasons). Instead, the aim is to accurately quantify the treatment response for each individual and allow for quantification of the personalized uncertainty of the treatment effect. The latter is captured in the distribution of the treatment effect, which allows us to answer the question about the CDF/quantiles of the treatment effect. For example, we would aim to answer a question like "What are the worst 10% of values of the treatment effect?". Here, we focus on the treatment effect for every single individual. The latter is more complex because we reason about the difference of two potential outcomes simultaneously. Hence, in natural situations when the potential outcomes are non-deterministic, both (a) the distributional treatment effect and (b) the distribution of the treatment effect will lend to very different interpretations, especially in medical practice. In particular, the distribution of the treatment effect (which we study in our paper) is important in medicine, where it allows quantifying the amount of harm/benefit after the treatment [60]. This may warn doctors about

situations where the averaged treatment effects are positive but where the probability of the negative treatment effect is still large.

- (ii) Inference. The efficient inference of the distributional treatment effects only requires the estimation of the relevant distributional aspects of the conditional outcome distributions (e. g., quantiles) and the propensity score [62]. However, in our setting of the bounds on the CDF/quantiles of the treatment effect, we also need to perform sup/inf convolutions of the CDF/quantiles of the conditional outcomes distributions. Hence, while the definitions of (a) the distributional treatment effects and (b) the distribution of the treatment effect appear related, their estimation is very different.

**Alternatives to Makarov bounds.** Potential alternatives to Makarov bounds vary depending on the type of aleatoric uncertainty. For example, explicit or implicit sharp bounds were proposed for:

- The variance of the treatment effect, $\mathrm{Var}(Y[1] - Y[0])$ [5]. Here, the sharp bounds are explicitly given by Fréchet-Hoeffding bounds [39, 49] and are, in general, different from Makarov bounds.
- The interval probabilities, $\mathbb{P}(\delta_1 \leq Y[1] - Y[0] \leq \delta_2)$ [33]. The sharp bounds on the interval probabilities are only defined implicitly and are also, in general, different from Makarov bounds (see Appendix B).

We are not aware of other bounds for measuring aleatoric uncertainty (e. g., kurtosis, skewness, or entropy). Importantly, the above-mentioned bounds on different measures of uncertainty are orthogonal to our work, and we focus on the bounds on the CDF/quantiles of the treatment effect.

**Efficient estimators and orthogonal learners.** Efficient estimation of causal quantities (target estimands) was studied in the scope of (i) *semi-parametric efficient estimation theory* and, more general, (ii) *orthogonal learning theory*. Both theories (i) and (ii) rely on the concept of influence functions (pathwise derivatives) [63, 72, 122], which allow performing a *one-step bias correction* for (i) a plug-in estimator of the target estimand or (ii) the population risk containing target estimand, respectively. (i) Semi-parametric efficient estimation theory [10, 73, 74, 107, 122] provides asymptotically efficient estimators for finite-dimensional target estimands (e. g., average treatment effect (ATE) and average potential outcomes (APOs)). On the other hand, (ii) *orthogonal learning theory* (or debiased ML) [15, 35, 96, 125] was developed for infinitely-dimensional (functional) estimands, like, CATE and CAPO. Orthogonal learning theory (or orthogonal learners) estimates target estimands by minimizing (Neyman-)orthogonal losses that are first-order insensitive to the misspecification of the nuisance functions. Specific examples of orthogonal learners for identifiable quantities include CAPO learners [125] and CATE learners [18, 64, 71, 96, 98].

**Estimation of partially identifiable quantities.** Orthogonal learners were also proposed for bounds on partially identifiable causal quantities. Examples include different interventional quantities such, e. g., in marginal sensitivity model [99], in instrumental variables setting [79], in noisy proxy variables setting [80, 118]. Examples from counterfactual quantities are, e. g., the variance of treatment effect and joint CDF of potential outcomes (Fréchet-Hoeffding bounds) [6]. More generally, estimation and learning for partially identifiable quantities was studied in econometrics as *estimation of intersection bounds* [14, 112, 113]. In this paper, we construct a novel orthogonal learner targeting at Makarov bounds on the CDF/quantiles of the CDTE.

**Total uncertainty quantification.** Total uncertainty in potential outcomes framework can be generally quantified with two approaches: (i) Bayesian methods (e. g., Gaussian processes) and (ii) conformal prediction framework. (i) Bayesian methods [2, 3, 46] allow to infer posterior predictive distributions or credible intervals and for both potential outcomes and treatment effect, but under additional identifiability assumptions (e. g., an assumption of additive latent outcome noise, which renders treatment effect distribution identifiable). (ii) Conformal prediction aims at providing a valid predictive interval and was applied to quantify total uncertainty of predicting potential outcomes [76, 78, 90, 111] and treatment effect [1, 55, 58, 67, 78]. The latter works either make a similar additive latent outcome noise assumption or 'hide' non-identifiable treatment effect into the predictive interval (however, oracle predictive interval could never be reached in this case). We argue that total uncertainty needs to be split into epistemic and aleatoric explicitly to provide insights on the origin of uncertainty [50], especially in our setting where aleatoric uncertainty of the treatment effect itself is partially identifiable (see Fig. 2).

# B  Makarov Bounds

## B.1  Bounds Construction

In the following, we provide an additional intuition on how Makarov bounds are inferred from conditional distributions of the potential outcomes, $\mathbb{P}(Y[a] \mid x)$. For example, Makarov bounds for the CDF of the CDTE are a composition of the sup/inf convolutions, applied to the conditional CDFs of the potential outcomes, and the rectifier functions (see Fig. 7).

Makarov bounds can be inferred (i) analytically or (ii) numerically. (i) Analytic formulas were proposed for very simple distributions (e. g., a normal distribution [26]). At the same time, the (ii) numerical approach is more flexible. For example, we can discretize the $\mathcal{Y}$-space or $[0, 1]$-interval and perform maximization/minimization on that grid. Notably, in our experiments, we infer the ground-truth bounds with the approach (i), when the ground-truth conditional potential outcomes distributions are normal; and (ii) otherwise.

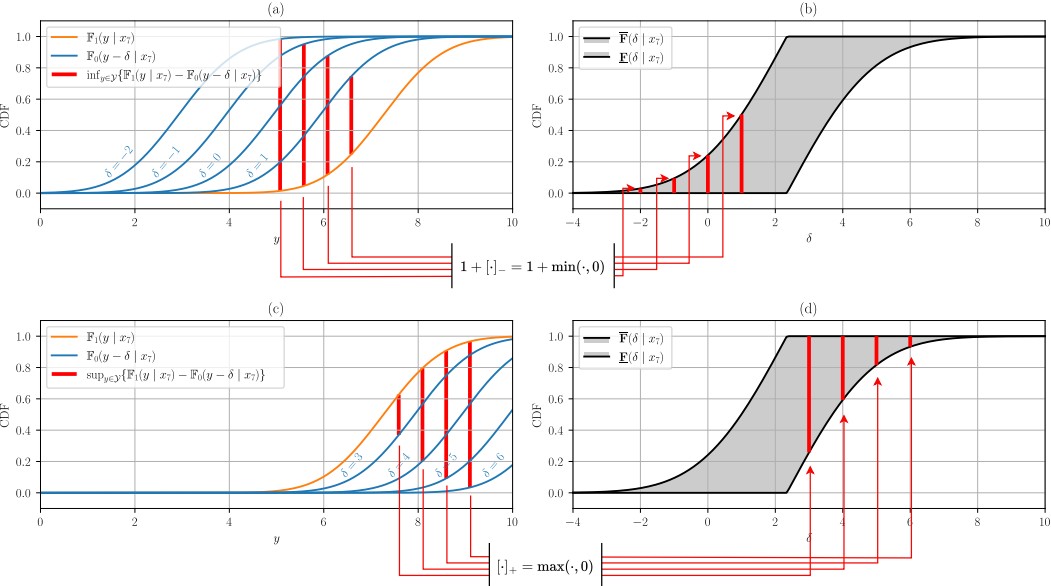

Figure 7: A example of the inference of Makarov bounds on the CDF of the CDTE based on $i = 7$-th instance of the semi-synthetic IHDP100 dataset [46]. The construction of the upper bound is shown in **(a)** and **(b)**; and the lower bound corresponds to subfigures **(c)** and **(d)**. The subfigures on the left, **(a)** and **(c)**, contain the conditional CDFs of both potential outcomes, namely, $\mathbb{F}_0(\cdot \mid x_7)$ and $\mathbb{F}_1(\cdot \mid x_7)$. Therein, the conditional CDF $\mathbb{F}_0(\cdot \mid x_7)$ is shifted wrt. four values of $\delta$. The figures on the right, **(b)** and **(d)**, then demonstrate the corresponding Makarov bounds values for the same four values of $\delta$ (shown in **red**). We also plot the full Makarov bounds with a gray color.

## B.2  Pointwise and Uniformly Sharp Bounds

The sharpness of Makarov bounds proposed in [26] has one important characteristic. Specifically, although the Makarov bounds on the CDFs, $\overline{\mathbf{F}}(\delta \mid X)$, are CDFs themselves, they are not valid solutions to the partial identification task. This can be easily checked, i. e., the expectation wrt. to the Makarov bounds does not coincide with the point-identifiable CATE, $\tau(x) = \mathbb{E}(\Delta \mid x)$:

$$-\int_{-\infty}^{0} \overline{\mathbf{F}}(\delta \mid x) \, \mathrm{d}\delta + \int_{0}^{\infty} \left(1 - \overline{\mathbf{F}}(\delta \mid x)\right) \mathrm{d}\delta < \tau(x) < -\int_{-\infty}^{0} \underline{\mathbf{F}}(\delta \mid x) \, \mathrm{d}\delta + \int_{0}^{\infty} \left(1 - \underline{\mathbf{F}}(\delta \mid x)\right) \mathrm{d}\delta. \tag{15}$$

Therefore, there is an important distinction between so-called *pointwise sharpness* and a *uniform sharpness* [33].

In the case of uniform sharpness, a sharp bound coincides with the solution to the partial identification task. This implies that if a bound is uniformly sharp, then the joint bound on the set of quantities,

evaluated in two (or more) points of $\Delta$ or $[0, 1]$, is also sharp. Many known bounds (e. g., Fréchet-Hoeffding bounds [39, 49], and marginal sensitivity model bounds [23, 37, 38, 119]) are uniformly sharp. Yet, Makarov bounds are only pointwise sharp.

Recent works developed uniformly sharp bounds on the CDF of the CDTE [33]. However, their inference requires a special computational routine for every value of $\delta/\alpha$ wrt. the CDF/quantiles of the CDTE. Their usage is further complicated by the fact that the CDFs of the uniformly sharp bounds correspond to mixed-type discrete/continuous random variables. We show an example of the uniformly sharp bounds on the CDF for $\delta = 0$ in Fig. 8. The uniformly sharp bounds may be useful for more complex aleatoric uncertainty quantities (e. g., interval quantities like $\mathbb{P}(\delta_1 \leq Y[1] - Y[0] \leq \delta_2 \mid x)$) or simultaneous bounds on the variance and the CDF of the CDTE. Nevertheless, in many practical applications, pointwise sharp bounds (Makarov bounds) are enough and we focus on those in our paper.

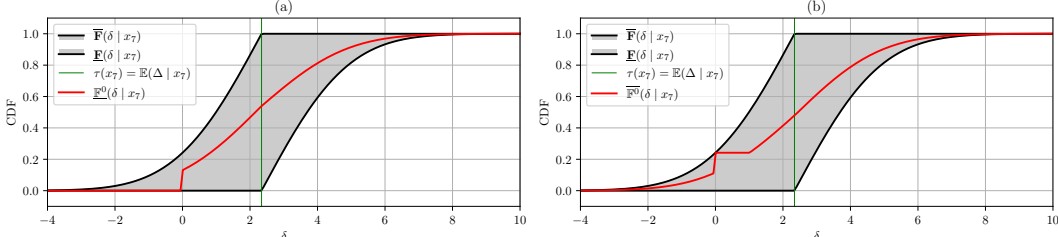

Figure 8: Comparison of pointwise (Makarov) and uniformly sharp bounds on the CDF of the CDTE based on $i = 7$-th instance of the semi-synthetic IHDP100 dataset [46]. Pointwise (Makarov) bounds are shown in gray . Also, we show uniformly sharp bounds inferred for $\mathbb{F}(0 \mid x_7) = \mathbb{P}(Y[1] - Y[0] \leq 0 \mid x_7)$, namely $\overline{\underline{\mathbb{F}}}^0(\delta \mid x_7)$ (shown in **red**). We display the lower bound in **(a)** and the upper bounds in **(b)**. Notably, the expectation wrt. $\mathbb{F}^0(\delta \mid x_7)$ coincides with the CATE, $\tau(x_7) = \mathbb{E}(\Delta \mid x_7)$.

### B.3 Makarov bounds for mixed-type outcomes

In the following, we provide a connection of the Makarov bounds for the continuous outcome $Y$ (the main target of our paper), with the bounds on the fraction of the negatively affected (FNA) from [60]. The FNA is defined for the binary outcome, $Y \in \{0, 1\}$, as

$$\text{FNA} = \mathbb{P}(Y[1] - Y[0] < 0) = \mathbb{P}(Y[1] - Y[0] \leq -1) = \mathbb{E}(\mathbb{F}(\delta = -1 \mid X)). \qquad (16)$$

As suggested by [60], sharp bounds on the FNA are given by

$$\underline{\text{FNA}} \leq \text{FNA} \leq \overline{\text{FNA}},$$
$$\underline{\text{FNA}} = -\mathbb{E}\Big[\min\big(0, \tau(X)\big)\Big] \quad \text{and} \quad \overline{\text{FNA}} = \mathbb{E}\Big[\min\big(\mu_0(X), \, 1 - \mu_1(X)\big)\Big]. \qquad (17)$$

To connect the Makarov bounds for continuous outcome and the bounds on the FNA, we formulate the following remark.

**Remark 1** (Makarov bounds for mixed-type outcome [129]). *Sharp bounds on the CDF of the treatment effect for the mixed-type outcome $Y$ are $\underline{\mathbf{F}}(\delta \mid x) \leq \mathbb{F}(\delta \mid x) \leq \overline{\mathbf{F}}(\delta \mid x)$, which are*

$$\underline{\mathbf{F}}(\delta \mid x) = [(\mathbb{F}_1 \,\overline{*}\, (\mathbb{F}_0 - \mathbb{P}_0))_{\mathbb{R}}(\delta \mid x)]_+ \quad \text{and} \quad \overline{\mathbf{F}}(\delta \mid x) = 1 + [(\mathbb{F}_1 \,\underline{*}\, (\mathbb{F}_0 - \mathbb{P}_0))_{\mathcal{R}}(\delta \mid x)]_-, \tag{18}$$

*where $\mathbb{P}_0 = \mathbb{P}(Y[0] = \cdot \mid x) = \mathbb{P}(Y = \cdot \mid x, A = 0)$ is a probability mass function.*

Notably, for (absolutely) continuous outcome $\mathbb{P}_0 = 0$, Eq. (18) matches Eq. (3) of Sec. 3.

Remark 1 immediately yields the bounds on the FNA, proposed by [60]:

$$\underline{\text{FNA}} = \mathbb{E}\Big[ \max \big( \sup_{y \in \mathbb{R}} \{ \mathbb{P}(Y \le y \mid X, A = 1) - \mathbb{P}(Y \le y + 1 \mid X, A = 0) + \mathbb{P}(Y = y + 1 \mid X, A = 0) \}, 0 \big) \Big] \quad (19)$$

$$= \mathbb{E}\Big[ \max \big( 0, \; \underbrace{0 - \mathbb{P}(Y \le 0 \mid X, A = 0) + \mathbb{P}(Y = 0 \mid X, A = 0)}_{=0}, \; \underbrace{0 - \mathbb{P}(Y \le 0 \mid X, A = 0) + 0}_{\le 0}, \quad (20)$$

$$\mathbb{P}(Y \le 0 \mid X, A = 1) - 1 + \mathbb{P}(Y = 1 \mid X, A = 0), \; \underbrace{\mathbb{P}(Y \le 0 \mid X, A = 1) - 1 + 0}_{\le 0} \big) \Big] \quad (21)$$

$$= \mathbb{E}\Big[ \max \big( 0, -\mathbb{P}(Y = 1 \mid X, A = 1) + \mathbb{P}(Y = 1 \mid X, A = 0) \big) \Big] = \mathbb{E}\Big[ \max \big( 0, -\tau(X) \big) \Big] = -\mathbb{E}\Big[ \min \big( 0, \tau(X) \big) \Big]; \quad (22)$$

$$\overline{\text{FNA}} = 1 + \mathbb{E}\Big[ \min \big( \inf_{y \in \mathbb{R}} \{ \mathbb{P}(Y \le y \mid X, A = 1) - \mathbb{P}(Y \le y + 1 \mid X, A = 0) + \mathbb{P}(Y = y + 1 \mid X, A = 0) \}, 0 \big) \Big] \quad (23)$$

$$= 1 + \mathbb{E}\Big[ \min \big( 0, \; \underbrace{0 - \mathbb{P}(Y \le 0 \mid X, A = 0) + \mathbb{P}(Y = 0 \mid X, A = 0)}_{=0}, \; 0 - \mathbb{P}(Y \le 0 \mid X, A = 0) + 0, \quad (24)$$

$$\mathbb{P}(Y \le 0 \mid X, A = 1) - 1 + \mathbb{P}(Y = 1 \mid X, A = 0), \; \mathbb{P}(Y \le 0 \mid X, A = 1) - 1 + 0 \big) \Big] \quad (25)$$

$$= \mathbb{E}\Big[ \min \big( \mathbb{P}(Y = 1 \mid X, A = 0), \; 1 + \mathbb{P}(Y = 1 \mid X, A = 0) - \mathbb{P}(Y = 1 \mid X, A = 1), \; 1 - \mathbb{P}(Y = 1 \mid X, A = 1) \big) \Big] \quad (26)$$

$$= \mathbb{E}\Big[ \min \big( \mu_0(X), \; \underbrace{\mu_0(X) + 1 - \mu_1(X)}_{\ge \mu_0(X) \text{ and } \ge 1 - \mu_1(X)}, \; 1 - \mu_1(X) \big) \Big] = \mathbb{E}\Big[ \min \big( \mu_0(X), \; 1 - \mu_1(X) \big) \Big]. \quad (27)$$

## C IPTW-learner

In the following, we develop an improved single-stage learner, namely, an inverse propensity of treatment weighted (IPTW) learner. First, we revisit the plug-in learner by defining its estimation objective. Then, we introduce the IPTW-learner, which addresses one of the shortcomings of the plug-in learner. At the end, we mention a surprising property of the IPTW-learner, namely, the orthogonality wrt. to target risk aiming at the potential outcome distributions.

**Empirical risk of the plug-in learner.** We assume that the plug-in learner (Sec. 4.1) uses the estimators of the conditional outcome CDFs, $\widehat{\mathbb{F}}_a$, that aim at minimizing the following empirical risk:

$$\widehat{\mathcal{L}}_{\text{S/T},a}(\hat{\eta} = \widehat{\mathbb{F}}_a) = \mathbb{P}_n\Big\{ \mathbb{1}\{A = a\}\, l\big(Y, \widehat{\mathbb{F}}_a(\cdot \mid X)\big) \Big\}, \tag{28}$$

where $l(\cdot, \cdot) > 0$ is a probabilistic loss (e. g., negative log-likelihood, check score, or CRPS with an empirical CDF). Here, the plug-in learner has two possible variants, namely, S- and T-learner, depending on whether the conditional outcome distribution is learned by a single model or two models [20, 69].

**IPTW-learner.** The IPTW-learner addresses the selection bias of the plug-in learner. For this, it additionally employs an estimated propensity score, $\hat{\pi}$. The estimated propensity score is used to re-weight the original probabilistic loss, $l(\cdot, \cdot)$, in the following way:

$$\widehat{\mathcal{L}}_{\text{IPTW},a}(\hat{\eta} = (\hat{\pi}, \widehat{\mathbb{F}}_a)) = \mathbb{P}_n\left\{ \frac{\mathbb{1}\{A = a\}}{\hat{\pi}_a(X)}\, l\big(Y, \widehat{\mathbb{F}}_a(\cdot \mid X)\big) \right\}, \tag{29}$$

where $\hat{\pi}_a(x) = a\,\hat{\pi}(x) + (1 - a)\,(1 - \hat{\pi}(x))$. The IPTW-learner up-weights the loss of $\widehat{\mathbb{F}}_0$ and $\widehat{\mathbb{F}}_1$ in the treated and untreated populations, respectively.

**Orthogonality of the IPTW-learner.** Interestingly, the IPTW-learner from Eq. (29) can be seen as an *orthogonal learner* targeting at the following risk:

$$\mathcal{L}_a(g) = \mathbb{E}\Big( l\big(Y[a], g(\cdot, X)\big) \Big), \tag{30}$$

where $g(\cdot, x) \in \mathcal{G}$ is a working model defined as in Sec. 4.2. This target risk aims to find the best approximation of the conditional potential outcome distribution, $\mathbb{P}(Y[a] \mid x)$, with the working model, $g \in \mathcal{G}$.

The orthogonality of the IPTW-learner wrt. the target risk aiming at the potential outcome distributions was formally proved in [125]. Therein, the authors notice that the target estimand (e. g., the CDF of one of the potential outcomes) coincides with one of the nuisance functions (i. e., $\mathbb{F}_a$). Informally, the orthogonality follows from the fact that the working model, $g$, simultaneously fits the target estimand and the nuisance function.

The orthogonality of the IPTW-learner can also be seen by (1) performing a one-step bias-correction of Eq. (28) and (2) setting the same estimator for the working model and the nuisance function, i. e., $g = \widehat{\mathbb{F}}_a$. For example, if the probabilistic loss is the CRPS with the empirical CDF, $l(Y, g(\cdot, X)) = \int_{\mathcal{Y}}(\mathbb{1}\{Y \le y\} - g(y, X))^2\, \mathrm{d}y$, the (1) one-step bias-corrected loss is given by [6, 43]

$$\widehat{\mathcal{L}}_{\text{DR},a}(g, \hat{\eta} = (\hat{\pi}, \widehat{\mathbb{F}}_a)) = \mathbb{P}_n\left\{ \frac{\mathbb{1}\{A = a\}}{\hat{\pi}_a(X)}\left( \int_{\mathcal{Y}}(\mathbb{1}\{Y \le y\} - g(y, X))^2\, \mathrm{d}y \right.\right.$$

$$\left.\left. -\int_{\mathcal{Y}}\big(\widehat{\mathbb{F}}_a(y \mid X) - g(y, X)\big)^2\, \mathrm{d}y \right) + \int_{\mathcal{Y}}\big(\widehat{\mathbb{F}}_a(y \mid X) - g(y, X)\big)^2\, \mathrm{d}y \right\}, \tag{31}$$

where $\hat{\pi}_a(x) = a\,\hat{\pi}(x) + (1 - a)\,(1 - \hat{\pi}(x))$. Then, after step (2), $g = \widehat{\mathbb{F}}_a$, we immediately yield the IPTW-learner from Eq. (29). Similarly, for the negative log-likelihood loss (NLL) $l(Y, g(\cdot, X)) = -\log g(Y, X)$, the (1) one-step bias-correction is given by [65, 94]

$$\widehat{\mathcal{L}}_{\text{DR},a}(g, \hat{\eta} = (\hat{\pi}, \widehat{\mathbb{P}})) = \mathbb{P}_n\left\{ \frac{\mathbb{1}\{A = a\}}{\hat{\pi}_a(X)}\left( -\log g(Y, X) + \int_{\mathcal{Y}}\log g(y, X)\, \widehat{\mathbb{P}}(Y = y \mid X, A = a)\, \mathrm{d}y \right) \right.$$

$$\left. -\int_{\mathcal{Y}}\log g(y, X)\, \widehat{\mathbb{P}}(Y = y \mid X, A = a)\, \mathrm{d}y \right\}, \tag{32}$$

where $\hat{\pi}_a(x) = a\,\hat{\pi}(x) + (1-a)\,(1-\hat{\pi}(x))$, and $\widehat{\mathbb{P}}$ is an estimator of the conditional density of the outcome. Again, after (2) setting $g = \widehat{\mathbb{P}}$, it is easy to see that the minimization of one-step bias corrected loss in Eq. (32) is equivalent to the minimization of the IPTW-learner's objective in Eq. (29) (where $\widehat{\mathbb{P}}$ is used in place of $\widehat{\mathbb{F}}_a$). This is possible due to two facts: (1) both entropy terms in Eq. (32), $-\int_{\mathcal{Y}} \log g(y, X)\,g(y, X)\,\mathrm{d}y$, only require the minimization wrt. to the working model $g$ under the logarithm; and (2) these entropies are minimal as the cross-entropy for any distribution is minimal when evaluated with itself.

# D Proofs

In the following, we provide the main theoretical results of our paper. We use the following additional notation: $\delta\{\cdot\}$ is a Dirac delta function, $a \lesssim b$ means there exists $C \geq 0$ such that $a \leq C \cdot b$, and $X_n = o_{\mathbb{P}}(r_n)$ means $X_n/r_n \xrightarrow{p} 0$. Also, in the following theorems, we use red color to show the nuisance functions of $\mathbb{P}$ that are influencing the target estimand.

## D.1 Efficient influence functions

We start with deriving the efficient influence functions for the average Makarov bounds and, afterwards, for the target risks. For that, we make two mild assumptions: (1) one the conditional outcome distributions and (2) another on the set of $\delta$ where linear rectifiers are differentiable. These assumptions allow us to (1) handle sup-/inf-convolutions as max-/min-convolutions with a finite number of argmax/argmin values and to (2) get a derivative of the linear rectifiers.

**Assumption 1** (Finite argument sets). *We assume that the outcome space $\mathcal{Y}$ is compact. Also, we assume that conditional outcome CDFs, $\mathbb{F}_a(y \mid x)$, are continuously differentiable and consist of a finite number of strictly concave / convex regions.*

Assumption 1 implies that sup-/inf-convolutions are achieved by some finite set of values in $\mathcal{Y}$. Furthermore, Assumption 1 is a special case of the margin assumption (Assumption 3.2) from [112]. Yet, we find our version to be more interpretable and many regular distributions satisfy it, e. g., an exponential family, finite mixtures of normal distributions, etc.

**Assumption 2** (Differentiability of linear rectifiers). *Values of $\delta \in \Delta$ are considered to satisfy differentiability of linear rectifier for some $x \in \mathcal{X}$ if $(\mathbb{F}_1 \bar{\ast} \mathbb{F}_0)_{\mathcal{Y}}(\delta \mid x) \neq 0$.*

It is easy to see that, if Assumption 1 holds, the new Assumption 2 will hold for almost all $\delta \in \Delta$ and $x \in \mathcal{X}$, namely, $\mathbb{P}\{(\mathbb{F}_1 \bar{\ast} \mathbb{F}_0)_{\mathcal{Y}}(\delta \mid X) = 0\} = 0$ or, equivalently, $\mathbb{P}\{|\mathbb{F}_1 \bar{\ast} \mathbb{F}_0)_{\mathcal{Y}}(\delta \mid X)| \leq \xi\} \leq \xi$ for some $\xi > 0$. Also, no additional requirements are needed for the values of $\alpha \in [0, 1]$. Notably, the Assumption 2 is analogous to the margin assumptions, e. g., Assumption 3.2 of [84] or Eq. (3) of [112]. Therefore, the formulation of the following theorem is not too restrictive.

**Theorem 1** (Efficient influence function for Makarov bounds). *Let $\mathbb{P}$ denotes $\mathbb{P}(Z) = \mathbb{P}(X, A, Y)$ and let $y_{\mathcal{Y}}^{\bar{\ast}}(\cdot \mid x)$ and $u_{[\alpha,1]}^{*}(\cdot \mid x)$ be argmax/argmin sets of the convolutions $(\mathbb{F}_1 \bar{\ast} \mathbb{F}_0)_{\mathcal{Y}}(\cdot \mid x)$ and $(\mathbb{F}_1^{-1} \underline{\ast} \mathbb{F}_0^{-1})_{[\alpha,1]}(\cdot - 0 \mid x)$, respectively. Then, under the mild assumption of the finite argument sets (Assumption 1), average Makarov bounds are pathwise differentiable for values of $\delta \in \Delta$ that satisfy the differentiability of linear rectifiers (Assumption 2) and for all values of $\alpha \in (0, 1)$. Further, the corresponding efficient influence functions, $\phi$, are as follows:*

$$\phi(\mathbb{E}(\overline{\mathbb{F}}(\delta \mid X)); \mathbb{P}) = \overline{\underline{C}}(\delta, Z; \eta) + \overline{\mathbb{F}}(\delta \mid X; \eta) - \mathbb{E}(\overline{\mathbb{F}}(\delta \mid X; \eta)), \tag{33}$$

$$\phi(\mathbb{E}(\overline{\mathbb{F}}^{-1}(\alpha \mid X)); \mathbb{P}) = \overline{\underline{C}}^{-1}(\alpha, Z; \eta) + \overline{\mathbb{F}}^{-1}(\alpha \mid X; \eta) - \mathbb{E}(\overline{\mathbb{F}}^{-1}(\alpha \mid X; \eta)), \tag{34}$$

$$\underline{C}(\delta, Z; \eta) = \mathbb{1}\{(\mathbb{F}_1 \bar{\ast} \mathbb{F}_0)_{\mathcal{Y}}(\delta \mid X) > 0\}\left[\frac{A}{\pi(X)}\Big(\mathbb{1}\{Y \leq y^*\} - \mathbb{F}_1(y^* \mid X)\Big)\right.$$
$$\left. - \frac{1-A}{1-\pi(X)}\Big(\mathbb{1}\{Y \leq y^* - \delta\} - \mathbb{F}_0(y^* - \delta \mid X)\Big)\right], \tag{35}$$

$$\underline{C}^{-1}(\alpha, Z; \eta) = \frac{A}{\pi(X)}\left(\frac{\mathbb{1}\{Y \leq \mathbb{F}_1^{-1}(u^* \mid X)\} - u^*}{\mathbb{P}(Y = \mathbb{F}_1^{-1}(u^* \mid X) \mid X, A = 1)}\right)$$
$$ - \frac{1-A}{1-\pi(X)}\left(\frac{\mathbb{1}\{Y \leq \mathbb{F}_0^{-1}(u^* - \alpha + 0 \mid X)\} - (u^* - \alpha + 0)}{\mathbb{P}(Y = \mathbb{F}_0^{-1}(u^* - \alpha + 0 \mid X) \mid X, A = 0)}\right), \tag{36}$$

*where $y^*$ is some value from $y_{\mathcal{Y}}^{\bar{\ast}}(\delta \mid X)$; $u^*$ is some value from $u_{[\alpha,1]}^{*}(\alpha \mid X)$; and $\overline{C}(\delta, Z; \eta)$ and $\overline{C}^{-1}(\delta, Z; \eta)$ can be then obtained by swapping the symbols $\{\bar{\ast}, >, y_{\mathcal{Y}}^{\bar{\ast}}, u_{[\alpha,1]}^{*}, -0, +0\}$ to $\{\underline{\ast}, <, y_{\mathcal{Y}}^{*}, u_{[0,\alpha]}^{\bar{\ast}}, -1, +1\}$.*

*Proof.* We start the proof by employing the properties of the efficient influence functions, namely, product rule and chain rule; and some existing efficient influence functions (e. g., for conditional expectation [63]).

The efficient influence function of the lower averaged Makarov bound is as follows:

$$\phi(\mathbb{E}(\underline{\mathbf{F}}(\delta \mid X)); \mathbb{P}) = \mathbb{IF}\Big(\mathbb{E}(\underline{\mathbf{F}}(\delta \mid X))\Big) \tag{37}$$

$$= \int_{\mathcal{X}} \left[ \mathbb{IF}(\underline{\mathbf{F}}(\delta \mid x)) \, \mathbb{P}(X = x) + \underline{\mathbf{F}}(\delta \mid x) \, \mathbb{IF}(\mathbb{P}(X = x)) \right] \mathrm{d}x \tag{38}$$

$$= \int_{\mathcal{X}} \left[ \mathbb{IF}(\underline{\mathbf{F}}(\delta \mid x)) \, \mathbb{P}(X = x) + \underline{\mathbf{F}}(\delta \mid x) \left\{ \delta\{X - x\} - \mathbb{P}(X = x) \right\} \right] \mathrm{d}x \tag{39}$$

$$= \underbrace{\int_{\mathcal{X}} \mathbb{IF}(\underline{\mathbf{F}}(\delta \mid x)) \, \mathbb{P}(X = x) \, \mathrm{d}x}_{(*)} + \underline{\mathbf{F}}(\delta \mid X) - \mathbb{E}(\underline{\mathbf{F}}(\delta \mid X)). \tag{40}$$

Furthermore, the inner term is

$$\mathbb{IF}(\underline{\mathbf{F}}(\delta \mid x)) = \mathbb{IF}\Big( [(\mathbb{F}_1 \,\overline{\ast}\, \mathbb{F}_0)_{\mathcal{Y}}(\delta \mid x)]_+ \Big) \tag{41}$$

$$= \begin{cases} \mathbb{IF}((\mathbb{F}_1 \,\overline{\ast}\, \mathbb{F}_0)_{\mathcal{Y}}(\delta \mid x)), & \text{if} \quad (\mathbb{F}_1 \,\overline{\ast}\, \mathbb{F}_0)_{\mathcal{Y}}(\delta \mid x) > 0, \\ 0, & \text{otherwise.} \end{cases} \tag{42}$$

Then, the efficient influence function of the sup-convolution can be derived under the finite argument sets assumption and using the envelope theorem (we refer to Appendix C.2.6 and C.2.7 of [83] for further details):

$$\mathbb{IF}((\mathbb{F}_1 \,\overline{\ast}\, \mathbb{F}_0)_{\mathcal{Y}}(\delta \mid x)) = \mathbb{IF}\left( \sup_{y \in \mathcal{Y}} \{ \mathbb{F}_1(y \mid x) - \mathbb{F}_0(y - \delta \mid x) \} \right) = \mathbb{IF}(\mathbb{F}_1(y^* \mid x) - \mathbb{F}_0(y^* - \delta \mid x)), \tag{43}$$

where $y^*$ is some value from $y_{\mathcal{Y}}^{\overline{\ast}}(\delta \mid x)$. The latter can be expanded by using product and chain rules and the efficient influence function for the conditional expectation. Specifically, we do the following:

$$\mathbb{IF}(\mathbb{F}_1(y^* \mid x) - \mathbb{F}_0(y^* - \delta \mid x)) = \mathbb{IF}(\mathbb{F}_1(y^* \mid x)) - \mathbb{IF}(\mathbb{F}_0(y^* - \delta \mid x)) \tag{44}$$

$$= \mathbb{IF}(\mathbb{E}(\mathbb{1}\{Y \leq y^*\} \mid x, A = 1)) - \mathbb{IF}(\mathbb{E}(\mathbb{1}\{Y \leq y^* - \delta\} \mid x, A = 0)) \tag{45}$$

$$= \int_{\mathcal{Y}} \left[ \underbrace{\mathbb{IF}(\mathbb{1}\{y \leq y^*\}) \, \mathbb{P}(Y = y \mid x, A = 1)}_{(1)} + \underbrace{\mathbb{1}\{y \leq y^*\} \, \mathbb{IF}(\mathbb{P}(Y = y \mid x, A = 1))}_{(2)} \right] \mathrm{d}y \tag{46}$$

$$- \int_{\mathcal{Y}} \left[ \underbrace{\mathbb{IF}(\mathbb{1}\{y \leq y^* - \delta\}) \, \mathbb{P}(Y = y \mid x, A = 0)}_{(3)} + \underbrace{\mathbb{1}\{y \leq y^* - \delta\} \, \mathbb{IF}(\mathbb{P}(Y = y \mid x, A = 0))}_{(4)} \right] \mathrm{d}y.$$

The terms $(2)$ and $(4)$ yield well-known efficient influence function for CDFs, i. e.,

$$\int_{\mathcal{Y}} \left[ \mathbb{1}\{y \leq y^*\} \, \mathbb{IF}(\mathbb{P}(Y = y \mid x, A = 1)) - \mathbb{1}\{y \leq y^* - \delta\} \, \mathbb{IF}(\mathbb{P}(Y = y \mid x, A = 0)) \right] \mathrm{d}y \tag{47}$$

$$= \left( \frac{A \, \delta\{X - x\}}{\mathbb{P}(X = x, A = 1)} \left( \mathbb{1}\{Y \leq y^*\} - \mathbb{F}_1(y^* \mid x) \right) - \frac{(1 - A) \, \delta\{X - x\}}{\mathbb{P}(X = x, A = 0)} \left( \mathbb{1}\{Y \leq y^* - \delta\} - \mathbb{F}_0(y^* - \delta \mid x) \right) \right). \tag{48}$$

Let us now consider the remaining terms $(1)$ and $(3)$:

$$\int_{\mathcal{Y}} \left[ \mathbb{IF}(\mathbb{1}\{y \leq y^*\}) \, \mathbb{P}(Y = y \mid x, A = 1) - \mathbb{IF}(\mathbb{1}\{y \leq y^* - \delta\}) \, \mathbb{P}(Y = y \mid x, A = 0) \right] \mathrm{d}y \tag{49}$$

$$= \int_{\mathcal{Y}} \left[ - \delta\{y - y^*\} \, \mathbb{IF}(y^*) \, \mathbb{P}(Y = y \mid x, A = 1) + \delta\{y - (y^* - \delta)\} \, \mathbb{IF}(y^*) \, \mathbb{P}(Y = y \mid x, A = 0) \right] \mathrm{d}y \tag{50}$$

$$= - \mathbb{IF}(y^*) \Big( \mathbb{P}(Y = y^* \mid x, A = 1) - \mathbb{P}(Y = y^* - \delta \mid x, A = 0) \Big) = 0, \tag{51}$$

where the last equality holds due to the properties of the argmax, $y^* \in y_{\mathcal{Y}}^{\overline{\ast}}(\delta \mid x)$. Namely, under the necessary condition for a local maximum, we have $\frac{\mathrm{d}}{\mathrm{d}y}(\mathbb{F}_1(y \mid x) - \mathbb{F}_0(y - \delta \mid x)) = \mathbb{P}(Y = y \mid x, A = 1) - \mathbb{P}(Y = y - \delta \mid x, A = 0) = 0$. In the context of the efficient influence functions, this

means that the Makarov bounds are first-order insensitive to the misspecification of argmax/argmin. Interestingly, a similar result was demonstrated for the efficient influence functions of the policy values of the optimal policies [82, 85].

Finally, we obtain an efficient influence function for the lower Makarov bound by expanding the part $(*)$ of Eq. (40):

$$\int_{\mathcal{X}} \mathbb{IF}\big(\underline{\mathbf{F}}(\delta \mid x)\big) \, \mathbb{P}(X = x) \, \mathrm{d}x \tag{52}$$

$$= \int_{\mathcal{X}} \left[ \mathbb{1}\big\{(\mathbb{F}_1 \mp \mathbb{F}_0)_{\mathcal{Y}}(\delta \mid x) > 0\big\} \left( \frac{A \, \delta\{X - x\}}{\mathbb{P}(X = x, A = 1)} \big(\mathbb{1}\{Y \le y^*\} - \mathbb{F}_1(y^* \mid x)\big) \right) \right. \tag{53}$$

$$\left. - \frac{(1 - A) \, \delta\{X - x\}}{\mathbb{P}(X = x, A = 0)} \big(\mathbb{1}\{Y \le y^* - \delta\} - \mathbb{F}_0(y^* - \delta \mid x)\big)\bigg) \right] \mathbb{P}(X = x) \, \mathrm{d}x$$

$$= \underline{C}(\delta, Z; \eta), \tag{54}$$

where $\underline{C}(\delta, Z; \eta)$ is defined in Eq. (35).

The Makarov bounds on the quantiles are derived analogously. However, several last steps are different. The differences start from Eq. (44):

$$\mathbb{IF}\big(\mathbb{F}_1^{-1}(u^* \mid x) - \mathbb{F}_0^{-1}(u^* - \alpha \mid x)\big) = \mathbb{IF}\big(\mathbb{F}_1^{-1}(u^* \mid x)\big) - \mathbb{IF}\big(\mathbb{F}_0^{-1}(u^* - \alpha \mid x)\big), \tag{55}$$

where $u^*$ is some value from $u_{[\alpha, 1]}^*(\alpha \mid x)$.

Now, again, the Makarov bounds for quantiles are first-order insensitive wrt. argmax/argmin values misspecification. Thus, we can employ the efficient influence function for the quantiles [31, 32]:

$$\mathbb{IF}\big(\mathbb{F}_1^{-1}(u^* \mid x)\big) - \mathbb{IF}\big(\mathbb{F}_0^{-1}(u^* - \alpha \mid x)\big) \tag{56}$$

$$= \frac{A \, \delta\{X - x\}}{\mathbb{P}(X = x, A = 1)} \left( \frac{\mathbb{1}\{Y \le \mathbb{F}_1^{-1}(u^* \mid x)\} - u^*}{\mathbb{P}\big(Y = \mathbb{F}_1^{-1}(u^* \mid x) \mid x, A = 1\big)} \right) \tag{57}$$

$$- \frac{(1 - A) \, \delta\{X - x\}}{\mathbb{P}(X = x, A = 0)} \left( \frac{\mathbb{1}\{Y \le \mathbb{F}_0^{-1}(u^* - \alpha \mid x)\} - (u^* - \alpha)}{\mathbb{P}\big(Y = \mathbb{F}_0^{-1}(u^* - \alpha \mid x) \mid x, A = 0\big)} \right). \tag{58}$$

The latter then yields the final part $(*)$ of Eq. (40) for the efficient influence function:

$$\int_{\mathcal{X}} \mathbb{IF}\big(\underline{\mathbf{F}}^{-1}(\alpha \mid x)\big) \, \mathbb{P}(X = x) \, \mathrm{d}x \tag{59}$$

$$= \int_{\mathcal{X}} \left[ \frac{A \, \delta\{X - x\}}{\mathbb{P}(X = x, A = 1)} \left( \frac{\mathbb{1}\{Y \le \mathbb{F}_1^{-1}(u^* \mid x)\} - u^*}{\mathbb{P}\big(Y = \mathbb{F}_1^{-1}(u^* \mid x) \mid x, A = 1\big)} \right) \right. \tag{60}$$

$$\left. - \frac{(1 - A) \, \delta\{X - x\}}{\mathbb{P}(X = x, A = 0)} \left( \frac{\mathbb{1}\{Y \le \mathbb{F}_0^{-1}(u^* - \alpha \mid x)\} - (u^* - \alpha)}{\mathbb{P}\big(Y = \mathbb{F}_0^{-1}(u^* - \alpha \mid x) \mid x, A = 0\big)} \right) \right] \mathbb{P}(X = x) \, \mathrm{d}x$$

$$= \underline{C}^{-1}(\alpha, Z; \eta), \tag{61}$$

where $\underline{C}^{-1}(\alpha, Z; \eta)$ is defined in Eq. (36). $\qquad \square$

**Corollary 1** (Efficient influence functions of the target risks)**.** *Let the Assumption 1 of the Theorem 1 hold. Then, the efficient influence functions, $\phi$, of the target risks in Eq. (6) and Eq. (7) are as follows:*

$$\phi(\overline{\mathcal{L}}_{CRPS}(g); \mathbb{P}) = \int_{\Delta} \left[ 2 \big(\overline{\mathbf{F}}(\delta \mid X; \eta) - g(\delta, X)\big) \, \overline{C}(\delta, Z; \eta) + \big(\overline{\mathbf{F}}(\delta \mid X; \eta) - g(\delta, X)\big)^2 \right] \mathrm{d}\delta \tag{62}$$

$$- \mathbb{E} \left( \int_{\Delta} \big(\overline{\mathbf{F}}(\delta \mid X; \eta) - g(\delta, X)\big)^2 \, \mathrm{d}\delta \right),$$

$$\phi(\overline{\mathcal{L}}_{W_2^2}(g^{-1}); \mathbb{P}) = \int_0^1 \left[ 2 \big(\overline{\mathbf{F}}^{-1}(\alpha \mid X; \eta) - g^{-1}(\alpha, X)\big) \, \overline{C}^{-1}(\alpha, Z; \eta) + \big(\overline{\mathbf{F}}^{-1}(\alpha \mid X; \eta) - g^{-1}(\alpha, X)\big)^2 \right] \mathrm{d}\alpha \tag{63}$$

$$- \mathbb{E} \left( \int_0^1 \big(\overline{\mathbf{F}}^{-1}(\alpha \mid X; \eta) - g^{-1}(\alpha, X)\big)^2 \, \mathrm{d}\alpha \right),$$

*where $\overline{C}(\delta, Z; \eta)$ and $\overline{C}^{-1}(\alpha, Z; \eta)$ are defined in Eq. (35) and (36), respectively.*

*Proof.* We start by using the properties of the efficient influence function, namely, chain and product rules. Then, the efficient influence function for the CRPS risk of the lower bound is as follows

$$\phi(\underline{\mathcal{L}}_{\text{CRPS}}(g); \mathbb{P}) = \mathbb{IF}\left(\mathbb{E}\left(\int_{\Delta}\left(\underline{\mathbf{F}}(\delta \mid X) - g(\delta, X)\right)^2 d\delta\right)\right) \tag{64}$$

$$= \underbrace{\int_{\mathcal{X}} \mathbb{IF}\left(\int_{\Delta}\left(\underline{\mathbf{F}}(\delta \mid x) - g(\delta, x)\right)^2 d\delta\right)\mathbb{P}(X = x)\,dx}_{(*)} + \int_{\Delta}\left(\underline{\mathbf{F}}(\delta \mid X) - g(\delta, X)\right)^2 d\delta \tag{65}$$

$$- \mathbb{E}\left(\int_{\Delta}\left(\underline{\mathbf{F}}(\delta \mid X) - g(\delta, X)\right)^2 d\delta\right). \tag{66}$$

We then note, that the inner term of the $(*)$ can be expanded as:

$$\mathbb{IF}\left(\int_{\Delta}\left(\underline{\mathbf{F}}(\delta \mid x) - g(\delta, x)\right)^2 d\delta\right) = \int_{\Delta}\mathbb{IF}\left(\left(\underline{\mathbf{F}}(\delta \mid x) - g(\delta, x)\right)^2\right)d\delta$$

$$= 2\int_{\Delta}\left(\underline{\mathbf{F}}(\delta \mid x) - g(\delta, x)\right)\mathbb{IF}\left(\underline{\mathbf{F}}(\delta \mid x)\right)d\delta. \tag{67}$$

Finally, the term $(*)$ of Eq. (65) equals to

$$\int_{\mathcal{X}}\mathbb{IF}\left(\int_{\Delta}\left(\underline{\mathbf{F}}(\delta \mid x) - g(\delta, x)\right)^2 d\delta\right)\mathbb{P}(X = x)\,dx \tag{68}$$

$$= 2\int_{\mathcal{X}}\int_{\Delta}\left(\underline{\mathbf{F}}(\delta \mid x) - g(\delta, x)\right)\mathbb{IF}\left(\underline{\mathbf{F}}(\delta \mid x)\right)\mathbb{P}(X = x)\,d\delta\,dx \tag{69}$$

$$= 2\int_{\Delta}\left(\underline{\mathbf{F}}(\delta \mid X) - g(\delta, X)\right)\underline{C}(\delta, Z; \eta)\,d\delta. \tag{70}$$

The derivations for the upper bound and for the $W_2^2$ target risk is fully analogous. $\qquad\square$

**Corollary 2** (One-step bias-corrected estimator of the target risks). *Let the Assumption 1 of the Theorem 1 hold. Then, the $\gamma$-scaled one-step bias-corrected estimator of the target risks Eq. (6) and Eq. (7) is given by the following:*

$$\widehat{\underline{\mathcal{L}}}_{AU,\,CRPS}(g, \hat{\eta} = (\hat{\pi}, \widehat{\mathbb{F}}_0, \widehat{\mathbb{F}}_1)) = \mathbb{P}_n\left\{\int_{\Delta}\left(\underline{\mathbf{F}}_{AU}(\delta, Z; \hat{\eta}, \gamma) - g(\delta, X)\right)^2 d\delta\right\}, \tag{71}$$

$$\widehat{\underline{\mathcal{L}}}_{AU,W_2^2}(g^{-1}, \hat{\eta} = (\hat{\pi}, \widehat{\mathbb{F}}_0^{-1}, \widehat{\mathbb{F}}_1^{-1})) = \mathbb{P}_n\left\{\int_0^1\left(\underline{\mathbf{F}}_{AU}^{-1}(\alpha, Z; \hat{\eta}, \gamma) - g^{-1}(\alpha, X)\right)^2 d\alpha\right\}, \tag{72}$$

$$\underline{\mathbf{F}}_{AU}(\delta, Z; \hat{\eta}, \gamma) = \underline{\mathbf{F}}_{PI}(\delta \mid X; \hat{\eta}) + \gamma\,\underline{C}(\delta, Z; \hat{\eta}) \quad \text{and} \quad \underline{\mathbf{F}}_{AU}^{-1}(\alpha, Z; \hat{\eta}, \gamma) = \underline{\mathbf{F}}_{PI}^{-1}(\alpha \mid X; \hat{\eta}) + \gamma\,\underline{C}^{-1}(\alpha, Z; \hat{\eta}),$$

*where $\underline{C}(\delta, Z; \hat{\eta})$ and $\underline{C}^{-1}(\alpha, Z; \hat{\eta})$ are given by Eq. (35) and (36), respectively; and $\gamma \in (0, 1]$ is a scaling hyperparameter.*

*Proof.* The $\gamma$-scaled one-step bias-corrected estimator of the CRPS target risk proceeds as follows:

$$\widehat{\underline{\mathcal{L}}}_{PI,\,CRPS}(g, \hat{\eta}) + \gamma\mathbb{P}_n\left\{\phi(\underline{\mathcal{L}}_{\text{CRPS}}(g); \widehat{\mathbb{P}})\right\} \tag{73}$$

$$= \mathbb{P}_n\left\{\int_{\Delta}\left(\underline{\mathbf{F}}_{PI}(\delta, Z; \hat{\eta}, \gamma) - g(\delta, X)\right)^2 d\delta\right\} \tag{74}$$

$$+ \gamma\mathbb{P}_n\left\{\int_{\Delta}\left[2\left(\underline{\mathbf{F}}_{PI}(\delta \mid X; \hat{\eta}) - g(\delta, X)\right)\underline{C}(\delta, Z; \hat{\eta}) + \left(\underline{\mathbf{F}}_{PI}(\delta \mid X; \hat{\eta}) - g(\delta, X)\right)^2\right]d\delta\right\}$$

$$- \gamma\mathbb{P}_n\left\{\int_{\Delta}\left(\underline{\mathbf{F}}_{PI}(\delta \mid X; \hat{\eta}) - g(\delta, X)\right)^2 d\delta\right\} = \tag{75}$$

$$= \mathbb{P}_n\left\{\int_{\Delta}\left(\underline{\mathbf{F}}_{PI}(\delta, Z; \hat{\eta}, \gamma) - g(\delta, X)\right)^2 d\delta\right\} + \gamma\mathbb{P}_n\left\{\int_{\Delta}2\left(\underline{\mathbf{F}}_{PI}(\delta \mid X; \hat{\eta}) - g(\delta, X)\right)\underline{C}(\delta, Z; \hat{\eta})\,d\delta\right\}. \tag{76}$$

The minimization of the latter wrt. $g$ is then equivalent to the minimization of

$$\mathbb{P}_n\left\{\int_\Delta \left(\overline{\mathbf{F}}_{\text{PI}}(\delta \mid X; \hat{\eta}) + \gamma\,\overline{C}(\delta, Z; \hat{\eta}) - g(\delta, X)\right)^2 \mathrm{d}\delta\right\}. \tag{77}$$

The proof for the $\gamma$-scaled one-step bias-corrected estimator of the $W_2^2$ target risk is analogous. $\quad\square$

### D.2 Neyman-orthogonality and quasi-oracle efficiency

Now, we proceed with the second main theoretical result. Here, we use additional notation.

**Definition 1** (Neyman-orthogonality [35, 96]). *A risk $\mathcal{L}$ is called Neyman-orthogonal if its pathwise cross-derivative equals to zero, namely,*

$$D_\eta D_g \mathcal{L}(g_*, \eta)[g - g_*, \hat{\eta} - \eta] = 0 \quad \text{for all } g \in \mathcal{G}, \tag{78}$$

*where $D_f F(f)[h] = \frac{\mathrm{d}}{\mathrm{d}t}F(f + th)|_{t=0}$ and $D_f^k F(f)[h_1, \ldots, h_k] = \frac{\partial^k}{\partial t_1 \ldots \partial t_k}F(f + t_1 h_1 + \cdots + t_k h_k)|_{t_1 = \cdots = t_k = 0}$ are pathwise derivatives [35], $g_* = \arg\min_{g \in \mathcal{G}} \mathcal{L}(g, \eta)$, and $\eta$ is the ground-truth nuisance function.*

Informally, this definition means that the risk is first-order insensitive wrt. to the misspecification of the nuisance functions. Notably, the pathwise derivative in the direction of the Dirac delta distribution coincides with the efficient influence function [34, 63], i. e., $D_{\mathbb{P}} F(\mathbb{P})[\delta\{Z - \cdot\} - \mathbb{P}(Z = \cdot)] = \phi(F(\mathbb{P}); \mathbb{P})$, where $\mathbb{P}(Z = \cdot)$ is the PDF of the $\mathbb{P}(Z)$.

**Theorem 2** (Neyman-orthogonality of AU-learner). *Under the assumptions of the Theorem 1, the following holds for* AU-learner *from Algorithm 1 with the scaling hyperparameter $\gamma = 1$:*

1. **Neyman-orthogonality.** *Population versions of the empirical risks in Eq. (71) and Eq. (71) are first-order insensitive wrt. to the misspecification of the nuisance functions, i. e.,*

$$D_\eta D_g \overline{\mathcal{L}}_{AU,CRPS}(g_*, \eta)[g - g_*, \hat{\eta} - \eta] = 0 \quad \text{for all } g \in \mathcal{G}, \tag{79}$$

$$D_\eta D_g \overline{\mathcal{L}}_{AU,W_2^2}(g_*^{-1}, \eta)[g^{-1} - g_*^{-1}, \hat{\eta} - \eta] = 0 \quad \text{for all } g^{-1} : g \in \mathcal{G}, \tag{80}$$

*where $g_* = \arg\min_{g \in \mathcal{G}} \overline{\mathcal{L}}_{AU,CRPS}(g, \eta)$ and $g_*^{-1} = \arg\min_{g^{-1}:g \in \mathcal{G}} \overline{\mathcal{L}}_{AU,W_2^2}(g^{-1}, \eta)$.*

2. **Quasi-oracle efficiency.** *The bias from the misspecification of the nuisance functions is of second order. Specifically, the following two inequalities hold:*

$$\begin{aligned}
\left\|\widehat{g} - g_*\right\|_{2,CRPS}^2 &\lesssim \mathcal{L}_{AU,CRPS}(\widehat{g}, \hat{\eta}) - \mathcal{L}_{AU,CRPS}(g_*, \hat{\eta}) \\
&\quad + \|\pi - \hat{\pi}\|_{L_2}^2 \left\|\mathbb{F}_1 \circ \hat{y}^* - \hat{\mathbb{F}}_1 \circ \hat{y}^*\right\|_{2,CRPS}^2 \\
&\quad + \|\pi - \hat{\pi}\|_{L_2}^2 \left\|\mathbb{F}_0 \circ (\hat{y}^* - \cdot) - \hat{\mathbb{F}}_0 \circ (\hat{y}^* - \cdot)\right\|_{2,CRPS}^2 \\
&\quad + \|y^* - \hat{y}^*\|_{4,CRPS}^4 + \left\|\mathbb{F}_1 \circ y^* - \hat{\mathbb{F}}_1 \circ \hat{y}^*\right\|_{4,CRPS}^4 + \left\|\mathbb{F}_0 \circ (y^* - \cdot) - \hat{\mathbb{F}}_0 \circ (\hat{y}^* - \cdot)\right\|_{4,CRPS}^4 \\
&\quad + \left\|\mathbb{F}_1 \circ y^* - \hat{\mathbb{F}}_1 \circ \hat{y}^*\right\|_{2,CRPS}^2 \left\|\mathbb{F}_0 \circ (y^* - \cdot) - \hat{\mathbb{F}}_0 \circ (\hat{y}^* - \cdot)\right\|_{2,CRPS}^2,
\end{aligned} \tag{81}$$

$$\begin{aligned}
\left\|\widehat{g}^{-1} - g_*^{-1}\right\|_{2,W_2^2}^2 &\lesssim \mathcal{L}_{AU,W_2^2}(\widehat{g}^{-1}, \hat{\eta}) - \mathcal{L}_{AU,W_2^2}(g_*^{-1}, \hat{\eta}) \\
&\quad + \|\pi - \hat{\pi}\|_{L_2}^2 \left\|\mathbb{F}_1^{-1} \circ \hat{u}^* - \hat{\mathbb{F}}_1^{-1} \circ \hat{u}^*\right\|_{2,W_2^2}^2 \\
&\quad + \|\pi - \hat{\pi}\|_{L_2}^2 \left\|\mathbb{F}_0^{-1} \circ (\hat{u}^* - \cdot + 0) - \hat{\mathbb{F}}_0^{-1} \circ (\hat{u}^* - \cdot + 0)\right\|_{2,W_2^2}^2 \\
&\quad + \|u^* - \hat{u}^*\|_{4,W_2^2}^4,
\end{aligned} \tag{82}$$

*where $\widehat{g} = \arg\min_{g \in \mathcal{G}} \mathcal{L}_{AU,CRPS}(g, \hat{\eta})$; $\widehat{g}^{-1} = \arg\min_{g^{-1}:g \in \mathcal{G}} \mathcal{L}_{AU,W_2^2}(g^{-1}, \hat{\eta})$; $\|f(Z)\|_{L_2} = \sqrt{\mathbb{E}(f(Z))^2}$; $\|f(\delta, Z)\|_{k,CRPS} = (\mathbb{E}(\int_\Delta |f(\delta, Z)|^k \mathrm{d}\delta))^{1/k}$; $\|f(\alpha, Z)\|_{k,W_2^2} = $*

$(\mathbb{E}(\int_0^1 |f(\alpha, Z)|^k \, d\alpha))^{(1/k)}$; $I(X; \eta) = \mathbb{1}\{(\mathbb{F}_1 * \mathbb{F}_0)_{\mathcal{Y}}(\delta \mid X) > 0\}$; $y^*$ *is some value from* $y_{\mathcal{Y}}^{\overline{*}}(\delta \mid X)$; $\hat{y}^*$ *is some value from* $\hat{y}_{\mathcal{Y}}^{\overline{*}}(\delta \mid X)$; $u^*$ *is some value from* $u_{[\alpha,1]}^{\overline{*}}(\alpha \mid X)$; *and* $\hat{u}^*$ *is some value from* $\hat{u}_{[\alpha,1]}^{\overline{*}}(\alpha \mid X)$. *The inequalities corresponding to the upper Makarov bound, can be obtained by swapping the symbols* $\{\overline{*}, >, y_{\mathcal{Y}}^{\overline{*}}, u_{[\alpha,1]}^{\overline{*}}, -0, +0\}$ *to* $\{\underline{*}, <, y_{\mathcal{Y}}^*, u_{[0,\alpha]}^{\overline{*}}, -1, +1\}$. *Furthermore, if the nuisance functions are estimated sufficiently fast, i.e.,* $\|y^* - \hat{y}^*\|_{4,CRPS} = o_{\mathbb{P}}(n^{-1/4})$; $\left\|\mathbb{F}_1 \circ y^* - \hat{\mathbb{F}}_1 \circ \hat{y}^*\right\|_{4,CRPS} = o_{\mathbb{P}}(n^{-1/4})$; $\left\|\mathbb{F}_0 \circ (y^* - \cdot) - \hat{\mathbb{F}}_0 \circ (\hat{y}^* - \cdot)\right\|_{4,CRPS} = o_{\mathbb{P}}(n^{-1/4})$; $\|\hat{\pi} - \pi\|_{L_2} \left\|\mathbb{F}_1 \circ \hat{y}^* - \hat{\mathbb{F}}_1 \circ \hat{y}^*\right\|_{2,CRPS} = o_{\mathbb{P}}(n^{-1/2})$; *and* $\|\hat{\pi} - \pi\|_{L_2} \left\|\mathbb{F}_0 \circ (\hat{y}^* - \cdot) - \hat{\mathbb{F}}_0 \circ (\hat{y}^* - \cdot)\right\|_{2,CRPS} = o_{\mathbb{P}}(n^{-1/2})$, *then the* AU-*learner (CRPS) achieves the quasi-oracle property (analogous result holds for* AU-*learner (* $W_2^2$ *)). This means that the estimation error of the second stage with the estimated nuisance functions behaves in the same way as if the ground-truth nuisance functions were used.*

*Proof.* *1. Neyman-orthogonality.* The Neyman-orthogonality follows by the construction of the *AU-learner* as a one-step bias-corrected estimator. Specifically, it is easy to verify that the pathwise cross-derivative from Eq. (78) is equal to zero.

Let us consider the CRPS risk. First, we find a pathwise derivative wrt. working model $g$:

$$D_g \overline{\mathcal{L}}_{\text{AU,CRPS}}(g_*, \eta)[g - g_*] = \frac{d}{dt} \mathbb{E}\Big[\int_\Delta \big(\overline{\mathbf{F}}_{\text{AU}}(\delta, Z; \eta, \gamma) - g_*(\delta, X) - t(g(\delta, X) - g_*(\delta, X))\big)^2 \, d\delta\Big]\Big|_{t=0} \tag{83}$$

$$= -2\mathbb{E}\Big[\int_\Delta \big(\overline{\mathbf{F}}_{\text{AU}}(\delta, Z; \eta, \gamma) - g_*(\delta, X) - t(g(\delta, X) - g_*(\delta, X))\big) \big(g(\delta, X) - g_*(\delta, X)\big) \, d\delta\Big]\Big|_{t=0} \tag{84}$$

$$= -2\mathbb{E}\Big[\int_\Delta \big(\overline{\mathbf{F}}_{\text{AU}}(\delta, Z; \eta, \gamma) - g_*(\delta, X)\big) \big(g(\delta, X) - g_*(\delta, X)\big) \, d\delta\Big]. \tag{85}$$

Then, we find derivatives wrt. to different nuisance functions (e.g., the propensity score):

$$D_\pi D_g \overline{\mathcal{L}}_{\text{AU,CRPS}}(g_*, \eta)[g - g_*, \hat{\pi} - \pi] \tag{86}$$

$$= \frac{d}{dt} - 2\mathbb{E}\Big[\int_\Delta \big(\overline{\mathbf{F}}_{\text{AU}}(\delta, Z; \eta = (\pi + t(\hat{\pi} - \pi), \mathbb{F}_0, \mathbb{F}_1), \gamma) - g_*(\delta, X)\big) \big(g(\delta, X) - g_*(\delta, X)\big) \, d\delta\Big]\Big|_{t=0} \tag{87}$$

$$= \frac{d}{dt} - 2\mathbb{E}\Big[\int_\Delta \gamma \overline{C}(\delta, Z; \eta = (\pi + t(\hat{\pi} - \pi), \mathbb{F}_0, \mathbb{F}_1)) \big(g(\delta, X) - g_*(\delta, X)\big) \, d\delta\Big]\Big|_{t=0} \tag{88}$$

$$= 2\gamma \, \mathbb{E}\Big[\int_\Delta I(X; \eta) \Big[\frac{A(\hat{\pi}(X) - \pi(X))}{(\pi(X))^2} \big(\mathbb{1}\{Y \leq y^*\} - \mathbb{F}_1(y^* \mid X)\big) \tag{89}$$

$$+ \frac{(1 - A)(\hat{\pi}(X) - \pi(X))}{(1 - \pi(X))^2} \big(\mathbb{1}\{Y \leq y^* - \delta\} - \mathbb{F}_0(y^* - \delta \mid X)\big)\Big] \big(g(\delta, X) - g_*(\delta, X)\big) \, d\delta\Big] \tag{90}$$

$$= 2\gamma \, \mathbb{E}_X\Big[\int_\Delta I(X; \eta) \, \mathbb{E}\Big[\frac{A}{(\pi(X))^2} \big(\mathbb{1}\{Y \leq y^*\} - \mathbb{F}_1(y^* \mid X)\big) \mid X\Big] \tag{91}$$

$$+ \mathbb{E}\Big[\frac{1 - A}{(1 - \pi(X))^2} \big(\mathbb{1}\{Y \leq y^* - \delta\} - \mathbb{F}_0(y^* - \delta \mid X)\big) \mid X\Big] (\hat{\pi}(X) - \pi(X)) \big(g(\delta, X) - g_*(\delta, X)\big) \, d\delta\Big] \tag{92}$$

$$= 2\gamma \, \mathbb{E}_X\Big[\int_\Delta I(X; \eta) \Big[\frac{\mathbb{P}(A = 1 \mid X)}{(\pi(X))^2} \big(\mathbb{E}(\mathbb{1}\{Y \leq y^*\} \mid X, A = 1) - \mathbb{F}_1(y^* \mid X)\big) \tag{93}$$

$$+ \frac{\mathbb{P}(A = 0 \mid X)}{(1 - \pi(X))^2} \big(\mathbb{E}(\mathbb{1}\{Y \leq y^* - \delta\} \mid X, A = 0) - \mathbb{F}_0(y^* - \delta \mid X)\big)\Big] (\hat{\pi}(X) - \pi(X)) \big(g(\delta, X) - g_*(\delta, X)\big) \, d\delta\Big] \tag{94}$$

$$= 2\gamma \, \mathbb{E}_X\Big[\int_\Delta I(X; \eta) \Big[\frac{1}{\pi(X)} 0 + \frac{1}{1 - \pi(X)} 0\Big] (\hat{\pi}(X) - \pi(X)) \big(g(\delta, X) - g_*(\delta, X)\big) \, d\delta\Big] = 0, \tag{95}$$

where $I(X; \eta) = \mathbb{1}\{(\mathbb{F}_1 \overline{*} \mathbb{F}_0)_{\mathcal{Y}}(\delta \mid X) > 0\}$ or $\mathbb{1}\{(\mathbb{F}_1 \underline{*} \mathbb{F}_0)_{\mathcal{Y}}(\delta \mid X) < 0\}$; and $y^*$ is some value from $y_{\mathcal{Y}}^{\overline{*}}(\delta \mid X)$.

The derivatives wrt. the conditional CDF $\mathbb{F}_1$ is

$$D_{\mathbb{F}_1} D_g \overline{\mathcal{L}}_{\mathrm{AU,CRPS}}(g_*, \eta)[g - g_*, \hat{\mathbb{F}}_1 - \mathbb{F}_1] \tag{96}$$

$$= \frac{\mathrm{d}}{\mathrm{d}t} - 2\mathbb{E}\Big[ \int_\Delta \Big( \overline{\mathbf{F}}_{\mathrm{AU}}(\delta, Z; \eta = (\pi, \mathbb{F}_0, \mathbb{F}_1 + t(\hat{\mathbb{F}}_1 - \mathbb{F}_1)), \gamma) - g_*(\delta, X) \Big) \, \big( g(\delta, X) - g_*(\delta, X) \big) \, \mathrm{d}\delta \Big] \Big|_{t=0} \tag{97}$$

$$= \frac{\mathrm{d}}{\mathrm{d}t} - 2\mathbb{E}\Big[ \int_\Delta \Big( \overline{\mathbf{F}}_{\mathrm{PI}}(\delta, X; \eta = (\mathbb{F}_0, \mathbb{F}_1 + t(\hat{\mathbb{F}}_1 - \mathbb{F}_1))) \tag{98}$$

$$+ \gamma \overline{C}(\delta, Z; \eta = (\pi, \mathbb{F}_0, \mathbb{F}_1 + t(\hat{\mathbb{F}}_1 - \mathbb{F}_1))) \Big) \big( g(\delta, X) - g_*(\delta, X) \big) \, \mathrm{d}\delta \Big] \Big|_{t=0}$$

$$= -2\mathbb{E}\Big[ \int_\Delta I(X; \eta = (\mathbb{F}_0, \mathbb{F}_1 + 0\,(\hat{\mathbb{F}}_1 - \mathbb{F}_1))) \frac{\mathrm{d}}{\mathrm{d}t}\Big[ \mathbb{F}_1\big(\tilde{y}^*(t) \mid X\big)$$

$$+ t\big(\hat{\mathbb{F}}_1\big(\tilde{y}^*(t) \mid X\big) - \mathbb{F}_1\big(\tilde{y}^*(t) \mid X\big)\big) - \mathbb{F}_0\big(\tilde{y}^*(t) - \delta \mid X\big) \tag{99}$$

$$+ \gamma\Big( \frac{A}{\pi(X)}\Big( \mathbb{1}\{Y \le \tilde{y}^*(t)\} - \mathbb{F}_1\big(\tilde{y}^*(t) \mid X\big) - t\big(\hat{\mathbb{F}}_1\big(\tilde{y}^*(t) \mid X\big) - \mathbb{F}_1\big(\tilde{y}^*(t) \mid X\big)\big)\Big)$$

$$- \frac{1-A}{1-\pi(X)}\Big( \mathbb{1}\{Y \le \tilde{y}^*(t) - \delta\} - \mathbb{F}_0\big(\tilde{y}^*(t) - \delta \mid X\big)\Big)\Big)\Big]\Big|_{t=0} \big( g(\delta, X) - g_*(\delta, X) \big) \, \mathrm{d}\delta \Big]$$

$$= -2\mathbb{E}\Big[ \int_\Delta I(X; \eta)\Big[ \underbrace{\frac{\mathrm{d}}{\mathrm{d}t}\Big[ \mathbb{F}_1\big(\tilde{y}^*(t) \mid X\big) - \mathbb{F}_0\big(\tilde{y}^*(t) - \delta \mid X\big)\Big]\Big|_{t=0}}_{(*) = 0} + 0\, \frac{\mathrm{d}}{\mathrm{d}t}\Big[ \hat{\mathbb{F}}_1\big(\tilde{y}^*(t) \mid X\big) - \mathbb{F}_1\big(\tilde{y}^*(t) \mid X\big)\Big]\Big|_{t=0}$$

$$+ \hat{\mathbb{F}}_1\big(\tilde{y}^*(0) \mid X\big) - \mathbb{F}_1\big(\tilde{y}^*(0) \mid X\big) \tag{100}$$

$$+ \gamma\Big( \frac{A}{\pi(X)}\Big( \frac{\mathrm{d}}{\mathrm{d}t}\Big[ \mathbb{1}\{Y \le \tilde{y}^*(t)\} - \mathbb{F}_1\big(\tilde{y}^*(t) \mid X\big)\Big]\Big|_{t=0} - 0\, \frac{\mathrm{d}}{\mathrm{d}t}\Big[ \hat{\mathbb{F}}_1\big(\tilde{y}^*(t) \mid X\big) - \mathbb{F}_1\big(\tilde{y}^*(t) \mid X\big)\Big]\Big|_{t=0}$$

$$- \big( \hat{\mathbb{F}}_1\big(\tilde{y}^*(t) \mid X\big) - \mathbb{F}_1\big(\tilde{y}^*(t) \mid X\big)\big)\Big)$$

$$- \frac{1-A}{1-\pi(X)} \frac{\mathrm{d}}{\mathrm{d}t}\Big[ \mathbb{1}\{Y \le \tilde{y}^*(t) - \delta\} - \mathbb{F}_0\big(\tilde{y}^*(t) - \delta \mid X\big)\Big]\Big|_{t=0}\Big)\Big] \big( g(\delta, X) - g_*(\delta, X) \big) \, \mathrm{d}\delta \Big]$$

$$= -2\mathbb{E}_X\Big[ \int_\Delta I(X; \eta)\Big[ \hat{\mathbb{F}}_1\big(\tilde{y}^*(0) \mid X\big) - \mathbb{F}_1\big(\tilde{y}^*(0) \mid X\big)$$

$$+ \gamma\Big( \frac{\pi(X)}{\pi(X)}\Big( \mathbb{E}\big[ -\delta\{Y - \tilde{y}^*(0)\} \mid X, A = 1\big] \frac{\mathrm{d}}{\mathrm{d}t}\big[ \tilde{y}^*(t)\big]\Big|_{t=0}$$

$$- \mathbb{E}\big[ -\delta\{Y - \tilde{y}^*(0)\} \mid X, A = 1\big] \frac{\mathrm{d}}{\mathrm{d}t}\big[ \tilde{y}^*(t)\big]\Big|_{t=0} - \big( \hat{\mathbb{F}}_1\big(\tilde{y}^*(0) \mid X\big) - \mathbb{F}_1\big(\tilde{y}^*(0) \mid X\big)\big)\Big)$$

$$- \frac{1-\pi(X)}{1-\pi(X)}\Big( \mathbb{E}\big[ -\delta\{Y - \tilde{y}^*(0) + \delta\} \mid X, A = 0\big] \frac{\mathrm{d}}{\mathrm{d}t}\big[ \tilde{y}^*(t)\big]\Big|_{t=0} \tag{101}$$

$$- \mathbb{E}\big[ -\delta\{Y - \tilde{y}^*(0) + \delta\} \mid X, A = 0\big] \frac{\mathrm{d}}{\mathrm{d}t}\big[ \tilde{y}^*(t)\big]\Big|_{t=0}\Big)\Big)\Big] \big( g(\delta, X) - g_*(\delta, X) \big) \, \mathrm{d}\delta \Big]$$

$$= -2\mathbb{E}_X\Big[ \int_\Delta I(X; \eta)(1 - \gamma) \big( \hat{\mathbb{F}}_1\big(\tilde{y}^*(0) \mid X\big) - \mathbb{F}_1\big(\tilde{y}^*(0) \mid X\big)\big) \big( g(\delta, X) - g_*(\delta, X) \big) \, \mathrm{d}\delta \Big] \underset{\gamma=1}{=} 0, \tag{102}$$

where $I(X; \eta) = \mathbb{1}\big\{(\mathbb{F}_1 \mp \mathbb{F}_0)_{\mathcal{Y}}(\delta \mid X) > 0\big\}$ or $\mathbb{1}\big\{(\mathbb{F}_1 \pm \mathbb{F}_0)_{\mathcal{Y}}(\delta \mid X) < 0\big\}$; $\tilde{y}^*(t)$ is some value from $\tilde{y}^{\overline{*}}_{\mathcal{Y}}(\delta \mid X)$; $\tilde{y}^{\overline{*}}_{\mathcal{Y}}(\cdot \mid X)$ are the argmax/argmin sets of the convolutions $(\mathbb{F}_1 + t(\hat{\mathbb{F}}_1 - \mathbb{F}_1) \mp \mathbb{F}_0)_{\mathcal{Y}}(\cdot \mid X)$; and $(*) = 0$ follows from the the same considerations as in Eq. (51). Analogously, the pathwise derivative wrt. $\mathbb{F}_0$ can be shown to be equal to zero. We refer to the appendices of [96] for more details.

The Neyman-orthogonality of the $W_2^2$ population risk can be proved in a similar fashion. First, the pathwise derivative wrt. $g^{-1}$ has a similar form to Eq. (85), namely

$$D_g \overline{\mathcal{L}}_{\mathrm{AU}, W_2^2}(g_*^{-1}, \eta)[g^{-1} - g_*^{-1}] = -2\mathbb{E}\Big[ \int_0^1 \big( \overline{\mathbf{F}}_{\mathrm{AU}}^{-1}(\alpha, Z; \hat{\eta}, \gamma) - g_*^{-1}(\alpha, X) \big) \big( g^{-1}(\alpha, X) - g_*^{-1}(\alpha, X) \big) \, \mathrm{d}\alpha \Big]. \tag{103}$$

Furthermore, similarly to the CRPS target risk, the cross-derivative wrt. to the propensity score is

$$D_\pi D_g \mathcal{L}_{\text{AU}, W_2^2}(g_*^{-1}, \eta)[g^{-1} - g_*^{-1}, \hat{\pi} - \pi] \tag{104}$$

$$= 2\gamma \mathbb{E}_X \left[ \int_0^1 \left[ \frac{\mathbb{P}(A = 1 \mid X)}{(\pi(X))^2} \left( \frac{\mathbb{E}(\mathbb{1}\{Y \le \mathbb{F}_1^{-1}(u^* \mid X)\} \mid X, A = 1) - u^*}{\mathbb{P}(Y = \mathbb{F}_1^{-1}(u^* \mid X) \mid X, A = 1)} \right) \right. \right. \tag{105}$$

$$\left. \left. + \frac{\mathbb{P}(A = 0 \mid X)}{(1 - \pi(X))^2} \left( \frac{\mathbb{E}(\mathbb{1}\{Y \le \mathbb{F}_0^{-1}(u^* - \alpha + 0 \mid X)\} \mid X, A = 0) - (u^* - \alpha + 0)}{\mathbb{P}(Y = \mathbb{F}_0^{-1}(u^* - \alpha + 0 \mid X) \mid X, A = 0)} \right) \right] (\hat{\pi}(X) - \pi(X)) \left( g^{-1}(\alpha, X) - g_*^{-1}(\alpha, X) \right) d\alpha \right]$$

$$= 2\gamma \mathbb{E}_X \left[ \int_0^1 \left[ \frac{1}{\pi(X)} \left( \frac{\mathbb{F}_1(\mathbb{F}_1^{-1}(u^* \mid X) \mid X) - u^*}{\mathbb{P}(Y = \mathbb{F}_1^{-1}(u^* \mid X) \mid X, A = 1)} \right) \right. \right. \tag{106}$$

$$\left. \left. + \frac{1}{1 - \pi(X)} \left( \frac{\mathbb{F}_0(\mathbb{F}_0^{-1}(u^* - \alpha + 0 \mid X) \mid X) - (u^* - \alpha + 0)}{\mathbb{P}(Y = \mathbb{F}_0^{-1}(u^* - \alpha + 0 \mid X) \mid X, A = 0)} \right) \right] (\hat{\pi}(X) - \pi(X)) \left( g^{-1}(\alpha, X) - g_*^{-1}(\alpha, X) \right) d\alpha \right]$$

$$= 2\gamma \mathbb{E}_X \left[ \int_0^1 \left[ \frac{1}{\pi(X)} 0 + \frac{1}{1 - \pi(X)} 0 \right] (\hat{\pi}(X) - \pi(X)) \left( g^{-1}(\alpha, X) - g_*^{-1}(\alpha, X) \right) d\alpha \right] = 0, \tag{107}$$

where $u^*$ is some value from $u^*_{[\alpha, 1]}(\alpha \mid X)$. The cross-derivative for the upper bound follows similarly.

Finally, to show that a cross-derivative wrt. $\mathbb{F}_1^{-1}$, we make use of the following property of the derivative of the quantiles (inverse function rule):

$$\frac{d}{d\alpha} \left[ \mathbb{F}^{-1}(\alpha) + t(\hat{\mathbb{F}}^{-1}(\alpha) - \mathbb{F}^{-1}(\alpha)) \right] = \frac{1}{\mathbb{P}(\tilde{Y} = \mathbb{F}^{-1}(\alpha) + t(\hat{\mathbb{F}}^{-1}(\alpha) - \mathbb{F}^{-1}(\alpha)); t)}, \tag{108}$$

where $\mathbb{P}(\tilde{Y} = \cdot; t)$ is the density function for a distribution with quantiles $\mathbb{F}^{-1}(\alpha) + t(\hat{\mathbb{F}}^{-1}(\alpha) - \mathbb{F}^{-1}(\alpha))$. Then, the pathwise derivative is

$$\frac{d}{dt} \left[ \frac{1}{\mathbb{P}(\tilde{Y} = \mathbb{F}^{-1}(\alpha) + t(\hat{\mathbb{F}}^{-1}(\alpha) - \mathbb{F}^{-1}(\alpha)); t)} \right] \Bigg|_{t=0} = \frac{d}{dt} \left[ \frac{d}{d\alpha} \left[ \mathbb{F}^{-1}(\alpha) \right] \right] \Bigg|_{t=0} + \frac{d}{d\alpha} \left[ \hat{\mathbb{F}}^{-1}(\alpha) - \mathbb{F}^{-1}(\alpha) \right]. \tag{109}$$

Therefore, the cross-derivative is

$$D_{\mathbb{F}_1^{-1}} D_g \mathcal{L}_{\text{AU}, W_2^2}(g_*^{-1}, \eta)[g^{-1} - g_*^{-1}, \hat{\mathbb{F}}_1^{-1} - \mathbb{F}_1^{-1}] \tag{110}$$

$$= -2\mathbb{E} \left[ \int_0^1 \frac{d}{dt} \left[ \mathbb{F}_1^{-1}(\tilde{u}^*(t) \mid X) + t(\hat{\mathbb{F}}_1^{-1}(\tilde{u}^*(t) \mid X) - \mathbb{F}_1^{-1}(\tilde{u}^*(t) \mid X)) - \mathbb{F}_0^{-1}(\tilde{u}^*(t) - \alpha \mid X) \right. \right.$$

$$\left. + \gamma \left( \frac{A}{\pi(X)} \left( \frac{\mathbb{1}\{Y \le \mathbb{F}_1^{-1}(\tilde{u}^*(t) \mid X) + t(\hat{\mathbb{F}}_1^{-1}(\tilde{u}^*(t) \mid X) - \mathbb{F}_1^{-1}(\tilde{u}^*(t) \mid X))\} - \tilde{u}^*(t)}{\mathbb{P}(\tilde{Y} = \mathbb{F}_1^{-1}(\tilde{u}^*(t) \mid X) + t(\hat{\mathbb{F}}_1^{-1}(\tilde{u}^*(t) \mid X) - \mathbb{F}_1^{-1}(\tilde{u}^*(t) \mid X)) \mid X, A = 1; t)} \right) \right. \right. \tag{111}$$

$$\left. \left. - \frac{1 - A}{1 - \pi(X)} \left( \frac{\mathbb{1}\{Y \le \mathbb{F}_0^{-1}(\tilde{u}^*(t) - \alpha + 0 \mid X)\} - (\tilde{u}^*(t) - \alpha + 0)}{\mathbb{P}(Y = \mathbb{F}_0^{-1}(\tilde{u}^*(t) - \alpha + 0 \mid X) \mid X, A = 0)} \right) \right) \right] \Bigg|_{t=0} \left( g^{-1}(\alpha, X) - g_*^{-1}(\alpha, X) \right) d\alpha \right]$$

$$= -2\mathbb{E} \left[ \int_0^1 \underbrace{\frac{d}{dt} \left[ \mathbb{F}_1^{-1}(\tilde{u}^*(t) \mid X) - \mathbb{F}_0^{-1}(\tilde{u}^*(t) - \alpha \mid X) \right] \Bigg|_{t=0}}_{(*)=0} + 0 \frac{d}{dt} \left[ \hat{\mathbb{F}}_1^{-1}(\tilde{u}^*(t) \mid X) - \mathbb{F}_1^{-1}(\tilde{u}^*(t) \mid X) \right] \Bigg|_{t=0} \right.$$

$$+ \hat{\mathbb{F}}_1^{-1}(\tilde{u}^*(0) \mid X) - \mathbb{F}_1^{-1}(\tilde{u}^*(0) \mid X)$$

$$+ \gamma \left( \frac{A}{\pi(X)} \left( \left( \frac{d}{dt} \left[ \hat{\mathbb{F}}_1^{-1}(\tilde{u}^*(t) \mid X) \right] \Bigg|_{t=0} + \frac{1}{\hat{\mathbb{P}}\left(Y = \hat{\mathbb{F}}_1^{-1}(\tilde{u}^*(t) \mid X) \mid X, A = 1\right)} \right. \right. \right. \tag{112}$$

$$\left. \left. - \frac{1}{\mathbb{P}\left(Y = \mathbb{F}_1^{-1}(\tilde{u}^*(t) \mid X) \mid X, A = 1\right)} \right) \left( \mathbb{1}\{Y \le \mathbb{F}_1^{-1}(\tilde{u}^*(t) \mid X)\} - \tilde{u}^*(t) \right) \right.$$

$$\left. + \frac{\frac{d}{dt} \left[ \mathbb{1}\{Y \le \mathbb{F}_1^{-1}(\tilde{u}^*(t) \mid X) + t(\hat{\mathbb{F}}_1^{-1}(\tilde{u}^*(t) \mid X) - \mathbb{F}_1^{-1}(\tilde{u}^*(t) \mid X))\} \right] \Big|_{t=0} - \frac{d}{dt} \left[ \tilde{u}^*(t) \right] \Big|_{t=0}}{\mathbb{P}\left(Y = \mathbb{F}_1^{-1}(\tilde{u}^*(t) \mid X) \mid X, A = 1\right)} \right)$$

$$- \frac{1 - A}{1 - \pi(X)} \left( \frac{d}{dt} \left[ \frac{1}{\mathbb{P}(Y = \mathbb{F}_0^{-1}(\tilde{u}^*(t) - \alpha + 0 \mid X) \mid X, A = 0)} \right] \Bigg|_{t=0} \left( \mathbb{1}\{Y \le \mathbb{F}_0^{-1}(\tilde{u}^*(t) - \alpha + 0 \mid X)\} \right. \right.$$

$$\left. - (\tilde{u}^*(t) - \alpha + 0) \right)$$

$$\left. \left. + \frac{\frac{d}{dt} \left[ \mathbb{1}\{Y \le \mathbb{F}_0^{-1}(\tilde{u}^*(t) - \alpha + 0 \mid X)\} \right] \Big|_{t=0} - \frac{d}{dt} \left[ (\tilde{u}^*(t) - \alpha + 0) \right] \Big|_{t=0}}{\mathbb{P}(Y = \mathbb{F}_0^{-1}(\tilde{u}^*(t) - \alpha + 0 \mid X) \mid X, A = 0)} \right) \right) \left( g^{-1}(\alpha, X) - g_*^{-1}(\alpha, X) \right) d\alpha \right]$$

$$
\begin{aligned}
= -2\mathbb{E}_X\Bigg[ &\int_0^1 \Bigg[ \hat{\mathbb{F}}_1^{-1}(\tilde{u}^*(0) \mid X) - \mathbb{F}_1^{-1}(\tilde{u}^*(0) \mid X) \\
&+ \gamma\Bigg( \frac{\pi(X)}{\pi(X)} \Bigg( \Big( \frac{\mathrm{d}}{\mathrm{d}t}\Big[\hat{\mathbb{F}}_1^{-1}(\tilde{u}^*(t) \mid X)\Big]\Big|_{t=0} + \frac{1}{\hat{\mathbb{P}}\Big(Y = \hat{\mathbb{F}}_1^{-1}(\tilde{u}^*(0) \mid X) \mid X, A=1\Big)} \frac{1}{\mathbb{P}\Big(Y = \mathbb{F}_1^{-1}(\tilde{u}^*(0) \mid X) \mid X, A=1\Big)} \Bigg) 0 \\
&+ \frac{\mathbb{E}\Big[ -\delta\{Y \le \mathbb{F}_1^{-1}(\tilde{u}^*(0) \mid X)\} \mid X, A=1\Big]\Big(\frac{\mathrm{d}}{\mathrm{d}t}\Big[\mathbb{F}_1^{-1}(\tilde{u}^*(t) \mid X)\Big]\Big|_{t=0} + \big(\hat{\mathbb{F}}_1^{-1}(\tilde{u}^*(0) \mid X) - \mathbb{F}_1^{-1}(\tilde{u}^*(0) \mid X)\big)\Big) - \frac{\mathrm{d}}{\mathrm{d}t}\Big[\tilde{u}^*(t)\Big]\Big|_{t=0}}{\mathbb{P}\Big(Y = \mathbb{F}_1^{-1}(\tilde{u}^*(0) \mid X) \mid X, A=1\Big)} \\
&- \frac{1-\pi(X)}{1-\pi(X)} \Bigg( \frac{\mathrm{d}}{\mathrm{d}t}\Big[ \frac{1}{\mathbb{P}(Y = \mathbb{F}_0^{-1}(\tilde{u}^*(t) - \alpha + 0 \mid X) \mid X, A=0)}\Big]\Big|_{t=0} 0 \quad (113)\\
&+ \frac{\mathbb{E}\Big[ -\delta\{Y - \mathbb{F}_0^{-1}(\tilde{u}^*(0) - \alpha + 0 \mid X)\} \mid X, A=0\Big]\frac{\mathrm{d}}{\mathrm{d}t}\Big[\mathbb{F}_0^{-1}(\tilde{u}^*(t) - \alpha + 0 \mid X)\Big]\Big|_{t=0} - \frac{\mathrm{d}}{\mathrm{d}t}\Big[\tilde{u}^*(t)\Big]\Big|_{t=0}}{\mathbb{P}(Y = \mathbb{F}_0^{-1}(\tilde{u}^*(0) - \alpha + 0 \mid X) \mid X, A=0)} \Bigg) \Bigg) \Bigg] \big(g^{-1}(\alpha, X) - g_*^{-1}(\alpha, X)\big) \, \mathrm{d}\alpha \Bigg]
\end{aligned}
$$

$$
\begin{aligned}
= -2\mathbb{E}_X\Bigg[ &\int_0^1 \Bigg[ \hat{\mathbb{F}}_1^{-1}(\tilde{u}^*(0) \mid X) - \mathbb{F}_1^{-1}(\tilde{u}^*(0) \mid X) \\
&+ \gamma\Bigg( \frac{\frac{\mathrm{d}}{\mathrm{d}t}\Big[\tilde{u}^*(t)\Big]\Big|_{t=0} - \mathbb{P}\Big(Y = \mathbb{F}_1^{-1}(\tilde{u}^*(0) \mid X) \mid X, A=1\Big)\big(\hat{\mathbb{F}}_1^{-1}(\tilde{u}^*(0) \mid X) - \mathbb{F}_1^{-1}(\tilde{u}^*(0) \mid X)\big) - \frac{\mathrm{d}}{\mathrm{d}t}\Big[\tilde{u}^*(t)\Big]\Big|_{t=0}}{\mathbb{P}\Big(Y = \mathbb{F}_1^{-1}(\tilde{u}^*(0) \mid X) \mid X, A=1\Big)} \Bigg) \\
& \quad\quad (114)
\end{aligned}
$$

$$
\begin{aligned}
&- \frac{\frac{\mathrm{d}}{\mathrm{d}t}\Big[\tilde{u}^*(t)\Big]\Big|_{t=0} - \frac{\mathrm{d}}{\mathrm{d}t}\Big[\tilde{u}^*(t)\Big]\Big|_{t=0}}{\mathbb{P}(Y = \mathbb{F}_0^{-1}(\tilde{u}^*(0) - \alpha + 0 \mid X) \mid X, A=0)} \Bigg) \Bigg] \big(g^{-1}(\alpha, X) - g_*^{-1}(\alpha, X)\big)\,\mathrm{d}\alpha \Bigg]
\end{aligned}
$$

$$
= -2\mathbb{E}_X\Bigg[ \int_0^1 (1-\gamma)\big(\hat{\mathbb{F}}_1^{-1}(\tilde{u}^*(0) \mid X) - \mathbb{F}_1^{-1}(\tilde{u}^*(0) \mid X)\big)\big(g^{-1}(\alpha, X) - g_*^{-1}(\alpha, X)\big)\,\mathrm{d}\alpha \Bigg] \underset{\gamma=1}{=} 0, \qquad (115)
$$

where $\tilde{u}^*(t)$ is some value from $\tilde{u}^*_{[\alpha,1]}(\alpha \mid X)$; and $\tilde{u}^*_{[\alpha,1]}(\alpha \mid X)$ is the argmin set of the convolution $(\mathbb{F}_1^{-1} + t(\hat{\mathbb{F}}_1^{-1} - \mathbb{F}_1^{-1}) \underline{*} \mathbb{F}_0^{-1})_{[\alpha,1]}(\cdot - 0 \mid X)$. The cross-derivatives for the upper bound wrt. $\mathbb{F}_1^{-1}$ and for both upper and lower bounds wrt. $\mathbb{F}_0^{-1}$ follow similarly.

*2. Quasi-oracle efficiency.* The result is a direct application of Theorem 1 in [35]: It is easy to see that the Assumptions 1–4 from [35] hold for the population versions of the empirical risks of our *AU-learner* (i.e., CRPS and $W_2^2$). In the following, we provide the derivation of the quasi-oracle efficiency property for the population version of CRPS loss (the derivation is similar to one in [96, 125]).

We first apply a functional Taylor expansion of $\overline{\mathcal{L}}_{\mathrm{AU,CRPS}}(\hat{\overline{g}}, \hat{\eta})$ at the $g_*$:

$$
\overline{\mathcal{L}}_{\mathrm{AU,CRPS}}(\hat{\overline{g}}, \hat{\eta}) = \mathbb{E}\Big[ \int_\Delta \big( \overline{\mathbf{F}}_{\mathrm{AU}}(\delta, Z; \hat{\eta}, \gamma) - \hat{\overline{g}}(\delta, X) + g_*(\delta, X) - g_*(\delta, X) \big)^2 \, \mathrm{d}\delta \Big] \qquad (116)
$$

$$
= \overline{\mathcal{L}}_{\mathrm{AU,CRPS}}(g_*, \hat{\eta}) - 2\mathbb{E}\Big[ \int_\Delta \big( \overline{\mathbf{F}}_{\mathrm{AU}}(\delta, Z; \hat{\eta}, \gamma) - g_*(\delta, X) \big) \big( \hat{\overline{g}}(\delta, X) - g_*(\delta, X) \big) \, \mathrm{d}\delta \Big] \qquad (117)
$$

$$
+ \mathbb{E}\Big[ \int_\Delta \big( \hat{\overline{g}}(\delta, X) - g_*(\delta, X) \big)^2 \, \mathrm{d}\delta \Big].
$$

Therefore, the following holds:

$$
\big\| \hat{\overline{g}} - g_* \big\|_{2,\mathrm{CRPS}}^2 = \mathbb{E}\Big[ \int_\Delta \big( \hat{\overline{g}}(\delta, X) - g_*(\delta, X) \big)^2 \, \mathrm{d}\delta \Big] = \overline{\mathcal{L}}_{\mathrm{AU,CRPS}}(\hat{\overline{g}}, \hat{\eta}) - \overline{\mathcal{L}}_{\mathrm{AU,CRPS}}(g_*, \hat{\eta}) \qquad (118)
$$

$$
+ 2\mathbb{E}\Big[ \int_\Delta \big( \overline{\mathbf{F}}_{\mathrm{AU}}(\delta, Z; \hat{\eta}, \gamma) - g_*(\delta, X) \big) \big( \hat{\overline{g}}(\delta, X) - g_*(\delta, X) \big) \, \mathrm{d}\delta \Big].
$$

The latter term then also allows for the distributional Taylor expansion around $\eta$:

$$
2\mathbb{E}\Big[ \int_\Delta \big( \overline{\mathbf{F}}_{\mathrm{AU}}(\delta, Z; \hat{\eta}, \gamma) - \overline{\mathbf{F}}(\delta \mid X) + \overline{\mathbf{F}}(\delta \mid X) - g_*(\delta, X) \big) \big( \hat{\overline{g}}(\delta, X) - g_*(\delta, X) \big) \, \mathrm{d}\delta \Big] \qquad (119)
$$

$$
= 2\mathbb{E}\Big[ \int_\Delta \big( \overline{\mathbf{F}}_{\mathrm{AU}}(\delta, Z; \hat{\eta}, \gamma) - \overline{\mathbf{F}}(\delta \mid X) \big) \big( \hat{\overline{g}}(\delta, X) - g_*(\delta, X) \big) \, \mathrm{d}\delta \Big] \qquad (120)
$$

$$
+ 2\mathbb{E}\Big[ \int_\Delta \big( \overline{\mathbf{F}}(\delta \mid X) - g_*(\delta, X) \big) \big( \hat{\overline{g}}(\delta, X) - g_*(\delta, X) \big) \, \mathrm{d}\delta \Big],
$$

where the first term is known as a second-order remainder term, $R_2(\eta, \hat{\eta})$, and the second term equals to $-D_g\overline{\mathcal{L}}_{\text{AU,CRPS}}(g_*, \eta)[\widehat{\overline{g}} - g_*]$. $R_2(\eta, \hat{\eta})$ can be further expressed as

$$R_2(\eta, \hat{\eta}) = 2\mathbb{E}\left[\int_\Delta \left(\overline{\mathbf{F}}_{\text{PI}}(\delta \mid X; \hat{\eta}) + \gamma\overline{C}(\delta, Z; \hat{\eta}) - \overline{\mathbf{F}}(\delta \mid X)\right)\left(\widehat{\overline{g}}(\delta, X) - g_*(\delta, X)\right)\mathrm{d}\delta\right] \tag{121}$$

$$= 2\mathbb{E}\left[\int_\Delta I(X; \hat{\eta})\left(\gamma\left[\frac{A}{\hat{\pi}(X)}\left(\mathbb{1}\{Y \le \hat{y}^*\} - \hat{\mathbb{F}}_1(\hat{y}^* \mid X)\right) - \frac{1-A}{1-\hat{\pi}(X)}\left(\mathbb{1}\{Y \le \hat{y}^* - \delta\} - \hat{\mathbb{F}}_0(\hat{y}^* - \delta \mid X)\right)\right]\right.\right. \tag{122}$$

$$\left.\left. + \hat{\mathbb{F}}_1(\hat{y}^* \mid X) - \hat{\mathbb{F}}_0(\hat{y}^* - \delta \mid X)\right) - I(X; \eta)\left(\left(\mathbb{F}_1(y^* \mid X) - \mathbb{F}_0(y^* - \delta \mid X)\right)\right)\left(\widehat{\overline{g}}(\delta, X) - g_*(\delta, X)\right)\mathrm{d}\delta\right]$$

$$= 2\mathbb{E}\left[\int_\Delta \gamma I(X; \hat{\eta})\left(\left[\frac{\pi(X)}{\hat{\pi}(X)}\left(\mathbb{F}_1(\hat{y}^* \mid X) - \hat{\mathbb{F}}_1(\hat{y}^* \mid X)\right) - \frac{1-\pi(X)}{1-\hat{\pi}(X)}\left(\mathbb{F}_0(\hat{y}^* - \delta \mid X) - \hat{\mathbb{F}}_0(\hat{y}^* - \delta \mid X)\right)\right]\right.\right.$$

$$\left.\left. - \left(\mathbb{F}_1(\hat{y}^* \mid X) - \hat{\mathbb{F}}_1(\hat{y}^* \mid X)\right) + \mathbb{F}_0(\hat{y}^* - \delta \mid X) - \hat{\mathbb{F}}_0(\hat{y}^* - \delta \mid X)\right)\left(\widehat{\overline{g}}(\delta, X) - g_*(\delta, X)\right)\mathrm{d}\delta\right]$$

$$+ 2\mathbb{E}\left[\int_\Delta \left(\left(I(X; \hat{\eta}) - I(X; \eta)\right)\left(\mathbb{F}_1(\hat{y}^* \mid X) - \mathbb{F}_0(\hat{y}^* - \delta \mid X)\right)\right.\right. \tag{123}$$

$$\left.\left. + I(X; \eta)\left(\mathbb{F}_1(\hat{y}^* \mid X) - \mathbb{F}_0(\hat{y}^* - \delta \mid X) - \mathbb{F}_1(y^* \mid X) - \mathbb{F}_0(y^* - \delta \mid X)\right)\right)\left(\widehat{\overline{g}}(\delta, X) - g_*(\delta, X)\right)\mathrm{d}\delta\right]$$

$$= 2\mathbb{E}\left[\int_\Delta I(X; \hat{\eta})\pi(X)\left(\frac{\gamma}{\hat{\pi}(X)} - \frac{1}{\pi(X)}\right)\left(\mathbb{F}_1(\hat{y}^* \mid X) - \hat{\mathbb{F}}_1(\hat{y}^* \mid X)\right)\left(\widehat{\overline{g}}(\delta, X) - g_*(\delta, X)\right)\mathrm{d}\delta\right]$$

$$- 2\mathbb{E}\left[\int_\Delta I(X; \hat{\eta})(1 - \pi(X))\left(\frac{\gamma}{1-\hat{\pi}(X)} - \frac{1}{1-\pi(X)}\right)\left(\mathbb{F}_0(\hat{y}^* - \delta \mid X) - \hat{\mathbb{F}}_0(\hat{y}^* - \delta \mid X)\right)\left(\widehat{\overline{g}}(\delta, X) - g_*(\delta, X)\right)\mathrm{d}\delta\right]$$

$$+ 2\mathbb{E}\left[\int_\Delta \left(I(X; \hat{\eta}) - I(X; \eta)\right)\left(\mathbb{F}_1(\hat{y}^* \mid X) - \mathbb{F}_0(\hat{y}^* - \delta \mid X)\right)\left(\widehat{\overline{g}}(\delta, X) - g_*(\delta, X)\right)\mathrm{d}\delta\right] \tag{124}$$

$$+ \mathbb{E}\left[\int_\Delta I(X; \eta)\frac{1}{2}\frac{\mathrm{d}^2}{\mathrm{d}y^2}\left[\mathbb{F}_1(y \mid X) - \mathbb{F}_0(y - \delta \mid X)\right]\Big|_{y=\tilde{y}^*}(y^* - \hat{y}^*)^2\left(\widehat{\overline{g}}(\delta, X) - g_*(\delta, X)\right)\mathrm{d}\delta\right],$$

where $I(X; \eta) = \mathbb{1}\{(\mathbb{F}_1 \overline{*} \mathbb{F}_0)_\mathcal{Y}(\delta \mid X) > 0\}$ or $\mathbb{1}\{(\mathbb{F}_1 \underline{*} \mathbb{F}_0)_\mathcal{Y}(\delta \mid X) < 0\}$; $y^*$ is some value from $y^{\overline{*}}_{\overline{\mathcal{Y}}}(\delta \mid X)$; $\hat{y}^*$ is some value from $\hat{y}^{\overline{*}}_{\overline{\mathcal{Y}}}(\delta \mid X)$; and $\tilde{y}^*$ is a value between $y^*$ and $\hat{y}^*$.

Considering there is such an $\epsilon > 0$, for which $\epsilon \le \hat{\pi}(x) \le (1 - \epsilon)$, the second-order remainder term can be upper bounded with the Cauchy-Schwarz inequality:

$$|R_2(\eta, \hat{\eta})| \le \frac{2}{\epsilon}\mathbb{E}\left[|\gamma\pi(X) - \hat{\pi}(X)|\int_\Delta C_{I,\hat{\eta}}(\delta, X)\left|\mathbb{F}_1(\hat{y}^* \mid X) - \hat{\mathbb{F}}_1(\hat{y}^* \mid X)\right|\left|\widehat{\overline{g}}(\delta, X) - g_*(\delta, X)\right|\mathrm{d}\delta\right] \tag{125}$$

$$+ \frac{2}{\epsilon}\mathbb{E}\left[|\gamma(\pi(X) - 1) - (\hat{\pi}(X) - 1)|\int_\Delta C_{I,\hat{\eta}}(\delta, X)\left|\mathbb{F}_0(\hat{y}^* - \delta \mid X) - \hat{\mathbb{F}}_0(\hat{y}^* - \delta \mid X)\right|\left|\widehat{\overline{g}}(\delta, X) - g_*(\delta, X)\right|\mathrm{d}\delta\right]$$

$$+ 2\mathbb{E}\left[\int_\Delta C_{\mathbb{F},\hat{y}^*}(\delta, X)|I(X, \hat{\eta}) - I(X, \eta)|\left|\widehat{\overline{g}}(\delta, X) - g_*(\delta, X)\right|\mathrm{d}\delta\right]$$

$$+ \mathbb{E}\left[\int_\Delta C_{y^*}(\delta, X)(y^* - \hat{y}^*)^2\left|\widehat{\overline{g}}(\delta, X) - g_*(\delta, X)\right|\mathrm{d}\delta\right]$$

$$\le \frac{2}{\epsilon}\|\gamma\pi - \hat{\pi}\|_{L_2}\|C_{I,\hat{\eta}}\|_{2,\text{CRPS}}\left\|\mathbb{F}_1 \circ \hat{y}^* - \hat{\mathbb{F}}_1 \circ \hat{y}^*\right\|_{2,\text{CRPS}}\left\|\widehat{\overline{g}} - g_*\right\|_{2,\text{CRPS}} \tag{126}$$

$$+ \frac{2}{\epsilon}\left(\|\gamma\pi - \hat{\pi}\|_{L_2} + |\gamma - 1|\right)\|C_{I,\hat{\eta}}\|_{2,\text{CRPS}}\left\|\mathbb{F}_0 \circ (\hat{y}^* - \cdot) - \hat{\mathbb{F}}_0 \circ (\hat{y}^* - \cdot)\right\|_{2,\text{CRPS}}\left\|\widehat{\overline{g}} - g_*\right\|_{2,\text{CRPS}}$$

$$+ 2\|C_{\mathbb{F},\hat{y}^*}\|_{2,\text{CRPS}}\|I(\cdot, \hat{\eta}) - I(\cdot, \eta)\|_{2,\text{CRPS}}\left\|\widehat{\overline{g}} - g_*\right\|_{2,\text{CRPS}}$$

$$+ \|C_{y^*}\|_{L_2}\|y^* - \hat{y}^*\|^2_{4,\text{CRPS}}\left\|\widehat{\overline{g}} - g_*\right\|_{2,\text{CRPS}},$$

where $C_{I,\hat{\eta}}(\delta, x) = |I(x, \hat{\eta})|$; $C_{y^*}(\delta, x) = I(x; \eta)\frac{1}{2}\left|\frac{\mathrm{d}^2}{\mathrm{d}y^2}\left[\mathbb{F}_1(y \mid x) - \mathbb{F}_0(y - \delta \mid x)\right]\Big|_{y=\tilde{y}^*}\right|$; and $C_{\mathbb{F},\hat{y}^*}(\delta, x) = \mathbb{F}_1(\hat{y}^* \mid x) - \mathbb{F}_0(\hat{y}^* - \delta \mid x)$.

By combining the previous expressions, the following holds:

$$\left\|\widehat{\overline{g}} - g_*\right\|^2_{2,\text{CRPS}} \le \overline{\mathcal{L}}_{\text{AU,CRPS}}(\widehat{\overline{g}}, \hat{\eta}) - \overline{\mathcal{L}}_{\text{AU,CRPS}}(g_*, \hat{\eta}) - D_g\overline{\mathcal{L}}_{\text{AU,CRPS}}(g_*, \eta)[\widehat{\overline{g}} - g_*] \tag{127}$$

$$+ \frac{2}{\epsilon}\left\|C_{I,\hat{\eta}}\right\|_{2,\text{CRPS}}\left\|\gamma\pi - \hat{\pi}\right\|_{L_2}\left\|\mathbb{F}_1 \circ \hat{y}^* - \hat{\mathbb{F}}_1 \circ \hat{y}^*\right\|_{2,\text{CRPS}}\left\|\widehat{\overline{g}} - g_*\right\|_{2,\text{CRPS}}$$

$$+ \frac{2}{\epsilon}\left\|C_{I,\hat{\eta}}\right\|_{2,\text{CRPS}}\left(\left\|\gamma\pi - \hat{\pi}\right\|_{L_2} + |\gamma - 1|\right)\left\|\mathbb{F}_0 \circ (\hat{y}^* - \cdot) - \hat{\mathbb{F}}_0 \circ (\hat{y}^* - \cdot)\right\|_{2,\text{CRPS}}\left\|\widehat{\overline{g}} - g_*\right\|_{2,\text{CRPS}}$$

$$+ 2\left\|C_{\mathbb{F},\hat{y}^*}\right\|_{2,\text{CRPS}}\left\|I(\cdot, \hat{\eta}) - I(\cdot, \eta)\right\|_{2,\text{CRPS}}\left\|\widehat{\overline{g}} - g_*\right\|_{2,\text{CRPS}}$$

$$+ \left\|C_{y^*}\right\|_{L_2}\left\|y^* - \hat{y}^*\right\|^2_{4,\text{CRPS}}\left\|\widehat{\overline{g}} - g_*\right\|_{2,\text{CRPS}}.$$

Furthermore, using the AM-GM inequality for the last three terms, for any constants $\delta_1 > 0$, $\delta_2 > 0$, $\delta_3 > 0$, $\delta_4 > 0$, $\delta_1 + \delta_2 + \delta_3 + \delta_4 < 1$, we obtain:

$$\left\|\widehat{\overline{g}} - g_*\right\|^2_{2,\text{CRPS}} \le \frac{1}{1 - \delta_1 - \delta_2 - \delta_3 - \delta_4}\left(\overline{\mathcal{L}}_{\text{AU,CRPS}}(\widehat{\overline{g}}, \hat{\eta}) - \overline{\mathcal{L}}_{\text{AU,CRPS}}(g_*, \hat{\eta}) - D_g\overline{\mathcal{L}}_{\text{AU,CRPS}}(g_*, \eta)[\widehat{\overline{g}} - g_*]\right)$$

$$+ \frac{1}{\epsilon\delta_1}\left\|C_{I,\hat{\eta}}\right\|^2_{2,\text{CRPS}}\left\|\gamma\pi - \hat{\pi}\right\|^2_{L_2}\left\|\mathbb{F}_1 \circ \hat{y}^* - \hat{\mathbb{F}}_1 \circ \hat{y}^*\right\|^2_{2,\text{CRPS}} \tag{128}$$

$$+ \frac{1}{\epsilon\delta_2}\left\|C_{I,\hat{\eta}}\right\|^2_{2,\text{CRPS}}\left(\left\|\gamma\pi - \hat{\pi}\right\|_{L_2} + |\gamma - 1|\right)^2\left\|\mathbb{F}_0 \circ (\hat{y}^* - \cdot) - \hat{\mathbb{F}}_0 \circ (\hat{y}^* - \cdot)\right\|^2_{2,\text{CRPS}}$$

$$+ \frac{1}{\delta_3}\left\|C_{\mathbb{F},\hat{y}^*}\right\|^2_{2,\text{CRPS}}\left\|I(\cdot, \hat{\eta}) - I(\cdot, \eta)\right\|^2_{2,\text{CRPS}}$$

$$+ \frac{1}{2\delta_4}\left\|C_{y^*}\right\|^2_{L_2}\left\|y^* - \hat{y}^*\right\|^4_{4,\text{CRPS}}.$$

We note that, given Assumptions 1 and 2, the term $\|I(\cdot, \hat{\eta}) - I(\cdot, \eta)\|^2_{2,\text{CRPS}}$ can be upper bounded as follows

$$\|I(\cdot, \hat{\eta}) - I(\cdot, \eta)\|^2_{2,\text{CRPS}} = \mathbb{E}\left[\int_\Delta \left(\mathbb{1}\{(\hat{\mathbb{F}}_1 \mp \hat{\mathbb{F}}_0)_{\mathcal{Y}}(\delta \mid X) > 0\} - \mathbb{1}\{(\mathbb{F}_1 \mp \mathbb{F}_0)_{\mathcal{Y}}(\delta \mid X) > 0\}\right)^2 d\delta\right] \tag{129}$$

$$= \mathbb{E}\left[\int_\Delta \left|\mathbb{1}\{(\hat{\mathbb{F}}_1 \mp \hat{\mathbb{F}}_0)_{\mathcal{Y}}(\delta \mid X) > 0\} - \mathbb{1}\{(\mathbb{F}_1 \mp \mathbb{F}_0)_{\mathcal{Y}}(\delta \mid X) > 0\}\right| d\delta\right] \tag{130}$$

$$= \int_\Delta \mathbb{P}\left\{\text{sign}\{(\hat{\mathbb{F}}_1 \mp \hat{\mathbb{F}}_0)_{\mathcal{Y}}(\delta \mid X)\} \ne \text{sign}\{(\mathbb{F}_1 \mp \mathbb{F}_0)_{\mathcal{Y}}(\delta \mid X)\}\right\} d\delta \tag{131}$$

$$\le \int_\Delta \mathbb{P}\left\{\left|(\hat{\mathbb{F}}_1 \mp \hat{\mathbb{F}}_0)_{\mathcal{Y}}(\delta \mid X) - (\mathbb{F}_1 \mp \mathbb{F}_0)_{\mathcal{Y}}(\delta \mid X)\right| \ge \xi\right\} d\delta \tag{132}$$

$$\overset{(*)}{\le} \frac{1}{\xi^4}\int_\Delta \mathbb{E}\left[\left((\hat{\mathbb{F}}_1 \mp \hat{\mathbb{F}}_0)_{\mathcal{Y}}(\delta \mid X) - (\mathbb{F}_1 \mp \mathbb{F}_0)_{\mathcal{Y}}(\delta \mid X)\right)^4\right] d\delta \tag{133}$$

$$= \frac{1}{\xi^4}\left(\left\|\mathbb{F}_1 \circ y^* - \hat{\mathbb{F}}_1 \circ \hat{y}^*\right\|^4_{4,\text{CRPS}} + \left\|\mathbb{F}_0 \circ (y^* - \cdot) - \hat{\mathbb{F}}_0 \circ (\hat{y}^* - \cdot)\right\|^4_{4,\text{CRPS}}\right. \tag{134}$$

$$\left. + 2\left\|\mathbb{F}_1 \circ y^* - \hat{\mathbb{F}}_1 \circ \hat{y}^*\right\|^2_{2,\text{CRPS}}\left\|\mathbb{F}_0 \circ (y^* - \cdot) - \hat{\mathbb{F}}_0 \circ (\hat{y}^* - \cdot)\right\|^2_{2,\text{CRPS}}\right),$$

where $\xi > 0$ and $(*)$ holds due to Markov's inequality.

On the other hand, since the term $D_g \overline{\mathcal{L}}_{\text{AU,CRPS}}(g_*, \eta)[\widehat{\overline{g}} - g_*]$ is always positive (by virtue of the optimization), we finally yield the desired quasi-oracle efficiency:

$$\left\| \widehat{\overline{g}} - g_* \right\|_{2,\text{CRPS}}^2 \leq \frac{1}{1 - \delta_1 - \delta_2 - \delta_3 - \delta_4} \left( \overline{\mathcal{L}}_{\text{AU,CRPS}}(\widehat{\overline{g}}, \hat{\eta}) - \overline{\mathcal{L}}_{\text{AU,CRPS}}(g_*, \hat{\eta}) \right) \tag{135}$$

$$+ \frac{1}{\epsilon \delta_1} \|C_{I,\hat{\eta}}\|_{2,\text{CRPS}}^2 \|\gamma \pi - \hat{\pi}\|_{L_2}^2 \left\| \mathbb{F}_1 \circ \hat{y}^* - \hat{\mathbb{F}}_1 \circ \hat{y}^* \right\|_{2,\text{CRPS}}^2$$

$$+ \frac{1}{\epsilon \delta_2} \|C_{I,\hat{\eta}}\|_{2,\text{CRPS}}^2 , \left( \|\gamma \pi - \hat{\pi}\|_{L_2} + |\gamma - 1| \right)^2 \left\| \mathbb{F}_0 \circ (\hat{y}^* - \cdot) - \hat{\mathbb{F}}_0 \circ (\hat{y}^* - \cdot) \right\|_{2,\text{CRPS}}^2$$

$$+ \frac{1}{\delta_3 \xi^4} \|C_{\mathbb{F},\hat{y}^*}\|_{2,\text{CRPS}}^2 \left( \left\| \mathbb{F}_1 \circ y^* - \hat{\mathbb{F}}_1 \circ \hat{y}^* \right\|_{4,\text{CRPS}}^4 + \left\| \mathbb{F}_0 \circ (y^* - \cdot) - \hat{\mathbb{F}}_0 \circ (\hat{y}^* - \cdot) \right\|_{4,\text{CRPS}}^4 \right.$$

$$\left. + 2 \left\| \mathbb{F}_1 \circ y^* - \hat{\mathbb{F}}_1 \circ \hat{y}^* \right\|_{2,\text{CRPS}}^2 \left\| \mathbb{F}_0 \circ (y^* - \cdot) - \hat{\mathbb{F}}_0 \circ (\hat{y}^* - \cdot) \right\|_{2,\text{CRPS}}^2 \right)$$

$$+ \frac{1}{2\delta_4} \|C_{y^*}\|_{L_2}^2 \|y^* - \hat{y}^*\|_{4,\text{CRPS}}^4$$

$$\underset{\gamma=1}{\leq} \frac{1}{1 - \delta_1 - \delta_2 - \delta_3} \left( \overline{\mathcal{L}}_{\text{AU,CRPS}}(\widehat{\overline{g}}, \hat{\eta}) - \overline{\mathcal{L}}_{\text{AU,CRPS}}(g_*, \hat{\eta}) \right) \tag{136}$$

$$+ \frac{1}{\epsilon \delta_1} \|C_{I,\hat{\eta}}\|_{2,\text{CRPS}}^2 \|\pi - \hat{\pi}\|_{L_2}^2 \left\| \mathbb{F}_1 \circ \hat{y}^* - \hat{\mathbb{F}}_1 \circ \hat{y}^* \right\|_{2,\text{CRPS}}^2$$

$$+ \frac{1}{\epsilon \delta_2} \|C_{I,\hat{\eta}}\|_{2,\text{CRPS}}^2 \|\pi - \hat{\pi}\|_{L_2}^2 \left\| \mathbb{F}_0 \circ (\hat{y}^* - \cdot) - \hat{\mathbb{F}}_0 \circ (\hat{y}^* - \cdot) \right\|_{2,\text{CRPS}}^2$$

$$+ \frac{1}{\delta_3 \xi^4} \|C_{\mathbb{F},\hat{y}^*}\|_{2,\text{CRPS}}^2 \left( \left\| \mathbb{F}_1 \circ y^* - \hat{\mathbb{F}}_1 \circ \hat{y}^* \right\|_{4,\text{CRPS}}^4 + \left\| \mathbb{F}_0 \circ (y^* - \cdot) - \hat{\mathbb{F}}_0 \circ (\hat{y}^* - \cdot) \right\|_{4,\text{CRPS}}^4 \right.$$

$$\left. + 2 \left\| \mathbb{F}_1 \circ y^* - \hat{\mathbb{F}}_1 \circ \hat{y}^* \right\|_{2,\text{CRPS}}^2 \left\| \mathbb{F}_0 \circ (y^* - \cdot) - \hat{\mathbb{F}}_0 \circ (\hat{y}^* - \cdot) \right\|_{2,\text{CRPS}}^2 \right)$$

$$+ \frac{1}{2\delta_4} \|C_{y^*}\|_{L_2}^2 \|y^* - \hat{y}^*\|_{4,\text{CRPS}}^4 ,$$

where $C_{I,\hat{\eta}}(\delta, x) = |I(x, \hat{\eta})|$; $C_{y^*}(\delta, x) = I(x; \eta)\frac{1}{2} \left| \frac{\mathrm{d}^2}{\mathrm{d}y^2} \left[ \mathbb{F}_1(y \mid x) - \mathbb{F}_0(y - \delta \mid x) \right] \right|_{y=\tilde{y}^*}$; and $C_{\mathbb{F},\hat{y}^*}(\delta, x) = \mathbb{F}_1(\hat{y}^* \mid x) - \mathbb{F}_0(\hat{y}^* - \delta \mid x)$.

The quasi-oracle efficiency of the $W_2^2$ loss follows similarly. $\qquad\square$

# E   Scaling in pseudo-CDFs and pseudo-quantiles

Fig. 9 provides a visual comparison of different pseudo-CDFs for our *AU-learner*, with and without scaling $\gamma$. Therein, we see that the scaling hyperparameter helps to enforce the constraints on the pseudo-CDFs (i. e., $[0, 1]$-boundedness and monotonicity).

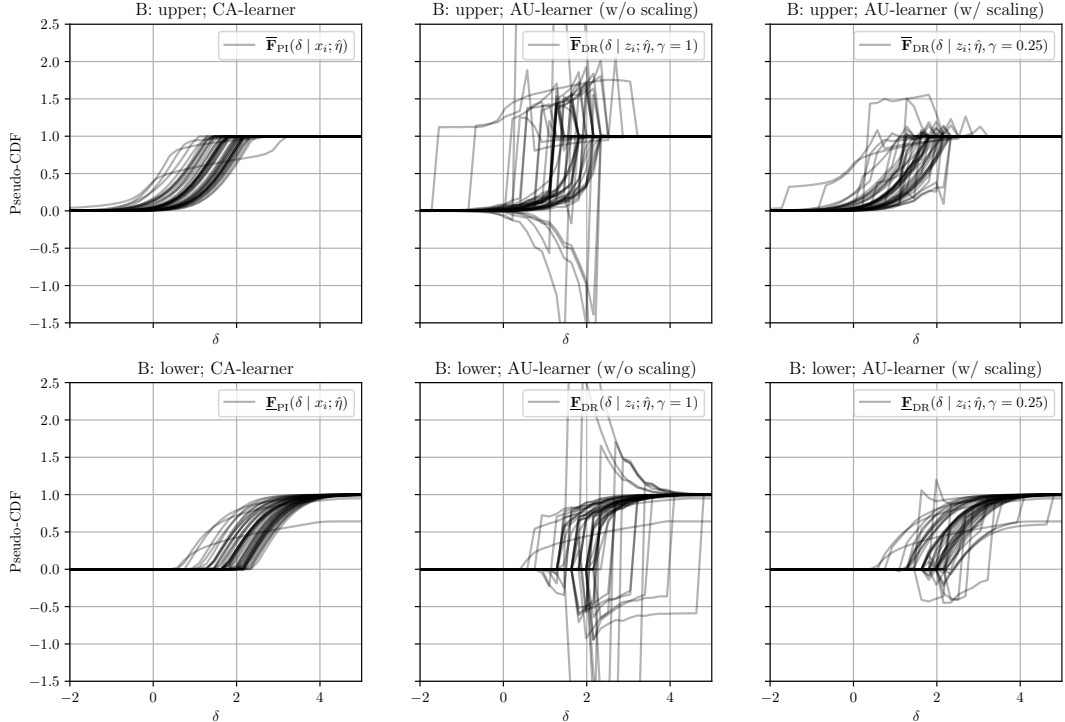

Figure 9: Comparison of estimated pseudo-CDFs based on $i = \{1, \ldots, 50\}$ instances of the semi-synthetic IHDP100 dataset [46]. Here, we compare CA-learner's pseudo-CDFs (first column) with two variants of *AU-learner*: w/o scaling ($\gamma = 1$, second column), and w/ scaling ($\gamma = 0.25$, third column). The scaling hyperparameter $\gamma = 0.25$ facilitates pseudo-CDFs to better comply with $[0, 1]$-boundedness and monotonicity constraints.

# F  Details on AU-CNFs

## F.1  Architecture

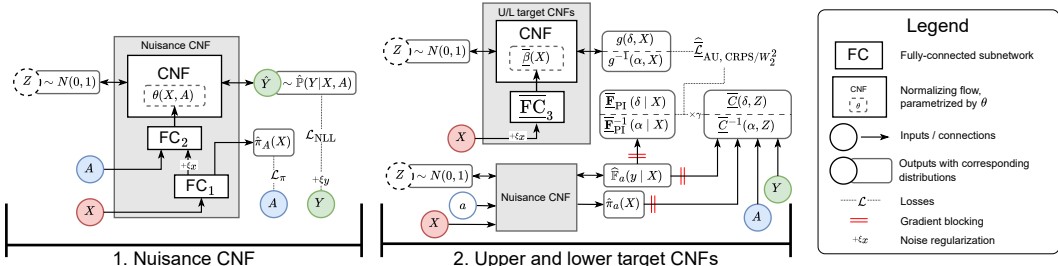

Figure 10: Overview of our AU-CNFs. AU-CNFs combine several conditional normalizing flows (CNFs), which we call a nuisance CNF and upper/lower target CNFs. The nuisance CNF is a first stage model and aims at estimating the nuisance functions, i.e., the propensity score, $\hat{\pi}_a(x) = a\hat{\pi}(x) + (1-a)\hat{\pi}(x)$; and the conditional outcome CDFs, $\widehat{F}_a(y \mid x)$. Upper/lower target CNFs are the second stage working models, $\overline{\mathcal{G}}$ and $\underline{\mathcal{G}}$, respectively. They aim at minimizing one of the losses of AU-learner, $\widehat{\underline{\mathcal{L}}}_{\text{AU, CRPS}/W_2^2}$.

Our AU-CNFs allow us to implement the Algorithm 1 of our *AU-learner* (see Fig. 10) by combining several conditional normalizing flows (CNFs) [105, 121]. It consists of a (i) nuisance CNF and (ii) two target CNFs (upper and lower). (1) The nuisance CNF aims to fit the nuisance functions, $(\hat{\pi}, \widehat{\mathbb{F}}_0, \widehat{\mathbb{F}}_1)$ or, equivalently, $(\hat{\pi}, \widehat{\mathbb{F}}_0^{-1}, \widehat{\mathbb{F}}_1^{-1})$. (2) Upper and lower target CNFs constitute the second stage working models, namely, $\overline{\mathcal{G}}$ and $\underline{\mathcal{G}}$, and minimize the loss of our *AU-learner*.

**(1) Nuisance CNF.** The nuisance CNF has three components, similarly to [94]. These are two fully-connected subnetworks (FC$_1$ and FC$_2$) and a CNF, parametrized by $\theta$. The two subnetworks FC$_1$ and FC$_2$ form a hypernetwork, which outputs the conditional parameters, $\theta = \theta(X, A)$. This allows us to flexibly model the conditional outcome distribution.

The nuisance CNF has the following joint loss for the nuisance functions: $\mathcal{L}_N = \mathcal{L}_{\text{NLL}} + \alpha\mathcal{L}_\pi$. Here, $\mathcal{L}_{\text{NLL}}$ is a conditional negative log-likelihood loss, $\mathcal{L}_\pi$ is a binary cross-entropy, and $\alpha > 0$ is a hyperparameter. We additionally employed noise regularization to regularize the conditional negative log-likelihood loss [108].

**(2) Upper and lower target CNFs.** The upper and lower target CNFs use the pseudo-CDFs / pseudo-quantiles, generated by the nuisance CNF, and then implement a second stage loss of our *AU-learner*. Both target CNFs have the same structure. Specifically, they have a fully-connected subnetwork, $\overline{\text{FC}}_3$, and a CNF, parametrized by $\overline{\beta}$. Analogously, $\overline{\text{FC}}_3$ serves as a hypernetwork so that the parameters can be conditioned on $X$: $\overline{\beta} = \overline{\overline{\beta}}(X)$.

To fit the target CNFs, we use a second stage loss of our *AU-learner*, namely, Eq. (13) or Eq. (14). For that, we discretize the $\mathcal{Y}$-space or the $[0, 1]$-interval of $u$ into $n_d$ values and infer argmin/argmax values based on those grids. Then, to approximate the integrals, we do the same for the $\Delta$-space and the $[0, 1]$-interval of $\alpha$. The later creates a $\delta/\alpha$-grid with $n_\delta/n_\alpha$ points. Those grids are later used for a rectangle quadrature integration. Furthermore, we also regularize the target CNFs by applying the noise regularization [108].

## F.2  Implementation

**Implementation.** We implemented our AU-CNFs using PyTorch and Pyro. For the CNFs of both stages of learning, we used neural spline flows [24] with a standard normal distribution as a base distribution. Neural spline flows build an invertible transformation based on invertible rational-quadratic splines and, thus, allow the direct inference of the (conditional) log-probability, CDF, and quantiles. Neural spline flows are characterized by two main hyperparameters, namely, a number of knots $n_{\text{knots}}$ and a span of the transformation interval, $[-B, B]$. The number of knots, $n_{\text{knots}}$, controls the expressiveness of the flow. The span $B$ defines the support of the transformation. In

our experiments, we tune the number of knots $n_{\text{knots}}$ and set the span $B$ via a heuristic depending on sample max/min values (as $\mathcal{Y}$ is assumed to be compact).

**Training.** To train our AU-CNFs, we make use of Algorithm 1. However, both first and second stage models are fit on the same training data $\mathcal{D}$ without cross-fitting as (regularized) CNFs as neural networks belong to the Donsker class of estimators [123]. Training of our AU-CNFs proceeds as follows: (1) we fit the nuisance CNF; (2) we freeze the nuisance CNF and generate pseudo-CDFs/pseudo-quantiles; and (3) we train the upper and lower target CNFs. The hyperparameters are then as follows:

1. **First stage.** We used stochastic gradient descent (SGD) with a minibatch size $b_{\text{N}}$, $n_{e,\text{N}} = 200$ epochs and a learning rate $\eta_{\text{N}}$. Furthermore, we set the loss coefficient to $\alpha = 1$. Both the number of hidden units of $FC_1/FC_2$ and the size of the output of the $FC_1$ are set to 10. The nuisance CNF has the number of knots $n_{\text{knots, N}}$ and the span $B = \max_i(\tilde{y}_i) - \min_i(\tilde{y}_i) + 5$, where $\tilde{y}_i$ are standard normalized outcomes $y_i$. The intensities of the noise regularization for the input and the output are set to $\sigma_x^2$ and $\sigma_y^2$, respectively.

2. **Intermediate stage.** We set $n_d = 200$ and $n_\delta = n_\alpha = 50$. Furthermore, we set the scaling hyperparameters $\gamma = 0.25$ for the CRPS loss and $\gamma = 0.01$ for the $W_2^2$ loss. We clipped too low propensity scores (lower than 0.05).

3. **Second stage.** The upper and lower target CNFs are also fit via SGD with the minibatch size $b_{\text{T}} = 64$, $n_{e,\text{T}} = 200$ epochs, and the learning rate $\eta_{\text{T}} = 0.005$. The intensities of the noise regularization for the input are the same as for the nuisance CNF, $\sigma_x^2$. The number of hidden units of $FC_3$ is also set to 10. The target CNFs have the number of knots twice larger than the nuisance flow, $n_{\text{knots, T}} = 2\, n_{\text{knots, N}}$, and the span $B = \max_i(a_i\, \tilde{y}_i) - \min_i(a_i\, \tilde{y}_i) + \max_i((1 - a_i)\, \tilde{y}_i) - \min_i((1 - a_i)\, \tilde{y}_i) + 5$, where $\tilde{y}_i$ are standard normalized outcomes $y_i$. To further stabilize the training of the target CNFs, we employed an exponential moving average (EMA) of the target CNFs parameters [103] with a smoothing hyperparameter $\lambda = 0.995$.

We demonstrate the detailed training procedure of our AU-CNFs (CRPS) in Algorithm 2 (AU-CNFs ($W_2^2$) follow analogously).

**Hyperparameter tuning.** We performed extensive hyperparameter tuning only for the nuisance CNF. The following hyperparameters are subjects to tuning: the minibatch size $n_{b,\text{N}}$, the learning rate $\eta_{\text{N}}$, the number of knots $n_{\text{knots, N}}$, and the intensities of the noise regularization, $\sigma_x^2$ and $\sigma_y^2$. Further details of hyperparameter tuning are provided in Appendix G. The hyperparameters of the target CNFs for all the experiments are either kept fixed or are inherited from the nuisance CNF.

**Algorithm 2** Training procedure of our AU-CNFs (CRPS)

---

1: **Input.** Training dataset $\mathcal{D} = \{x_i, a_i, y_i\}_{i=1}^n$; scaling $\gamma \in (0, 1]$; hyperparameter $\alpha$; number of epochs $n_{e,\mathrm{N}}, n_{e,\mathrm{T}}$; minibatch sizes $b_\mathrm{N}, b_\mathrm{T}$; learning rates $\eta_\mathrm{N}, \eta_\mathrm{T}$; intensities of the noise regularization $\sigma_x^2, \sigma_y^2$; EMA smoothing $\lambda$; $\delta$-grid $\{\delta_j \in \Delta\}_{j=1}^{n_\delta}$; $\mathcal{Y}$-grid $\{y_j \in \mathcal{Y}\}_{j=1}^{n_d}$

2: **Init.** Parameters of the nuisance CNF: $\mathrm{FC}_1^{(0)}$ and $\mathrm{FC}_2^{(0)}$ ▷ *First stage*

3: **for** $i = 0$ **to** $\lceil n_{e,\mathrm{N}} \cdot n/b_\mathrm{N} \rceil$ **do**

4:      Draw a minibatch $\mathcal{B} = \{X, A, Y\}$ of size $b_\mathrm{N}$ from $\mathcal{D}$

5:      $(R, \hat{\pi}_a(X)) \leftarrow \mathrm{FC}_1^{(i)}(X)$

6:      Noise regularization: $\xi_x \sim N(0, \sigma_x^2), \xi_y \sim N(0, \sigma_y^2), \quad (\tilde{R}, \tilde{Y}) \leftarrow (R + \xi_x, Y + \xi_y)$

7:      $\theta(X, A) \leftarrow \mathrm{FC}_2^{(i)}(A, \tilde{R})$

8:      $\hat{\mathbb{P}}(Y \mid X, A) \leftarrow$ density of a CNF with parameters $\theta(X, A)$

9:      $\hat{\mathcal{L}}_\mathrm{N}(\hat{\mathbb{P}}, \hat{\pi}) \leftarrow \mathbb{P}_{b_\mathrm{N}}\{-\log \hat{\mathbb{P}}(Y = \tilde{Y} \mid X, A) + \alpha \, \mathrm{BCE}(\hat{\pi}_A(X), A)\}$

10:      $(\mathrm{FC}_1^{(i+1)}, \mathrm{FC}_2^{(i+1)}) \leftarrow$ optimization step wrt. $\hat{\mathcal{L}}_\mathrm{N}(\hat{\mathbb{P}}, \hat{\pi})$ with the learning rate $\eta_\mathrm{N}$

11: **end for**

12: **Output.** Estimator of the nuisance functions $\hat{\eta} = (\hat{\pi}, \widehat{\mathbb{F}}_0, \widehat{\mathbb{F}}_1)$

13: **for** $i = 0$ **to** $n$ **do** ▷ *Intermediate stage*

14:      Infer argmax/argmin of sup/inf-convolutions based on the $\mathcal{Y}$-grid: $\{\hat{y}_{\mathcal{Y}\text{-grid}}^{\overline{*}}(\delta_j \mid x_i)\}_{j=1}^{n_\delta}$

15:      Clip propensity scores for bias-correction terms $\{\overline{\underline{C}}(\delta_j, z_i; \hat{\eta})\}_{j=1}^{n_\delta}$ (Eq. (11))

16:      Use $\hat{\eta}$ to infer the pseudo-CDF for the $\delta$-grid: $\{\overline{\underline{\mathbf{F}}}_\mathrm{AU}(\delta_j, z_i; \hat{\eta})\}_{j=1}^{n_\delta}$ (Eq. (13))

17: **end for**

18: **Init.** Parameters of the upper and lower target CNFs: $\overline{\mathrm{FC}}_3^{(0)}$ and $\underline{\mathrm{FC}}_3^{(0)}$ ▷ *Second stage*

19: **for** $i = 0$ **to** $\lceil n_{e,\mathrm{T}} \cdot n/b_\mathrm{T} \rceil$ **do**

20:      Draw a minibatch $\mathcal{B} = \{X, A, Y\}$ of size $b_\mathrm{T}$ from $\mathcal{D}$

21:      Noise regularization: $\xi_x \sim N(0, \sigma_x^2), \quad \tilde{X} \leftarrow X + \xi_x$

22:      $\overline{\beta(X)} \leftarrow \overline{\mathrm{FC}}_3^{(i)}(\tilde{X}), \quad \underline{\beta(X)} \leftarrow \underline{\mathrm{FC}}_3^{(i)}(\tilde{X})$

23:      $\widehat{\overline{g}}(\cdot, X) \leftarrow$ CDF of a CNF with parameters $\overline{\beta(X)}, \quad \widehat{\underline{g}}(\cdot, X) \leftarrow$ CDF of a CNF with parameters $\underline{\beta(X)}$

24:      $\widehat{\overline{\mathcal{L}}}_{\mathrm{AU, CRPS}}(\widehat{\overline{g}}, \hat{\eta}) = \mathbb{P}_{b_\mathrm{T}}\{\frac{1}{n_\delta} \sum_{j=1}^{n_\delta} (\overline{\mathbf{F}}_\mathrm{AU}(\delta_j, Z; \hat{\eta}) - \widehat{\overline{g}}(\delta_j, X))^2\}$

25:      $\widehat{\underline{\mathcal{L}}}_{\mathrm{AU, CRPS}}(\widehat{\underline{g}}, \hat{\eta}) = \mathbb{P}_{b_\mathrm{T}}\{\frac{1}{n_\delta} \sum_{j=1}^{n_\delta} (\underline{\mathbf{F}}_\mathrm{AU}(\delta_j, Z; \hat{\eta}) - \widehat{\underline{g}}(\delta_j, X))^2\}$

26:      $\overline{\mathrm{FC}}_3^{(i+1)} \leftarrow$ optimization step wrt. $\widehat{\overline{\mathcal{L}}}_{\mathrm{AU, CRPS}}(\widehat{\overline{g}}, \hat{\eta})$ with the learning rate $\eta_\mathrm{T}$

27:      $\underline{\mathrm{FC}}_3^{(i+1)} \leftarrow$ optimization step wrt. $\widehat{\underline{\mathcal{L}}}_{\mathrm{AU, CRPS}}(\widehat{\underline{g}}, \hat{\eta})$ with the learning rate $\eta_\mathrm{T}$

28:      EMA update for $\overline{\mathrm{FC}}_3^{(i+1)}$ and $\underline{\mathrm{FC}}_3^{(i+1)}$ with smoothing $\lambda$

29: **end for**

30: **Output.** Estimator of Makarov bounds with CNFs: EMA smoothed $\widehat{\overline{g}}$ and $\widehat{\underline{g}}$

---

# G  Hyperparameter tuning

We performed hyperparameters tuning of the nuisance function estimators for all the baselines based on five-fold cross-validation using the training dataset. For each baseline, we did a grid search wrt. different tuning criteria (see the details in Table 3). The optimal hyperparameters can be found as YAML files in our GitHub.

Table 3: Hyperparameter tuning for baselines.

| Model | Sub-model | Hyperparameter | Range / Value |
|---|---|---|---|
| Plug-in DKME [94, 97] | — | Kernel smoothness ($\sigma_k = 2h_k^2$) | 0.0001, 0.001, 0.01, 0.1, 1, 10, 20 |
| | | Regularization parameter ($\varepsilon$) | 0.0001, 0.001, 0.01, 0.1, 1, 10 |
| | | Tuning strategy | full grid search |
| | | Tuning criterion | MSE of ridge regression |
| CA-CNFs, AU-CNFs | nuisance CNF ($\hat{=}$ Plug-in CNF) ($\hat{=}$ IPTW-CNF) | Number of knots ($n_{\text{knots,N}}$) | 5, 10, 20 |
| | | Intensity of noise regularization ($\sigma_x^2$) | $0.0, 0.01^2, 0.05^2, 0.1^2$ |
| | | Intensity of noise regularization ($\sigma_y^2$) | $0.0, 0.01^2, 0.05^2, 0.1^2$ |
| | | Learning rate ($\eta_{\text{N}}$) | 0.001, 0.005 |
| | | Minibatch size ($b_{\text{N}}$) | 32, 64 |
| | | Tuning strategy | random grid search with 50 runs |
| | | Tuning criterion | $\mathcal{L}_{\text{NLL}}$ |
| | | Number of epochs ($n_{e,\text{N}}$) | 200 |
| | | Optimizer | SGD (momentum = 0.9) |
| | target CNFs | Number of knots ($n_{\text{knots,T}}$) | $2\,n_{\text{knots,N}}$ |
| | | Intensity of noise regularization | $\sigma_x^2$ |
| | | Learning rate ($\eta_{\text{T}}$) | 0.005 |
| | | Minibatch size ($b_{\text{T}}$) | 64 |
| | | Tuning strategy | w/o tuning |
| | | Number of epochs ($n_{e,\text{T}}$) | 200 |
| | | Optimizer | SGD (momentum = 0.9) |

# H  Dataset details

## H.1  Synthetic data

Our synthetic data generator is adapted from [61, 93]. Although the original synthetic benchmark contains hidden confounding, we include the confounder as the second observed covariate. We created three settings with different conditional outcome distributions: (1) normal, (2) multi-modal and (3) exponential. Specifically, synthetic covariates, $X_1, X_2$, a treatment, $A$, and an outcome, $Y$, are sampled from the following data generating mechanisms:

$$\begin{cases} X_1 \sim \text{Unif}(-2, 2), \\ X_2 \sim N(0, 1), \\ A \sim \text{Bern}\left(\frac{1}{1+\exp(-(0.75\,X_1 - X_2 + 0.5))}\right), \\ \mu_A(X) := (2\,A - 1)\,X_1 + A - 2\sin(2\,X_1 + X_2) - 2\,X_2\,(1 + 0.5\,X_1), \\ Y \sim \mathbb{P}_j(Y \mid \mu_A(X), A), \end{cases} \tag{137}$$

where $X_1, X_2$ are mutually independent and $\mathbb{P}_j(Y \mid \mu_A(X), A)$ are defined by three settings $j \in \{1, 2, 3\}$:

$$\mathbb{P}_1(Y \mid \mu_A(X), A) = N(\mu_A(X), 1), \tag{138}$$

$$\mathbb{P}_2(Y \mid \mu_A(X), A) = \begin{cases} \text{Mixture}\left\{0.7\,N(\mu_0(X) - 0.5, 1.5^2) + 0.3\,N(\mu_0(X) + 1.5, 0.5^2)\right\}, & \text{if } A = 0, \\ \text{Mixture}\left\{0.3\,N(\mu_0(X) - 2.5, 0.35^2) + 0.4\,N(\mu_0(X) + 0.5, 0.75^2) + 0.3\,N(\mu_0(X) + 2, 0.5^2)\right\}, & \text{if } A = 1, \end{cases} \tag{139}$$

$$\mathbb{P}_2(Y \mid \mu_A(X), A) = \text{Exp}(1/\,|\mu_A(X)|), \tag{140}$$

where $N(\mu, \sigma^2)$ is the normal distribution and where $\text{Exp}(\lambda)$ is the exponential distribution.

The synthetic benchmark allows us to infer or approximate the ground-truth Makarov bounds. For the (1) normal distribution, they are given by the analytical solution [26], namely, the CDFs of half-normal distributions. However, in the settings (2) and (3), they need to be approximated numerically. Thus, for the (2) multi-modal distribution, we only infer the Makarov bounds on the CDF, as the quantiles are not directly available for the mixture distribution. For the (3) exponential distribution, however, we can infer both the Makarov bounds on the CDF and the quantiles.

## H.2  HC-MNIST dataset

HC-MNIST dataset was introduced as a high-dimensional, semi-synthetic dataset [52] based on the MNIST image dataset [75]. The HC-MNIST dataset builds on $n_{\text{train}} = 60,000$ train and $n_{\text{test}} = 10,000$ test images. HC-MNIST takes original high-dimensional images and maps them onto a one-dimensional manifold, where potential outcomes depend in a complex way on the average intensity of light and the label of an image. The treatment also uses this one-dimensional summary, $\phi$, together with an additional (hidden) synthetic confounder, $U$ (we consider this hidden confounder as another observed covariate). HC-MNIST is then defined by the following data-generating mechanism:

$$\begin{cases} U \sim \text{Bern}(0.5), \\ X \sim \text{MNIST-image}(\cdot), \\ \phi := \left(\text{clip}\left(\frac{\mu_{N_x} - \mu_c}{\sigma_c}; -1.4, 1.4\right) - \text{Min}_c\right)\frac{\text{Max}_c - \text{Min}_c}{1.4 - (-1.4)}, \\ \alpha(\phi; \Gamma^*) := \frac{1}{\Gamma^*\,\text{sigmoid}(0.75\phi + 0.5)} + 1 - \frac{1}{\Gamma^*}, \\ \beta(\phi; \Gamma^*) := \frac{\Gamma^*}{\text{sigmoid}(0.75\phi + 0.5)} + 1 - \Gamma^*, \\ A \sim \text{Bern}\left(\frac{u}{\alpha(\phi;\Gamma^*)} + \frac{1-u}{\beta(\phi;\Gamma^*)}\right), \\ Y \sim N\left((2A - 1)\phi + (2A - 1) - 2\sin(2(2A - 1)\phi) - 2(2U - 1)(1 + 0.5\phi), 1\right), \end{cases} \tag{141}$$

where $c$ is a label of the digit from the sampled image $X$; $\mu_{N_x}$ is the average intensity of the sampled image; $\mu_c$ and $\sigma_c$ are the mean and standard deviation of the average intensities of the images with the label $c$; and $\text{Min}_c = -2 + \frac{4}{10}c, \text{Max}_c = -2 + \frac{4}{10}(c + 1)$. The parameter $\Gamma^*$ defines what factor influences the treatment assignment to a larger extent, i.e., the additional confounder or the one-dimensional summary. We set $\Gamma^* = \exp(1)$. For further details, we refer to [52].

Similarly to the synthetic data with the normal distribution, the ground-truth Makarov bounds for the HC-MNIST dataset are given by the analytical solution, namely, the CDFs of half-normal distributions [26].

### H.3 IHDP100 dataset

The Infant Health and Development Program (IHDP100) [46, 114] is a standard semi-synthetic benchmark for treatment effect estimation. It contains 100 train/test splits with $n_{\text{train}} = 672$, $n_{\text{test}} = 75$, and $d_x = 25$. Yet, this dataset contains severe overlap violations, which makes the methods using propensity re-weighting unstable [20, 21].

The IHDP100 dataset samples synthetic outcomes from the conditional normal distribution, $N(\mu_i, 1)$, where $\mu_i$ are CAPOs provided in the dataset. Therefore, the ground-truth Makarov bounds are given by the half-normal distributions [26].

# I  Additional results

In the following, we provide additional results for our experiments with synthetic data, the results of the semi-synthetic IHDP100 benchmark, and the runtime information for all the baselines.

## I.1  Synthetic data

We provide additional results of our synthetic benchmark in Fig. 11. Therein, the out-of-sample performance is reported wrt. $W_2$ evaluation score for two settings, i.e., normal and exponential setting.[10] Our AU-CNFs achieve superior performance in the normal setting and perform well in the exponential setting. Notably, the IPTW-CNF also perform well in the exponential setting, mainly due to the orthogonality wrt. potential outcome distributions.

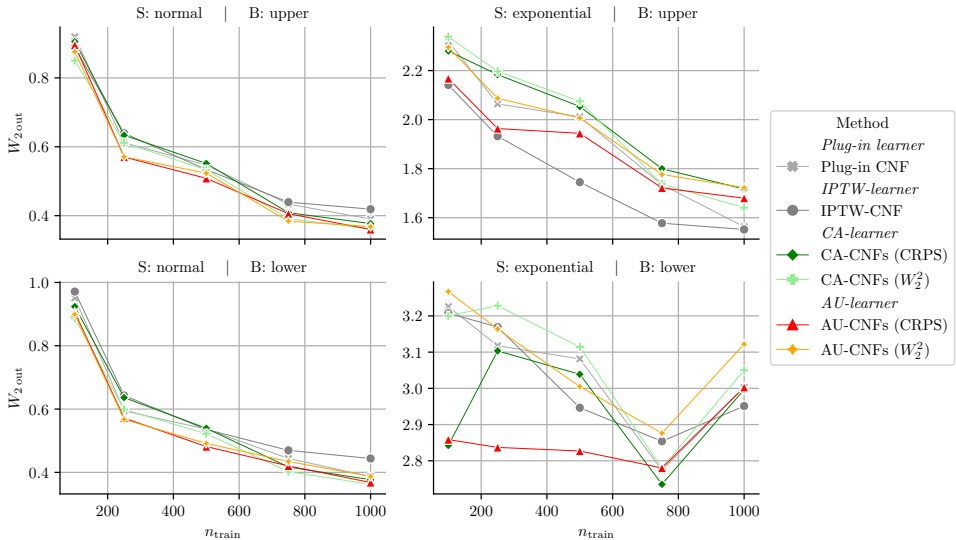

Figure 11: Results for synthetic experiments with varying size of training data, $n_{\text{train}}$, in the settings: normal and exponential setting. Reported: mean out-sample $W_2$ over 20 runs. The results for Plug-in DKME are omitted for the $W_2$ evaluation score, as Plug-in DKME does not provide direct quantiles inference.

## I.2  IHDP100 dataset

We report the in- and out-sample results for the IHDP100 dataset in Table 4. As expected, our CA-CNFs achieve the best performance. This happens due to the severe overlap violations in the IHDP100 dataset [20, 21], and learners with propensity-score re-weighting are theoretically expected to perform worse.

## I.3  Runtime comparison

Table 5 provides the runtime comparison of different methods used to estimate Makarov bounds. Therein, our AU-CNFs are well scalable.

---

[10]We omitted the multi-modal setting, as the mixture distribution does not provide direct ground-truth quantiles and, thus, Makarov bounds on the quantiles.

Table 4: Results for IHDP100 dataset. Reported: median in-sample and out-sample rCRPS $\pm$ sd / $W_2 \pm$ sd over 100 train/test splits. The results for Plug-in DKME are omitted for the $W_2$ evaluation score, as Plug-in DKME does not provide direct quantiles inference.

| | B: upper | | | |
| --- | --- | --- | --- | --- |
| | $\text{rCRPS}_{\text{in}}$ | $\text{rCRPS}_{\text{out}}$ | $W_{2\,\text{in}}$ | $W_{2\,\text{out}}$ |
| Plug-in DKME | $0.718 \pm 0.831$ | $0.778 \pm 0.865$ | — | — |
| Plug-in CNF | $0.302 \pm 0.269$ | $0.317 \pm 0.260$ | $0.631 \pm 1.195$ | $0.644 \pm 1.237$ |
| IPTW-CNF | $0.367 \pm 0.198$ | $0.380 \pm 0.206$ | $0.783 \pm 0.971$ | $0.783 \pm 1.067$ |
| CA-CNFs (CRPS) | $\mathbf{0.281 \pm 0.284}$ | $0.299 \pm 0.294$ | $\underline{0.609 \pm 1.508}$ | $\underline{0.649 \pm 1.625}$ |
| CA-CNFs ($W_2^2$) | $\underline{0.283 \pm 0.273}$ | $\mathbf{0.292 \pm 0.284}$ | $\mathbf{0.583 \pm 1.318}$ | $\mathbf{0.606 \pm 1.425}$ |
| AU-CNFs (CRPS) | $0.314 \pm 0.264$ | $0.324 \pm 0.278$ | $0.681 \pm 1.385$ | $0.709 \pm 1.503$ |
| AU-CNFs ($W_2^2$) | $0.285 \pm 0.300$ | $\underline{0.297 \pm 0.308}$ | $0.589 \pm 1.375$ | $0.620 \pm 1.467$ |

| | B: lower | | | |
| --- | --- | --- | --- | --- |
| | $\text{rCRPS}_{\text{in}}$ | $\text{rCRPS}_{\text{out}}$ | $W_{2\,\text{in}}$ | $W_{2\,\text{out}}$ |
| Plug-in DKME | $0.709 \pm 0.827$ | $0.757 \pm 0.866$ | — | — |
| Plug-in CNF | $0.312 \pm 0.212$ | $0.326 \pm 0.227$ | $0.650 \pm 0.973$ | $0.660 \pm 1.073$ |
| IPTW-CNF | $0.384 \pm 0.208$ | $0.381 \pm 0.227$ | $0.814 \pm 1.028$ | $0.781 \pm 1.193$ |
| CA-CNFs (CRPS) | $\mathbf{0.303 \pm 0.310}$ | $\mathbf{0.308 \pm 0.329}$ | $\underline{0.675 \pm 1.617}$ | $\underline{0.658 \pm 1.797}$ |
| CA-CNFs ($W_2^2$) | $\underline{0.307 \pm 0.302}$ | $\underline{0.311 \pm 0.323}$ | $\mathbf{0.639 \pm 1.451}$ | $\mathbf{0.642 \pm 1.625}$ |
| AU-CNFs (CRPS) | $0.330 \pm 0.254$ | $0.344 \pm 0.267$ | $0.726 \pm 1.379$ | $0.744 \pm 1.550$ |
| AU-CNFs ($W_2^2$) | $0.308 \pm 0.284$ | $0.314 \pm 0.304$ | $0.642 \pm 1.362$ | $0.647 \pm 1.549$ |

Lower = better (best in bold, second best underlined)

| Method | Training stages | Average duration (in mins) |
| --- | --- | --- |
| Plug-in DKME | First stage | $\approx 10.7$ |
| Plug-in CNFs | First stage | $\approx 1.4$ |
| IPTW-CNF | First stage | $\approx 1.5$ |
| CA-CNFs | First & second stages | $\approx 3.3$ |
| AU-CNFs | First & second stages | $\approx 4.1$ |

Table 5: Total runtime (in seconds) for different methods to estimate Makarov bounds. Reported: average runtime duration (lower is better). Experiments were carried out on 2 GPUs (NVIDIA A100-PCIE-40GB) with IntelXeon Silver 4316 CPUs @ 2.30GHz.

## J  Case study: Lockdown effectiveness

In the following, we provide a case study, where we apply our *AU-learner* to a real-world problem. Here, we want to study the effectiveness of lockdowns during the COVID-19 pandemic by using the observational data collected in the first half-year of 2020 [7]. Specifically, we aim to estimate the probability that the incidence falls after the implementation of the strict lockdown, i. e., a probability of individual benefit from treatment (intervention) (PITB).

### J.1  Dataset

We used multi-county data provided by [7].[11] The outcome $Y \in [-7, 0]$ is defined as the relative case growth per week (in log), namely, the number of new cases divided by the number of cumulative cases. Then, the treatment $A \in \{0, 1\}$ is taken as an implementation of the strict lockdown one week before. We also choose three ($d_x = 3$) pre-treatment covariates $X$: the relative case growths from the previous week, the relative case growth from two weeks ago, and the implementation of the strict lockdown from two weeks ago. We assume that the data is i.i.d. and that the causal assumptions (1)–(3) are satisfied. We filtered out observations where the number of cumulative cases is fewer than 20. As a result, we ended up with $n = n_0 + n_1 = 152 + 112$ treated and untreated observations, respectively.

### J.2  Results

We present the results for our case study in Fig. 12. Therein, we report two quantities: the estimated bounds on the probability of the individual treatment benefit (PITB), $\mathbb{P}(Y[1] - Y[0] \leq 0 \mid x)$, for 20 countries during week 15 of 2020. Additionally, we show bounds on a population analogue of PITB, namely, a probability of the population treatment benefit, $\mathbb{P}(Y[1] - Y[0] \leq 0)$. We estimated the bounds on the PITB with both AU-CNFs (CRPS) and AU-CNFs ($W_2^2$), and both methods produced very similar results (which implies a robustness of our *AU-learner*). For the population analogue of the PITB, we first efficiently estimated the distributions of the potential outcomes with interventional normalizing flows (INFs) [94] and then used them to infer the Makarov bounds at the population level.

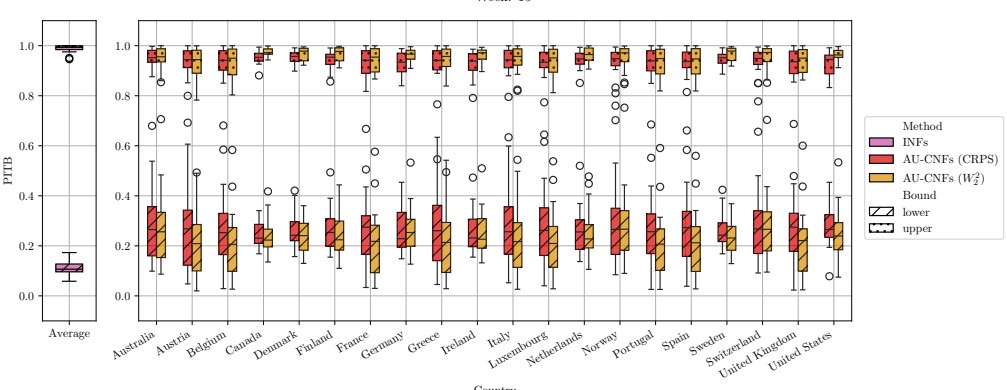

Figure 12: Results for the real-world case study analyzing the effectiveness of lockdowns during the COVID-19 pandemic. Reported: in-sample estimated bounds on the probability of the individual treatment benefit (PITB), $\mathbb{P}(Y[1] - Y[0] \leq 0 \mid x)$, over 20 runs for 20 countries during week 15 of 2020. Also, we show bounds on a probability of the population treatment benefit, i.e., $\mathbb{P}(Y[1] - Y[0] \leq 0)$. These are displayed in the small figure on the left. Each estimated bound is shown as two boxplots; hence, we also display the epistemic uncertainty.

There are two important takeaways: (1) The bounds on the PITB are more shifted towards 1, suggesting the drop in the incidence is highly probable after the implementation of the strict lockdown in all the studied countries. (2) The bounds on PITB are much tighter than their population analogue (e. g., average upper-lower bound width is 0.66 for AU-CNFs (CRPS) and 0.88 for INFs). The latter implies that individualization enhances decision-making and makes the Makarov bounds on the aleatoric uncertainty tighter and, thus, more informative.

---

[11]The data is available at `https://github.com/nbanho/npi_effectiveness_first_wave/blob/master/data/data_preprocessed.csv`.

