# OpenReview forum: "Quantifying Aleatoric Uncertainty of the Treatment Effect: A Novel Orthogonal Learner"
_NeurIPS.cc/2024/Conference — NeurIPS 2024 poster_

### Official Review · Reviewer_8SMd · 2024-06-18

**Soundness:** 4
**Presentation:** 1
**Contribution:** 2
**Rating:** 3
**Confidence:** 5

**Summary:**

The paper provides the orthogonal estimator for the distributional treatment effect (the conditional CDF of $Y[1]-Y[0]$).

**Strengths:**

This paper is technically strong, demonstrating a high level of mathematic rigor and diligence. It presents an in-depth description of the proposed estimator. I think the proposed estimator is useful in practice.

**Weaknesses:**

Despite the strong technical details, the paper is poorly written in overall.

__1. Weak motivation__

A current shape of Introduction is weakly motivated. Firstly, the way that the paper motivates the problem is misleading:
> Methods for quantifying the aleatoric uncertainty of the treatment effect have gained surprisingly little attention in the causal machine learning community.

This sentence is incorrect, since there are literatures on quantile regression and semiparametric density estimation, as reviewed in Section 2.

More importantly, the introduction doesn't provide the motivation of the problem against the following question: _why a community need the proposed estimator, given that the quantile estimator can capture the distributional treatment effect_.


__2. Difficult to understand due to insufficient information__

Another issue that the paper has (especially in Introduction) is its lack of back-ground information that readers may need to comprehend. Specifically, in Introduction, the paper doesn’t provide any definition or clue what CDTE is. I understand that the CDTE is defined in the caption of Figure 1 as $P(Y[1] - Y[0] \leq \delta \mid x)$. However, $Y[a]$ is undefined, and this key quantity should be in the main body of the text. Also, even if Figure 1 aims to provide a whole summary of the paper, authors at Introduction have insufficient knowledge to comprehend it. In other words, Figure 1 is too detailed to be presented in Introduction section. Since Figure 1 can only be understood by those who entirely digested the paper from the beginning to the end, Introduction section is not the right position where the Figure 1 is located. I understand the goal of Figure 1, but it doesn't achieve the goal because of insufficient background information. The same issue happens to Figure 2 and Table 1. For example, in Table 1, technical terms like AIPTW, hold-out residual, optimization assumptions are undefined, so it's hard to appreciate the contribution of this paper.

__3. Fuzzy description on contribution__

First, the terms like "Aleatoric uncertainty" and "distributional treatment effect" are used for denoting the same target quantity. Given that the "distributional treatment effect" is clearly describing the problem, I don't see why the authors want to use "Aleatoric uncertainty" as a title and employ these two words to denote the same estimand.

Second, the contribution is wrongly described. Consider this sentence:
> AU-learner solves all of the above-mentioned challenges 1 – 3 .

The AU-learner doesn't address Challenge 1 and 2. Challenge 1 means that the distributional treatment effect is not identifiable. The Makarov's bound, _NOT_ AU-learner, is employed to address Challenge 1. Challenge 2 is actually the same as Challenge 1, since it means that there are no known nuisances-based functional for the distributional treatment effect. Again, the Makarov's bound is used to address Challenge 2. AU-learners are representing the approximated quantity of the target estimand in terms of nuisance functionals.

Finally, third contribution "flexible deep learning instantiation of our AU-learner" is scarcely described only in Section 5. The description needs to be much improved.

__4. Little focus on the real contribution__

The real contribution of this paper, compared to the existing works in Table 1, is to provide the doubly robust conditional distributional treatment effect for the bounds of the conditional CDF of treatment effects. However, little focus and efforts have been made for this contribution. For example, if developing a doubly robust estimator is a contribution, then corresponding results such as detailed error analysis, a closed form of estimators, a detailed recipe of the proposed estimator for the specified working model, how to minimize the losses in Equations (8,9), a simple example, and assumptions should be described.

**Questions:**

1. Are rate-doubly-robustness and Neyman orthogonality violated when the scaling hyperpameters are not $1$?

2. Is the Makarov bound sharp?

3. What are the practical examples of the distributional treatment effect?

4. In line 178, is this phenomenon officially termed the selection bias?

**Limitations:**

The paper is limited to the setting where the ignitability holds.

---

> ### Author Rebuttal · Authors · 2024-08-04
>
> First of all, thank you for the detailed and positive review of our paper. Below, we respond to the mentioned weaknesses and questions. Importantly, all the issues will be easily fixed for the camera-ready version of the manuscript.
>
> ### Response to weaknesses
> 1. We want to stress that there are several – seemingly similar – research streams connecting the aleatoric uncertainty and potential outcomes/treatment effects, yet these have **different causal quantities** with **crucial differences**:
>    - _Aleatoric uncertainty of potential outcomes._ This stream includes works on quantile regression and semiparametric density estimation. Here, the causal quantities are **point identifiable** and given by an explicit functional of the nuisance functions.
>    - _Distributional treatment effects._ This stream includes all the works, which infer and estimate the distributional distances/divergences between the potential outcomes distributions (e.g., Wasserstein distances, KL-divergence, or the quantile differences).   Here, similarly, the causal quantities are **point-identifiable**.
>    - _Aleatoric uncertainty of the treatment effect (= distribution of the treatment effect)._ Our paper is located in this stream. Therein, the casual quantities are, for example, the variance of the treatment effect, and the CDF/quantiles of the treatment effect (our paper). Importantly, they are **not point-identifiable** and given by the implicit functional of the nuisance functions.
>
>    The **above-mentioned streams of literature contain different causal quantities with different interpretations**. For example: quantile differences != quantiles of the difference (treatment effect). Further: the distributional treatment effects != the distribution of the treatment effect. Hence, we argue that the original statement holds, given the differences in terminology. Nevertheless, we realized upon reading your comment that we need to elaborate on the differences more carefully.
>
>    **Action**. We will elaborate more carefully on the above differences in Appendix A (Extended Related Work) and thereby point out how our work is novel.
>
> 2. We acknowledge that the Introduction might be hard to understand for non-causal ML practitioners. Initially, we did not want to overburden the Introduction with the exact notation and definitions (apart from the summary in Fig. 1). However, we understand that more explanations are helpful.
>
>    **Action**. We will follow your advice, and provide more explanations for the potential outcomes, $Y[a]$, and for the CDTE. Also, we will simplify Figures 1 and 2, and add the details regarding Table 2.
>
> 3. Again, we argue that the “distributional treatment effects” and “the aleatoric uncertainty of the treatment effect” are two **very different causal quantities** (see Answer to 1.).
>
>    We thank you for your feedback and agree that the sentence “AU-learner solves all of the above-mentioned challenges 1 – 3.” needs to be rephrased. Given that the Makarov bounds were already proposed in the literature, we will improve our text by shifting Challenge 1 to the fact that there were no efficient influence functions (EIFs) derived for the Makarov bounds. The latter is solved in our paper. Then, Challenge 2 would still hold, as both the Makarov bounds and its EIFs are **implicit** functionals of the (estimated) nuisance functions, i.e., we need to perform sup/inf convolutions. This not only complicates the practical implementation of the AU-learner, but also the design of the experiments, as the ground-truth has to be derived in advance or approximately inferred numerically.
>
>    **Action**. We are happy to implement the above-mentioned changes to the final version of the manuscript. Also, we will put more emphasis on the deep-learning instantiation.
>
> 4. **Action**. We are happy to reformulate our contribution so that it is centered around the derivation and implementation of doubly-robust learner (AU-learner).
>
> ### Response to questions
> 1. Yes, when the scaling parameter $\gamma \neq 1$, Neyman-orthogonality does not fully hold (the cross-derivative wrt. CDFs are not zero). Hence, rate-doubly-robustness also does not hold.
>
>    **Action**. We will expand the proof of the Neyman-orthogonality and rate-doubly-robustness in the final version of the paper, so this fact would be more visible.
> 2. Yes, the Makarov bounds are sharp [1]. See the discussion in Appendix B.2 Pointwise and Uniformly Sharp Bounds.
> 3. Although the distributional treatment effects are **not the target of our paper**, the examples include distributional distances between potential outcomes [2]; quantile / super-quantile treatment effects and $f$-risk treatment effects [3], etc. Our paper is focused on the CDF/quantiles of the treatment effect, which is a different causal quantity.
> 4. Yes, it was referred to as a selection bias, e.g., by [4-6]. An alternative name is a covariate shift [6].
>
> ### References:
> - [1] Fan, Yanqin, and Sang Soo Park. "Sharp bounds on the distribution of treatment effects and their statistical inference." Econometric Theory 26.3 (2010): 931-951.
> - [2] Kennedy, Edward H., Sivaraman Balakrishnan, and L. A. Wasserman. "Semiparametric counterfactual density estimation." Biometrika 110.4 (2023): 875-896.
> - [3] Kallus, Nathan, and Miruna Oprescu. "Robust and agnostic learning of conditional distributional treatment effects." AISTATS. PMLR, 2023.
> - [4] Curth, Alicia, and Mihaela van der Schaar. "Nonparametric estimation of heterogeneous treatment effects: From theory to learning algorithms." AISTATS. PMLR, 2021.
> - [5] Alaa, Ahmed, and Mihaela van der Schaar. "Limits of estimating heterogeneous treatment effects: Guidelines for practical algorithm design." ICML. PMLR, 2018.
> - [6] Johansson, Fredrik D., et al. "Generalization bounds and representation learning for estimation of potential outcomes and causal effects." Journal of Machine Learning Research 23.166 (2022): 1-50.

---

> > ### Comment · Reviewer_8SMd · 2024-08-10
> > **Response**
> >
> > Overall, I am embarrassed by this response. The response is written as though the authors' categorization—(1) Aleatoric uncertainty of potential outcomes, (2) Distributional treatment effects, and (3) Aleatoric uncertainty of the treatment effect—has been clearly established in the paper, and my questions/concerns arise from a lack of understanding of this framework (e.g., " the distributional treatment effects are not the target of our paper,", where the term "distribution treatment effect" is just defined by this response, not the paper).
> >
> > However, as stated in the authors' response, this categorization is not mentioned in the paper, and no clues are provided regarding this categorization.
> >
> > Furthermore, this categorization seems counterintuitive -- does it really make sense to state that "distributional treatment effects" and "distribution of treatment effects" are _very different_? I do understand that they are different, and that the distribution of the treatment effect, P(Y[1] - Y[0]), is not pointwise identifiable due to the fundamental problem of causal inference. However, the categorization seems to come out of nowhere and requires justification.
> >
> > I do understand which research streams this work is located in. Using your framework, my questions and concerns (Q3 and W1) were about what practically interesting scenarios "Aleatoric uncertainty of the treatment effect" can capture that other research streams cannot. I don't believe your response has fully addressed this question/concern yet.
> >
> > I adjusted my score based on this response.

---

> ### Author Response · Authors · 2024-08-12
>
> [1/3]
>
> Thank you for the quick response! We sincerely appreciate the time and effort you have taken to provide us with valuable feedback. We are very sorry for the ambiguities in the terminology across existing research research streams and for our original manuscript not providing enough context to resolve them. We further apologize if our previous comments were unclear or could be interpreted as implying a lack of understanding – this was not what we meant, and **we apologize if our response may have come across in the wrong way**. Rather, we feel grateful for having received such thorough and knowledgeable reviews that both commended the quality of our paper and also made further suggestions to improve our work for the camera-ready version.
>
>
>
> ### Regarding the unanswered Q3 and W1
>
> > “Using your framework, my questions and concerns (Q3 and W1) were about what practically interesting scenarios "Aleatoric uncertainty of the treatment effect" can capture that other research streams cannot.”
>
> > “why a community need the proposed estimator, given that the quantile estimator can capture the distributional treatment effect.”
>
> **We apologize for misunderstanding your questions**. In our paper, we work with the aleatoric uncertainty of the treatment effect in the form of the CDF/quantiles of the treatment effect at the covariate-conditional level. More specifically, we aim at inferring the probability that the treatment effect is less than or equal to a certain value ($\delta$), conditional on the covariates. For $\delta=0$, the latter becomes the covariate-conditional probability of the treatment harm (benefit), i.e., $\mathbb{P}(Y[1] \le Y[0] \mid x )$.
>
> _Why is the distribution of the treatment effect relevant?_ Here are three examples of how the covariate-conditional probability of the treatment harm (benefit) is useful in practice and how other research streams are unable to provide this information:
>
> 1. **Medicine**. In cancer care, for instance, the conditional average treatment effect may suggest whether a treatment is beneficial _on average_ while it can not offer insights into how probable negative outcomes are. For example, consider a patient with a tumor and a drug for which the conditional average treatment effect is larger than zero, suggesting that the treatment has a benefit _on average_ (=  the tumor size reduces on average).  However, the average treatment effect does not tell us how likely such a reduction is. Given the randomness of the potential outcomes, there could be a chance that the tumor size will increase after the treatment. Hence, medical practitioners are often interested in understanding the probability of treatment benefit or harm [1,2], which is captured by the distribution of the treatment effect. This allows us to answer questions such as: _what is the probability that the treatment effect is larger than zero_? In the above example, this means how probable is a reduction of the tumor after treatment? Crucially, such questions cannot be answered by distributional treatment effects and require knowledge of _the distribution of the treatment effect_, which is the focus of our paper.
>
> 2. **Public health**. As another practical example, we refer to our case study analyzing the effectiveness of lockdowns during the COVID-19 pandemic. Here, policy-makers are interested in knowing the probability that the incidence after a strict lockdown will be lower than or equal to the incidence without it (=probability of treatment benefit) (see Appendix I).
>
> 3. **Post-approval monitoring of drugs.** Understanding the aleatoric uncertainty of the treatment effect is also relevant when monitoring the efficacy of drugs post-approval. Here, substantial increases in the aleatoric uncertainty serve as an early warning mechanism for when treatments are not working well for certain subgroups of patients or when the pharmacodynamics are not fully understood for all parts of the patient population.
>
> In sum, there are many examples – especially in medicine – where the distribution of the treatment effect is relevant for practice. Importantly, the distribution of the treatment effect is necessary in order to understand the probability of treatment benefit (or of treatment harm). Below, we also discuss why the distributional treatment effects are very different from the distribution of the treatment effect, and why only the latter can answer the above questions.
>
> **References**:
> - [1] Bordley, Robert F. "The Hippocratic Oath, effect size, and utility theory." Medical Decision Making 29.3 (2009): 377-379.
> - [2] Nicholson, Kate M., and Deborah Hellman. "Opioid prescribing and the ethical duty to do no harm." American journal of law & medicine 46.2-3 (2020): 297-310.

---

> ### Author Response · Authors · 2024-08-12
>
> [2/3]
>
>
> ### Regarding the categorization of causal quantities
>
>
> Below, we respond to your original question about the differences in causal quantities and what appears to be your main concern. We are convinced that the problem is easy to fix for the final version of the manuscript.
>
> > “However, as stated in the authors' response, this categorization is not mentioned in the paper, and no clues are provided regarding this categorization.”
>
>
> We apologize that we mentioned the categorization into the three streams of literature only as plain text, while, after reading your comment, we realized that we should have made it more explicit (e.g., by adding a formal categorization via a table). In general, three different streams are relevant to our work:
> - the AU of potential outcomes (lines 95-97),
> - the distributional treatment effects (lines 98-99), and
> - the AU of the treatment effect (lines 101-122).
>
> In the submitted version of the paper, we only made a clear cut between the identifiable causal quantities (1.)+(2.) vs. non-identifiable (3.), which was the main distinction we aimed to communicate due to reasons of space. Below, we appreciate the opportunity to explain the rationale behind our categorization.
>
> **Action:** We will revise our paper and make the above categorization in our related work section more explicit.
>
> > "Furthermore, this categorization seems counterintuitive -- does it really make sense to state that "distributional treatment effects" and "distribution of treatment effects" are very different?"
>
> Thank you for asking this important question. There are indeed two major differences between these streams:
> 1. **Interpretation**. _Distributional treatment effects_ represent the differences between different distributional aspects of the potential outcomes [3]. Hence, they can answer questions like “How are 10% of the worst-possible outcomes _with treatment_ different from the worst 10% of the outcomes _without treatment_?”. Here, the two groups (treated and untreated) of the worst 10% contain, in general, **different individuals**. This is problematic in many applications like clinical decision support and drug approval. Here, the aim is not to compare individuals from treated vs. untreated groups (where the groups may differ due to various, unobserved reasons). Instead, the aim is to accurately quantify the treatment response for each individual and allow for quantification of the personalized uncertainty of the treatment effect.    The latter is captured in the distribution of the treatment effect, which allows us to answer the question about the CDF/quantiles of the treatment effect. For example, we would aim to answer a question like “What are the worst 10% of values of the treatment effect?”. Here, we focus on the treatment effect **for every single individual**. The latter is more complex because we reason about the difference of two potential outcomes simultaneously. Hence, in natural situations when the potential outcomes are non-deterministic, both (a) the distributional treatment effect and (b) the distribution of the treatment effect will lend to _very different_ interpretations, especially in medical practice. In particular, the distribution of the treatment effect (which we study in our paper) is important in medicine, where it allows quantifying the amount of harm/benefit after the treatment [4]. This may warn doctors about situations where the averaged treatment effects are positive but where the probability of the negative treatment effect is still large.
> 2. **Inference**. The efficient inference of the distributional treatment effects only requires the estimation of the relevant distributional aspects of the conditional outcomes distributions (e.g., quantiles) and the propensity score [1]. However, in our setting of the bounds on the  CDF/quantiles of the treatment effect, we also need to perform sup/inf convolution of the CDF/quantiles of the conditional outcomes distributions. Hence, while the definitions of (a) the distributional treatment effects and (b) the distribution of the treatment effect appear related, their estimation is very different.
> As you can see above, the distributional treatment effects and the distribution of the treatment effect are related to different questions in practice and help in different situations.
>
> **References**:
> - [3] Kallus, Nathan, and Miruna Oprescu. "Robust and agnostic learning of conditional distributional treatment effects." International Conference on Artificial Intelligence and Statistics. PMLR, 2023.
> - [4] Nathan Kallus. “What’s the harm? Sharp bounds on the fraction negatively affected by treatment”. In: Advances in Neural Information Processing Systems. 2022.

---

> ### Author Response · Authors · 2024-08-12
>
> [3/3]
>
> > "However, the categorization seems to come out of nowhere and requires justification."
>
> Thank you for the question. Our justification for the above categorization of causal quantities is based on the following rationale.
> 1. **The AU of the potential outcomes:** The distribution of the treatment is non-identifiable, and, hence, we wanted to first distinguish our work from causal quantities that are identifiable. The reason is that the latter can be addressed by point identification, while the former (our problem) must be addressed by partial identification or making stronger assumptions.
> 2. **The distributional treatment effects:** Here, we want to distinguish our work from causal quantities that are contrasts between AUs of both the potential outcomes. Thereby, we aim to spell out clearly that the distributional treatment effects and our distribution of the treatment effect are both very different interpretationally and inferentially.
> 3. **The distribution of the treatment effect**: Here, we aim to survey works related to our setting, namely, the distribution of the treatment effect.
>
> Importantly, the above categorization is neither universal nor final but we used it as an informal guidance to structure the related work in our paper. If there is a better way to categorize the causal quantities, we would be happy to incorporate it into our paper.
>
> **Action**: We will spell out the rationale for the categorization in our Related Work section more clearly.
>
>
> Again, we are sorry for misinterpreting your initial question and for the confusion this has caused. We hope that we have addressed all of your concerns and assure you that they can be easily fixed in our revised manuscript. Should you have any further questions, please let us know so -- we would do our best to answer them promptly. Thanks again for reviewing our submission.

---

### Official Review · Reviewer_d6jE · 2024-07-07

**Soundness:** 4
**Presentation:** 4
**Contribution:** 3
**Rating:** 7
**Confidence:** 4

**Summary:**

The authors propose a partial identification of quantiles of the individual treatment effect, which are not point-identifiable in general. The authors justifiably argue that characterizing the distribution of individual treatment effects gives a better idea of the aleatoric uncertainty in a causal-inference problem. The proposed method is doubly robust, works with heterogeneous treatment effects, and requires minimal additional assumptions about the data.

**Strengths:**

The problem of aleatoric uncertainty in causal effects is clearly important, the contribution is significant, and the presentation is solid and concise.

 * Explanations of aleatoric versus empirical uncertainty are clear.
 * The numerous diagrams are informative.
 * Algorithm 1 is also easy to understand and succinct.

**Weaknesses:**

* The benefit and/or novelty of the CA-learner is questionable (also reflected in the results Table 2) and I wonder if its exposition is taking up valuable space. It is an insightful point on lines 182--185 that learning Makarov bounds could benefit from inductive biases of lower heterogeneity than the conditional CDFs. However, it is unclear if the CA-learner loss really incorporates that inductive bias and if so, how much.

 * The conclusion is a bit grand. Arguably, previous papers like those highlighted in Table 1 have proposed robust methods for quantifying a version of aleatoric uncertainty of causal effects.

**Questions:**

My main question has to do with recent related work. In particular, Ji et al. [52] appear to solve a similar problem, and the quick dismissal of that approach in this paper because they "made special optimization assumptions" needs further discussion. It would be helpful to spell out what these assumptions are in a concrete sense. In doing so, the authors could also discuss whether these two approaches have any fundamental commonalities, or if the partial identifications are expected to be materially different. Finally, it would be nice to see [52] appear as a baseline in the empirical evaluations, although I understand this could be difficult to implement, especially since [52] is relatively recent.

---

> ### Author Rebuttal · Authors · 2024-08-04
>
> Thank you for your positive review! It is great to hear that you found our contribution significant.
>
> ### Response to weaknesses
>
> **Benefit of the CA-learner**.  We introduce the CA-learner only as an interim step to develop the full AU-learner. We follow the classical hierarchy of learners, which were developed for other individualized treatment effect estimators [1-4], namely: “plug-in leaner -> IPTW-learner/CA-learner -> DR-learner”. Additionally, the CA-learner serves as a natural ablation of the full AU-learner.
>
> Regarding the incorporation of the inductive bias: both the CA- and AU-learners are able to do so by adding a regularization to the working model at the second stage (importantly, this regularization is independent of the regularization of the nuisance models). In our instantiations of CA-CNFs and AU-CNFs, we employed an exponential smoothing of model weights [5] as a regularizer of the working model (see Appendix E.2 Implementation (Training) for further details).
>
> **Action**. We will spell out these important details in the manuscript more clearly.
>
> **Conclusion**. The methods listed in Table 1 are _either_ (a) robust but aimed at the averaged (population level) Makarov bounds, _or_ (b) aiming at the individualized Makarov bounds but are not robust. Thus, the statement of the conclusion still holds, as _neither_ the efficient influence function _nor_ doubly-robust (orthogonal) learners for general observational data were proposed for the individualized Makarov bounds.
>
> **Action**. We will be more specific in the conclusion by saying “individualized Makarov bounds” instead of “Makarov bounds”.
>
>
> ### Response to questions
> You have raised an important question. The work of  Ji et al. [7] targets at estimating the **averaged (population level)** Makarov bounds among the other, more general, non-identifiable functionals of the joint distribution of the potential outcomes. In contrast, we focus on **individualized** Makarov bounds.
>
> Ji et al. also assume the **possibility of finding feasible Kantorovich dual functions** to the target functional (=special optimization assumption). By doing so, the authors are able to infer valid partial identification bounds on very general functionals. However, the **sharpness of [7] can not be practically guaranteed**. Specifically, feasible Kantorovich dual functions have to be chosen from a finite set of functions, which makes the bounds less sharp (this goes on top of the usual errors from estimating nuisance functions and the working model). Therefore, the solution of [7] is **completely impractical** in our setting, where the expression for sharp bounds already exists (Makarov bounds).
>
> Additionally to the above-mentioned problem: there are other practical obstacles to making  [7] a relevant baseline. For example, it is unclear how to adapt it to the individualized level Makarov bounds or how to fit the working model for several grid values of $\delta / \alpha$  at once.
>
> **Action**. We will add more background on why the work of Ji et al. [7]  is not a relevant baseline.
>
> ### References:
> - [1] Morzywolek, Pawel, Johan Decruyenaere, and Stijn Vansteelandt. "On a general class of orthogonal learners for the estimation of heterogeneous treatment effects." arXiv preprint arXiv:2303.12687 (2023).
> - [2] Vansteelandt, Stijn, and Paweł Morzywołek. "Orthogonal prediction of counterfactual outcomes." arXiv preprint arXiv:2311.09423 (2023).
> - [3]  Curth, Alicia, and Mihaela van der Schaar. "Nonparametric estimation of heterogeneous treatment effects: From theory to learning algorithms." International Conference on Artificial Intelligence and Statistics. PMLR, 2021.
> - [4] Valentyn Melnychuk, Dennis Frauen, and Stefan Feuerriegel. “Normalizing flows for interventional density estimation”. In: International Conference on Machine Learning. 2023.
> - [5] Polyak, Boris T., and Anatoli B. Juditsky. "Acceleration of stochastic approximation by averaging." SIAM journal on control and optimization 30.4 (1992): 838-855.
> - [6] Semenova, Vira. "Adaptive estimation of intersection bounds: a classification approach." arXiv preprint arXiv:2303.00982 (2023).
> - [7] Ji, Wenlong, Lihua Lei, and Asher Spector. "Model-agnostic covariate-assisted inference on partially identified causal effects." arXiv preprint arXiv:2310.08115 (2023).

---

> > ### Comment · Reviewer_d6jE · 2024-08-11
> >
> > I thank the authors for their helpful response and maintain my current score.

---

### Official Review · Reviewer_Cpq8 · 2024-07-10

**Soundness:** 3
**Presentation:** 2
**Contribution:** 3
**Rating:** 7
**Confidence:** 3

**Summary:**

In this paper, the authors propose a method to quantify the aleatoric uncertainty of the treatment effect.
For this, authors estimated Makarov bounds on the CDF and quantiles of the CDTE, and then showed, how one can build a learner, which has properties of Neyman-orthogonality and double robustness.
The authors proved the abovementioned theoretical properties of the resulting learner and demonstrated the usefulness of the proposed approach in a series of experiments on synthetic and real-world data.

**Strengths:**

I think the paper has the following strengths: It
- Addresses an important problem in the field of treatment effect estimation;
- Proposed a theoretically grounded method, that has useful theoretical properties, to quantify the aleatoric uncertainty of the treatment effect;
- Published code;

**Weaknesses:**

I don't have critical concerns about the paper, but I believe the following can improve the paper:
- I find the paper logically well-structured, but at the same time challenging to read, as it is too concentrated with technical details. I would suggest authors reconsider the narrative, concentrating more on the conceptual ideas in the main part, and moving technical details to the Supplementary;
- The field of treatment effect estimation is quite special and narrow (from my point of view) among the machine learning community, and it is worth adding some clarifications to the terms and notation used. For example, what is $Y$? It is just said that it is a continuous outcome, without any intuitions of what it could be.
- I feel that the paper missing the discussion of the alternatives of the proposed approach. For example, why specifically Makarov bounds were chosen, but not other possible alternatives?
- In Table 2, it is worth explicitly writing that CNF corresponds to the Plug-in learner (as it was called throughout the paper). I would suggest Plug-in-CNF (like written for IPTW-CNF).
- In Line 268 there is a minor typo: Should be Eq.13 and Eq. 14.

**Questions:**

- What are the alternatives for Makarov's bounds?
- Are such options like conformal predictions or plain confidence intervals somehow useful to estimate AU in the context of treatment effect?
- Compared to the vanilla Plug-in estimator (say Plug-in-CNF), what is the additional computational overhead of AU-learner?
- In lines 212-216, it is mentioned that CA-learned still has shortcomings (a) (and a new shortcoming (c)). In practice, how severe is the shortcoming (a) (selection bias) compared to AU-learner? Can you provide a toy example?

**Limitations:**

One of the limitations, not mentioned by authors, is that the approach requires (conditional) Normalizing Flows, and hence might not work well in the high dimensional scenario.

---

> ### Author Rebuttal · Authors · 2024-08-04
>
> Thank you for the positive review. Below, we respond to your questions.
>
> ### Response to weaknesses
> Thank you for the suggestions on how to improve the paper. We are more than happy to implement in the final version of the paper.
>
> **Action**: We will improve our paper as follows:
> - We realized that our paper draws upon two different streams (namely, uncertainty quantification and treatment effect estimation), which may make it harder to understand for reviewers unfamiliar with the terminology in causal inference. We will thus (1) add more background and more explanations around causal inference terminology (e.g., explaining the potential outcomes framework where $Y$ is typically the outcome of treatment), and (2) revisit the introduction so that it focuses on the conceptual ideas while delegating the technical details to a separate section.
> - We will proofread the typos and rename “CNF” to “Plug-in-CNF” (thank you for the suggestion!).
>
> _Choice the Makarov bounds:_ Potential alternatives to Makarov bounds vary depending on the type of aleatoric uncertainty. For example, explicit or implicit sharp bounds were proposed for:
> - The variance of the treatment effect , $\operatorname{Var}(Y[1] - Y[0])$. The sharp bounds are explicitly given by Fréchet-Hoeffding bounds [1]. We briefly discuss those in Sec. 2.
> - The interval probabilities, $\mathbb{P}(\delta_1 \le Y[1] - Y[0] \le \delta_2)$. The sharp bounds on the interval probabilities are, in general, different from Makarov bounds and are only implicitly defined [2]. We briefly discuss those in Appendix B.2.
>
> Yet, we are **not** aware of other bounds for measuring aleatoric uncertainty (e.g., kurtosis/skewness or entropy). Note that the above-mentioned bounds on different measures of uncertainty are _orthogonal_ to our work, as we consider bounds on the CDF/quantiles of the treatment effect as the main target. The bounds on the CDF  provide **more information** than the bounds on the variance but are the **first step** towards developing bounds on the interval probabilities.
>
> **Action**: We will highlight different alternatives to the Makarov bounds in our paper.
>
> ### Response to questions
>
> 1. We are happy to discuss alternatives to the Makarov bounds (see the answer above). Yet, these are not relevant to our work, as we focus on the estimation of the individualized level bounds on the CDF/quantiles of the treatment effect.
> 2. Conformal predictive intervals have a **different* objective as they are a measure of the total uncertainty (without the distinction of AU and EU). Thus, conformal predictive intervals on the treatment effect would rather relate to the **interval probabilities** with additional EU. Nevertheless, this would be a very interesting extension of our paper for future work.
>
>     **Action**: We will add this as a suggestion for future work (i.e., bridging conformal prediction and partial identification of AU).
> 3. The AU-learner additionally (1) fits a propensity score model (jointly or separately with the nuisance CNF), (2) infers the pseudo-CDFs, and (3) fits a second-stage working model. Roughly speaking, the AU-learner takes twice as much time to train compared to the “Plug-in-CNF” (as the pseudo-CDFs inference time is negligible).
>
>    **Action**: We will clarify this in Appendix H.3 (Runtime comparison).
>
> 4. Addressing the selection bias matters considerably in a low-sample regime for any CATE estimator [3], and, thus, also affects other casual quantities (e.g., individualized level Makarov bounds). The importance of addressing the selection bias can be understood with the **following toy thought experiment**. In our context, both the CA-learner and the AU-learner use the estimated nuisance CDFs, $\hat{\mathbb{F}}_a(y \mid x)$. Here, in the presence of the selection bias, $\hat{\mathbb{F}}_0$ will be oversmoothed for the treated population and  $\hat{\mathbb{F}}_1$, for the untreated, as there are very little samples to fit the corresponding CDFs. Then, the CA-learner uses the oversmoothed nuisance CDFs as they are, but the AU-learner will aim to correct this oversmoothing. The exact error of both learners due to the misspecification of their CDFs can be then verified with the help of the pathwise derivatives. We proved that, for a doubly-robust AU-learner, it is first-order insensitive to the misspecification of the nuisance CDFs (aka Neyman-orthogonality, see Theorem 2). Additionally, we derive a pathwise derivative for the CA-learner’s risk to show that it is **first-order sensitive (non-zero)** for the setting of CATE estimation (see the **rebuttal PDF**). Hence, this demonstrates a shortcoming of the CA-learner.
>
>     **Action**: We will add the toy example to explain the shortcomings of the CA-learner. Also, we will add the derivation of the pathwise derivative for the CA-learner’s risk targeting at the original Makarov bounds to demonstrate its first-order sensitivity (see the **rebuttal PDF**).
>
>
> ### Response to limitations
>
> The validation of our method requires the datasets where the ground-truth CDF are known (and thus the Makarov bounds are known). A prominent example is HC-MNIST, which is one of the largest public datasets for treatment effect estimation. Here, our methods work well and have a reasonable runtime.
> **Action**: We will state the possible extension of Makarov bounds that are tailored to high-dimensional outcomes as an idea for future work.
>
> ### References:
> - [1] Aronow, Peter M., Donald P. Green, and Donald KK Lee. "Sharp bounds on the variance in randomized experiments." (2014): 850-871.
> - [2] Firpo, Sergio, and Geert Ridder. "Partial identification of the treatment effect distribution and its functionals." Journal of Econometrics 213.1 (2019): 210-234.
> - [3] Alaa, Ahmed, and Mihaela van der Schaar. "Limits of estimating heterogeneous treatment effects: Guidelines for practical algorithm design." International Conference on Machine Learning. PMLR, 2018.

---

> > ### Comment · Reviewer_Cpq8 · 2024-08-10
> >
> > I would like to thank the authors for their detailed response to my review.
> >
> > My concerns and questions were well addressed. I believe that the revised version of the paper, with the incorporated changes, will be easier for readers to follow.
> >
> > As a result, I am raising the score from 6 to 7.

---

### Official Review · Reviewer_bvQh · 2024-07-10

**Soundness:** 3
**Presentation:** 2
**Contribution:** 3
**Rating:** 7
**Confidence:** 2

**Summary:**

The authors introduce AU-learner, a method to estimate the conditional distribution of treatment effects (CDTE) and hence capture the  variability in the treatment effect. They use Makarov bounds for partial identification and use conditional normalizing flows for estimation. Further, they show that AU-learner satisfies Neyman-orthogonality and double robustness.

**Strengths:**

(S1) The paper is well written and the authors provide a good experimental validation of the proposed method.

(S2) The authors are the first to propose a doubly robust estimator for the Makarov bounds on the distributional treatment effect.

**Weaknesses:**

(W1) I would have liked a more comprehensive discussion of how the proposed bounds compare against existing ones in terms of sharpness. For instance, it is not clear to me if the bounds you get are sharper than [58] in the binary outcomes setting.

(W2) I think it would be useful to compare the proposed methods against the non-doubly robust estimators. In practical settings both outcome and propensity models are likely misspecified, hence the main advantage of doubly robust estimators are the faster theoretical rates. In this setting (with both nuisances misspecified), it is not clear if the doubly robust version of the bounds is always better in practice and some more experiments would be nice to explore this.

(Minor) In the extended related work on partial identification and sensitivity models (line 658) you could also mention more recent approaches that incorporate RCT data together with proxy and instrumental variables, e.g. the works of [1] and [2].


[58] Nathan Kallus. “What’s the harm? Sharp bounds on the fraction negatively affected by treatment”. In: Advances in Neural Information Processing Systems. 2022.

[1] Falsification of Internal and External Validity in Observational Studies via Conditional Moment Restrictions. Hussain et al. AISTATS 2023.


[2] Hidden yet quantifiable: A lower bound for confounding strength using randomized trials. De Bartolomeis et al. AISTATS 2024.

**Questions:**

(Q1) For a binary outcome, could you comment on how your bounds relate to Kallus [58]? Why don't you consider them as a baseline (say for estimating bounds on the quantiles of the treatment effect distribution)?

**Limitations:**

Yes

---

> ### Author Rebuttal · Authors · 2024-08-04
>
> Thank you for your positive review and the interesting questions. It’s great that you found our paper well-written and that you appreciate the theoretical contributions.
>
> ### Response to weaknesses
> **(W1)** This is an interesting question. In our paper, we adopted the Makarov bounds, which were shown to be sharp for the left-continuous definition of the CDF ($\mathbb{P}(Y < y)$) in [1] and (very recently) for the right-continuous definition of the CDF ($\mathbb{P}(Y \le y)$) in [2]. This distinction does not matter in our context of (absolutely) continuous outcome, and, hence, we used the traditional, right-continuous definition. Yet, it matters for the binary, ordinal, and, more generally, mixed-type outcomes. Given the results of [2], we show how the Makarov bounds for binary variables **exactly match** the ones, proposed by Kallus for the fraction of the negatively affected (FNA) in [3]. Due to space limitations, we moved the full derivation to **the rebuttal PDF**
>
> **Action**: We will add the theoretical derivation from our above to the Appendix of our camera-ready version so that we show the correspondence of the FNA bounds and our Makarov bounds.
>
> **(W2)** Thank you for the suggestion to highlight the advantages of the doubly-robust estimators. Upon reading your comment, we realized that we should have explained our experiments better because, therein, we actually show that the doubly robust learner is almost always better in practice. In our paper, we actually report a comparison of the doubly-robust AU-learner and other non-doubly-robust (two-stage) learners: CA-learner and IPTW-learner. Notably, in **all the experiments** and for all learners, each of the nuisance functions can be considered to be **misspecified due to the low-sample uncertainty**. Should you see the need for further experiments, we would be glad to include them.  Importantly, our AU-learner is asymptotically guaranteed to achieve the best performance among the  CA- and IPTW-learners even when both the nuisance functions are misspecified, as the second-order estimation error is the multiplication of both nuisances second-order errors (=double-robustness).
>
> Also, we would like to mention, that doubly-robust learners are only guaranteed to be optimal asymptotically. In very low-sample regimes, another learner can be preferable [4]. Yet, to reliably check it, we would need the ground-truth counterfactuals.
>
> **Action**: We will add the above clarifications to the camera-ready version of the paper to highlight the benefits of the doubly-robust estimators.
>
> **(Minor)** Thank you – great idea! We will extend the related work in the Appendix with the above-mentioned streams of work.
>
> ### Response to questions
> **(Q1)** To answer your question, we first would like to point to our answer in **(W1)** where we now show that the Makarov bounds for the binary outcome exactly match the ones proposed in [3]. **Therefore, our work can be seen as a generalization of [3] in three novel directions:** (1) from binary to the continuous outcomes; (2) from population to individualized level; (3) targeting at the bounds for multiple values of $\delta$, not just $\delta = 0$ as in [3]. Hence, [3] is too limited and, thus, not a relevant baseline in our setting.
>
> **Action**: We will add the above explanation to our paper.
>
>
> ### References:
> - [1]  Williamson, R. C. and T. Downs (1990). “Probabilistic Arithmetic I: Numerical Methods for Calculating Convolutions and Dependency Bounds”. International Journal of Approximate Reasoning 4, 89-158.
> - [2] Zhang, Zhehao, and Thomas S. Richardson. "Bounds on the Distribution of a Sum of Two Random Variables: Revisiting a problem of Kolmogorov with application to Individual Treatment Effects." arXiv preprint arXiv:2405.08806 (2024).
> - [3] Kallus, Nathan. "What's the harm? Sharp bounds on the fraction negatively affected by treatment." Advances in Neural Information Processing Systems 35 (2022): 15996-16009.
> - [4] Curth, Alicia, and Mihaela van der Schaar. "Nonparametric estimation of heterogeneous treatment effects: From theory to learning algorithms." International Conference on Artificial Intelligence and Statistics. PMLR, 2021.

---

> > ### Comment · Reviewer_bvQh · 2024-08-07
> >
> > I thank the authors for their response, the comparison to previous bounds cleared my doubts. I maintain my original score of accept.

---

### Official Review · Reviewer_U9cv · 2024-07-12

**Soundness:** 4
**Presentation:** 3
**Contribution:** 3
**Rating:** 7
**Confidence:** 3

**Summary:**

The authors study the distribution over the individual treatment effect for binary treatments, continuous outcomes, and observed, potentially high-dimensional confounders. The authors build up on prior work on Makarov bounds, to develop a new method to lower/upper-bound the conditional CDF and the quantile function of the treatment effect. They develop a doubly robust learner and perform a theoretical and empirical evaluation. In a series of experiments, They compare several instances of their method with a baseline based on kernel density estimation. While the bound-based methods clearly outperform the baseline, they yield rather similar results.

**Strengths:**

The authors clearly state the problem and derive a viable solution. They first argue that learning estimators of the CDFs of the two potential outcomes, and plugging them into the bound computation does not yield optimal bounds. Instead, they then propose to learn the outcome CDFs which directly target the Makarov bounds. Third, they augment this loss with a term accounting for selection bias, which yields a doubly robust learner. Finally, the authors derive an implementation based on neural normalizing flows.

The authors present a great set of supplementary material including additional discussion of related work, additional experiments, and implementation details. The material might be sufficient for a journal publication.

**Weaknesses:**

The plugin estimator based on a conditional normalizing flow seems to perform rather well in the experiments. The CA learner w/o bias correction seems to perform as good as the AU learner (slightly worse in Table 2, slightly better in Table 4 in the appendix). The added complexity of the AU learner, in particular, the one-step bias correction which comes with an additional tuning parameter, renders the practical value questionable. It seems that a lot of work went into the AU learner; but it might still be worth focusing the presentation more on the simpler CA learner (e.g., discussing the role of g in the main doc).

**Questions:**

I do not fully understand the role of the working model g \in G in the CA learner. The Makarov bound is a CDF and G may include all possible CDFs; so if G is rich enough, the CA learner may return the same CDF estimate as the plugin estimator. If we restrict G, why does this cause tighter bounds?

**Limitations:**

NeurIPS Paper Checklist is provided; no concerns.

---

> ### Author Rebuttal · Authors · 2024-08-04
>
> Thank you for the positive review. In the following, we respond to the weaknesses and questions.
>
> ### Response to weaknesses
>
> Thank you. We will follow your suggestion and expand our explanation around the CA-learner. Here, we would like to give more intuition as to why we think that the AU-learner has clear benefits over the CA-learner. The reason is due to the differences in low-sample performance vs. asymptotic properties. Importantly, the best low-sample learner and the asymptotically best learner can, in general, be different [2], and there is no single “one-fits-all” data-driven solution to choose the former one [1]. Therefore (as you mentioned), in some experiments, the CA-learner or even the plug-in approach are performing nearly as well or even sometimes better than the AU-learner. This can be explained by too small data sizes or the severe overlap violations (as is the case with the IHDP dataset [6] and the results in Table 4). Yet, **only our doubly-robust AU-learner offers asymptotic properties** in the sense that it is asymptotically closest to the oracle (see Figure 4). We thus argue for a pragmatic choice in practice (i.e., in the absence of ground-truth counterfactuals or additional RCT data) where our AU-learner should be the preferred method for the individualized Makarov bounds even in low-sample data.
>
> Regarding the additional tuning parameter $\gamma$: we found the fixed values to work well in **all of the synthetic and semi-synthetic experiments** (except for the IHDP dataset, where the overlap assumption is violated).
>
> **Action**: We will follow your suggestion and expand our explanation of the CA-learner in comparison to the AU-learner (e.g., discussing the role of $\gamma$). We further will clarify the distinction between low-sample performance and asymptotic properties to better motivate the relevance of the AU-learner.
>
>
> ### Response to Questions
>
> Indeed, by postulating a restricted working model class, $g \in \mathcal{G}$, we might compromise on the sharpness and end up having looser bounds. Yet, this looseness bias is only relevant in the infinite data regime. In the finite-sample regime, the feasibility of the low-error estimation is a much more important problem. Thus, by having a projection on the restricted model class $g \in \mathcal{G}$, we have a significantly lower variance of estimation than the variance from the unrestricted plug-in model (as confirmed by the experiments in the paper).
>
> The above-mentioned bias-variance tradeoff becomes directly apparent when we assume an inductive bias that **the treatment effect is less heterogeneous than both of the potential outcomes**. Such inductive bias is widely made in the literature on CATE estimation [3-5]. In extreme cases, the treatment effect can be – on average – zero with constant Makarov bounds, while the conditional CDFs would depend on the covariates in a complicated manner. In this case, the plug-in (single-stage) learner would suffer from high variance and there is no direct way to regularize it without compromising on the fit of the conditional CDFs. We refer to Fig. 2 of [3] with an analogous example for the plug-in learner and DR-leaner for CATE estimation. Hence, this illustrates the benefits of having a restricted working model class, $g \in \mathcal{G}$.
>
> Importantly, our synthetic experiments contain this inductive bias (especially, in normal and multi-modal settings). Therein, the two-stage learners systematically outperform the plug-in CNF learner.
>
> **Action**: We will clarify the above in the revised paper.
>
> ### References:
> - [1] Curth, Alicia, and Mihaela van der Schaar. "In search of insights, not magic bullets: Towards demystification of the model selection dilemma in heterogeneous treatment effect estimation." International Conference on Machine Learning. PMLR, 2023.
> - [2] Curth, Alicia, and Mihaela van der Schaar. "Nonparametric estimation of heterogeneous treatment effects: From theory to learning algorithms." International Conference on Artificial Intelligence and Statistics. PMLR, 2021.
> - [3] Kennedy, Edward H. "Towards optimal doubly robust estimation of heterogeneous causal effects." Electronic Journal of Statistics 17.2 (2023): 3008-3049.
> - [4] Morzywolek, Pawel, Johan Decruyenaere, and Stijn Vansteelandt. "On a general class of orthogonal learners for the estimation of heterogeneous treatment effects." arXiv preprint arXiv:2303.12687 (2023).
> - [5] Vansteelandt, Stijn, and Paweł Morzywołek. "Orthogonal prediction of counterfactual outcomes." arXiv preprint arXiv:2311.09423 (2023).
> - [6] Curth, Alicia, et al. "Really doing great at estimating CATE? A critical look at ML benchmarking practices in treatment effect estimation." Thirty-fifth conference on neural information processing systems datasets and benchmarks track (round 2). 2021.

---

> > ### Comment · Reviewer_U9cv · 2024-08-11
> >
> > Thanks for clarifying my question!
> >
> > I read the other reviews and the authors' response. I agree with some feedback asking to further clarifying the setting and how it compares/differs to related settings. I have no concerns about the validity and novelty of the method, and stick to my rating.

---

### Author Rebuttal · Authors · 2024-08-04

We are grateful for the insightful and high-quality reviews. We appreciate seeing that the reviewers found our paper to be “well-written”, “theoretically grounded”, “demonstrating a high level of mathematic rigour and diligence”, containing numerous informative diagrams and explanations, and with “a good experimental validation”.

We provide point-by-point responses for each reviewer below. We uploaded additional proofs and clarifications in a **PDF file**. Here, we summarize our key improvements:

1. **Theoretical explanations**. We added the connections between the Makarov bounds from our paper to the previous works, namely, bounds on the binary outcomes [Kallus, 2022] (see the **rebuttal PDF**). Also, we show that the CA-learner is first-order sensitive wrt. to the misspecification of the nuisance functions (see the **rebuttal PDF**).
 2. **Better contrast between learners**. We better clarified the distinction between low-sample performance and asymptotic properties of different learners. Also, we provided toy examples of where CA-learner can be inferior compared to our AU-leaner. Additionally, we provided more intuition on the need to use the working model, $g \in \mathcal{G}$.
3. **Better contextualization**. We provided more context on how our work is different from several other streams in the context of the aleatoric uncertainty and causal inference. We highlighted better what are the alternatives to the Makarov bounds and why they are limited / not applicable in our case. We also listed multiple future work ideas, based on our work. Additionally, we will simplify the Introduction section and sharpen the contributions, so that the significance of the paper is more clear to the general audience.

We will incorporate all changes (marked with **Action**) into the camera-ready version of our paper (if accepted). Given these improvements, we are confident that our paper provides valuable contributions to the causal machine learning literature and is a good fit for NeurIPS 2024.

---

### Decision · Program_Chairs · 2024-09-25

**Decision:**

Accept (poster)

**Comment:**

The paper introduces a method called AU-learner to estimate the conditional distribution of treatment effects (CDTE) for binary treatments and continuous outcomes. The authors use Makarov bounds for partial identification and conditional normalizing flows for estimation, developing a doubly robust learner that satisfies Neyman-orthogonality.

Pros:

+ Addresses an important problem in causal inference by quantifying aleatoric uncertainty in treatment effects

+ Develops a theoretically grounded method with useful properties (double robustness, Neyman-orthogonality)

+ Provides a comprehensive set of experiments and supplementary materials

Cons:

+ The presentation is dense with technical details, making it challenging to read and understand